# SINKHORN DISTRIBUTIONAL REINFORCEMENT LEARNING

## ABSTRACT

The empirical success of distributional reinforcement learning (RL) highly depends on the distribution representation and the choice of distribution divergence. In this paper, we propose *Sinkhorn distributional RL (SinkhornDRL)* that learns unrestricted statistics from each return distribution and then leverages Sinkhorn divergence to minimize the difference between current and target Bellman return distributions. Theoretically, we prove the convergence properties of SinkhornDRL in the tabular setting, consistent with the interpolation nature of Sinkhorn divergence between Wasserstein distance and Maximum Mean Discrepancy (MMD). We also establish the equivalence between Sinkhorn divergence and a regularized MMD, contributing to interpreting the superiority of SinkhornDRL. Empirically, we show that SinkhornDRL is consistently better or comparable to existing algorithms on the suite of 55 Atari games.

## 1 INTRODUCTION

The design of classical reinforcement learning (RL) algorithms is mainly based on the expectation of cumulative rewards that an agent observes while interacting with the environment. Recently, a new class of RL algorithms called *distributional RL* estimates the full distribution of total returns and has exhibited state-of-the-art performance in a wide range of environments, such as C51 (Bellemare et al., 2017a), Quantile-Regression DQN (QR-DQN) (Dabney et al., 2018b; Rowland et al., 2023), Implicit Quantile Networks (IQN) (Dabney et al., 2018a), Fully Parameterized Quantile Function (FQF) (Yang et al., 2019), Non-Crossing QR-DQN (Zhou et al., 2020), MMD-DRL (Nguyen et al., 2020), Spline DQN (SPL-DQN) (Luo et al., 2021) and Bayesian distributional policy gradients (Li & Faisal, 2021). Meanwhile, distributional RL has also enjoyed other benefits in risk-sensitive control (Ma et al., 2020; Dabney et al., 2018a), policy exploration (Mavrin et al., 2019; Rowland et al., 2019), offline setting (Ma et al., 2021; Wu et al., 2023), robustness (Sun et al., 2023) and optimization (Sun et al., 2022). In this work, we propose a new distributional RL family via Sinkhorn divergence (Sinkhorn, 1967) motivated by its advantages over existing algorithms.

**Advantages over Quantile-based / Wasserstein Distance Distributional RL. 1) Finer Approximation and Adaptability.** By adding regularization, Sinkhorn divergence can directly approximate (high-dimensional) Wasserstein distance via tractable computation instead of using quantile regression in Quantile-based algorithms (Dabney et al., 2018b). Quantile-based algorithms suffer from not only the curse of dimensionality, when the reward function is multi-dimensional (Zhang et al., 2021) in many cases, but also the non-crossing issue (Zhou et al., 2020), i.e., the learned quantile curves may not be non-decreasing. **2) Flexibility.** SinkhornDRL uses samples to depict the return distribution. The sample representation offers greater flexibility than quantiles, which need to specify a fixed number of quantiles and may not capture the full data complexity (Rowland et al., 2019).

**Advantages over MMDDRL. Richer Data Geometry.** The optimal transport-based Sinkhorn divergence can inherently capture the spatial or geometric layout of the distributions. In contrast, the geometry captured by MMD is highly dependent on Reproducing Kernel Hilbert space (RKHS) and the choice of kernels. The mean embedding nature of MMD and linearity in RHKS also overlook finer geometric properties of distributions, e.g., higher moment characteristics and nonlinear shapes.

**Contributions.** While Sinkhorn divergence interpolates Wasserstein distance and MMD, the distributional RL community has yet to investigate a Sinkhorn divergence-based distributional RL family comprehensively. Therefore, our proposed SinkhornDRL algorithm is not only theoretically

grounded but also timely, contributing significantly to the fast-evolving landscape of distributional RL research. Our research also paves the way for a deeper understanding of different behaviors across existing distributional RL algorithms. Below we summarize our contributions in this study:

**Methodologically,** we propose a Sinkhorn distributional RL algorithm that interpolates Quantile Regression-based and MMD distributional RL families. SinkhornDRL inherits the advantage of learning unrestricted statistics and can be easily implemented based on existing model architectures.

**Theoretically,** we prove the convergence property of SinkhornDRL in the tabular setting (introduced in Section 4.2). Beyond the existing optimal transport literature, we reveal its new equivalent form to a special regularized MMDDRL algorithm, contributing to explaining its empirical success.

**Experimentally,** we compare SinkhornDRL with typical distributional RL algorithms across 55 Atari games with a rigorous sensitivity analysis to allow its deployment.

## 2 PRELIMINARY KNOWLEDGE

### 2.1 DISTRIBUTIONAL REINFORCEMENT LEARNING

In classical RL, an agent interacts with an environment via a Markov decision process (MDP), a 5-tuple $(\mathcal{S}, \mathcal{A}, R, P, \gamma)$, where $\mathcal{S}$ and $\mathcal{A}$ are the state and action spaces. $P$ is the environment transition dynamics, $R$ is the reward function and $\gamma \in (0, 1)$ is the discount factor.

Given a policy $\pi$, the discounted sum of future rewards $Z^\pi$ is a random variable with $Z^\pi(s, a) = \sum_{t=0}^{\infty} \gamma^t R(s_t, a_t)$, where $s_0 = s$, $a_0 = a$, $s_{t+1} \sim P(\cdot|s_t, a_t)$, and $a_t \sim \pi(\cdot|s_t)$. In expectation-based RL, the action-value function $Q^\pi$ is defined as $Q^\pi(s, a) = \mathbb{E}\left[Z^\pi(s, a)\right]$, which is iteratively updated via Bellman operator $\mathcal{T}^\pi$ through $\mathcal{T}^\pi Q(s, a) = \mathbb{E}[R(s, a)] + \gamma \mathbb{E}_{s' \sim p, \pi}\left[Q\left(s', a'\right)\right]$, where $s' \sim P(\cdot|s, a)$ and $a' \sim \pi(\cdot|s')$. In contrast, distributional RL focuses on the action-value distribution, the full distribution of $Z^\pi(s, a)$, which is updated via the distributional Bellman operator $\mathfrak{T}^\pi$ through $\mathfrak{T}^\pi Z(s, a) :\stackrel{D}{=} R(s, a) + \gamma Z\left(s', a'\right)$, where the equality implies random variables of both sides are equal in distribution and $D$ denotes the distribution. The distributional Bellman operator $\mathfrak{T}^\pi$ is contractive under certain distribution divergence metrics (Elie & Arthur, 2020).

### 2.2 DIVERGENCES BETWEEN MEASURES

**Optimal Transport (OT) and Wasserstein Distance.** The optimal transport (OT) metric $W_c$ between two probability measures $(\mu, \nu)$ is defined as the solution of the linear program $W_c = \inf_{\Pi \in \mathbf{\Pi}(\mu,\nu)} \int c(x, y) \mathrm{d}\Pi(x, y)$, where $c$ is the cost function and $\Pi$ is the joint distribution with marginals $(\mu, \nu)$. The $p$-Wasserstein distance $W_p = \left(\inf_{\Pi \in \mathbf{\Pi}(\mu,\nu)} \int \|x - y\|_p \mathrm{d}\Pi(x, y)\right)^{1/p}$ is a special case of optimal transport with the Euclidean norm as the cost function. The desirable geometric property of Wasserstein distance allows it to recover full support of measures, but it suffers from the curse of dimension and computational inefficiency (Genevay et al., 2019; Arjovsky et al., 2017).

**Maximum Mean Discrepancy.** The squared Maximum Mean Discrepancy (MMD) $\mathrm{MMD}_k^2$ with the kernel $k$ is formulated as $\mathrm{MMD}_k^2 = \mathbb{E}\left[k\left(X, X'\right)\right] + \mathbb{E}\left[k\left(Y, Y'\right)\right] - 2\mathbb{E}\left[k(X, Y)\right]$, where $k(\cdot, \cdot)$ is a continuous kernel on $\mathcal{X}$. $X'$ (resp. $Y'$) is a random variable independent of $X$ (resp. $Y$). Mathematically, the "flat" geometry that MMD induces on the space of probability measures does not faithfully lift the ground distance (Feydy et al., 2019), but MMD is cheaper to compute than OT and has a smaller *sample complexity*, i.e., approximating the distance with samples of measures (Genevay et al., 2019). We provide more detailed definitions of various distribution divergences, their relationships, and related contraction results under $\mathfrak{T}^\pi$ in distributional RL in Appendix A.

**Notations.** We constantly use the unrectified kernel $k_\alpha = -\|x - y\|^\alpha$ in the MMDDRL and SinkhornDRL algorithm analysis. With a slight abuse of notations, we also use $Z_\theta$ to denote $\theta$ parameterized return distribution, and $d_p$ as the distribution divergence.

## 3 RELATED WORK

According to the choice of distribution divergences and the distribution representation ways, distributional RL algorithms can be mainly categorized into three classes.

**Categorical Distributional RL.** As the first successful distributional RL, categorical distributional RL (Bellemare et al., 2017a), e.g., C51, represents the return distribution by the categorical distribution defined on discrete fixed supports within a pre-specified interval. C51 performs favorably on the suite of Atari games, but it is inferior to Quantile Regression distributional RL proposed afterward mainly due to the expressive restriction of its pre-defining fixed supports (Dabney et al., 2018b).

**Quantile Regression (Wasserstein Distance) Distributional RL.** QR-DQN (Dabney et al., 2018b) was proposed to use quantile regression to approximate Wasserstein distance, under which the contraction property of distributional Bellman operator can be guaranteed. Given a series of fixed quantiles, QR-DQN learns the quantile values with a more flexible support range to represent a continuous distribution. IQN (Dabney et al., 2018a) utilizes an implicit model to output quantile values more expressively, instead of the fixed ones in QR-DQN, while FQF (Yang et al., 2019) further improves IQN by proposing a more expressive quantile network. However, Quantile Regression distributional RL suffers from the non-crossing issue raised in (Zhou et al., 2020), and needs to be carefully addressed, for example, by a monotonic splines (Luo et al., 2021). By contrast, SinkhornDRL aims at approximating an entropy regularized Wasserstein distance via Sinkhorn iterations (Sinkhorn, 1967) instead of quantile regression, while naturally circumventing the non-crossing issue.

**MMD Distributional RL.** Orthogonal to Quantile Regression distributional RL, MMD distributional RL (MMDDRL) (Nguyen et al., 2020) learns samples to represent the return distribution and then optimizes with MMD. The less limited statistical budget via learning samples (Rowland et al., 2019) allows MMDDRL to outperform other algorithms with predefined statistical principles, e.g., quantiles and categorical distribution. Similarly, the sample-based SinkhornDRL preserves this advantage, although Sinkhorn divergence is directly based on optimal transport. It is worthwhile to mention that SinkhornDRL tends to "interpolate" between Wasserstein and MMD distributional RL.

## 4 SINKHORN DISTRIBUTIONAL RL (SINKHORNDRL)

The algorithmic evolution of distributional RL can be primarily viewed along two dimensions (Nguyen et al., 2020). 1) Proposing new distributional RL families beyond the aforementioned three ones based on other distribution divergences with the density estimation techniques. 2) extending existing algorithms within one family by increasing the model capacity, e.g., IQN and FQF. In contrast, SinkhornDRL aims to expand algorithm families along the first dimension.

### 4.1 SINKHORN DIVERGENCE AND ALGORITHM

Sinkhorn divergence (Sinkhorn, 1967) is a tractable loss to approximate the optimal transport problem by leveraging an entropic regularization. It allows us to find a sweet trade-off that simultaneously leverages the geometry property of Wasserstein distance on the one hand, and the favorable sample complexity advantage and unbiased gradient estimates of MMD (Genevay et al., 2018; Feydy et al., 2019). We introduce the entropic regularized Wasserstein distance $\mathcal{W}_{c,\varepsilon}(\mu, \nu)$ as

$$\mathcal{W}_{c,\varepsilon}(\mu, \nu) = \min_{\Pi \in \mathbf{\Pi}(\mu,\nu)} \int c(x,y) \mathrm{d}\Pi(x,y) + \varepsilon \mathrm{KL}(\Pi | \mu \otimes \nu), \tag{1}$$

where $\mathrm{KL}(\Pi | \mu \otimes \nu) = \int \log \left( \frac{\Pi(x,y)}{\mathrm{d}\mu(x)\mathrm{d}\nu(y)} \right) \mathrm{d}\Pi(x,y)$ is a strongly convex regularization. The impact of this entropy regularization is similar to $\ell_2$ ridge regularization in linear regression that contributes to the optimization. Next, the Sinkhorn divergence between two measures $\mu$ and $\nu$ is defined as

$$\overline{\mathcal{W}}_{c,\varepsilon}(\mu, \nu) = 2\mathcal{W}_{c,\varepsilon}(\mu, \nu) - \mathcal{W}_{c,\varepsilon}(\mu, \mu) - \mathcal{W}_{c,\varepsilon}(\nu, \nu). \tag{2}$$

Sinkhorn divergence $\overline{\mathcal{W}}_{c,\varepsilon}(\mu, \nu)$ is convex, smooth and positive definite that metricizes the convergence in law (Feydy et al., 2019). In statistical physics, $\mathcal{W}_{c,\varepsilon}(\mu, \nu)$ can be re-factored as a projection problem:

$$\mathcal{W}_{c,\varepsilon}(\mu, \nu) := \min_{\Pi \in \mathbf{\Pi}(\mu,\nu)} \mathrm{KL}\left( \Pi | \mathcal{K} \right), \tag{3}$$

where $\mathcal{K}$ is the Gibbs distribution and its density function satisfies $d\mathcal{K}(x,y) = e^{-\frac{c(x,y)}{\varepsilon}} d\mu(x) d\nu(y)$. This problem is often referred to as the "static Schrödinger problem" (Léonard, 2013; Rüschendorf & Thomsen, 1998) as it was initially considered in statistical physics.

**Distributional RL with Sinkhorn Divergence and Particle Representation.** The key to applying Sinkhorn divergence in distributional RL is to leverage the Sinkhorn loss $\overline{\mathcal{W}}_{c,\varepsilon}$ to measure the distance between the current action-value distribution $Z_\theta(s,a)$ and the target distribution $\mathfrak{T}^\pi Z_\theta(s,a)$, yielding $\overline{\mathcal{W}}_{c,\varepsilon}(Z_\theta(s,a), \mathfrak{T}^\pi Z_\theta(s,a))$ for each $s,a$ pair. In terms of the representation for $Z_\theta(s,a)$, we employ the unrestricted statistics, i.e., deterministic samples, due to its superiority in MMD-DRL (Nguyen et al., 2020), instead of using predefined statistic functionals, e.g., quantiles in QR-DQN (Dabney et al., 2018b) or categorical distribution in C51 (Bellemare et al., 2017a). More concretely, we use neural networks to generate samples to approximate the return distribution. This can be expressed as $Z_\theta(s,a) := \{Z_\theta(s,a)_i\}_{i=1}^N$, where $N$ is the number of generated samples. We refer to the samples $\{Z_\theta(s,a)_i\}_{i=1}^N$ as *particles*. Then we leverage the Dirac mixture $\frac{1}{N}\sum_{i=1}^N \delta_{Z_\theta(s,a)_i}$ to approximate the true density function of $Z^\pi(s,a)$, thus minimizing the Sinkhorn divergence between the approximate distribution and its distributional Bellman target. A generic Sinkhorn distributional RL algorithm with particle representation is provided in Algorithm 1.

---

**Algorithm 1** Generic Sinkhorn distributional RL Update

**Require**: Number of generated samples $N$, the cost function $c$, hyperparameter $\varepsilon$ and the target network $Z_{\theta^*}$.

**Input**: Sample transition $(s, a, r', s')$

1: **Policy evaluation**: $a^* \sim \pi(\cdot|s')$ or **Control**: $a^* \leftarrow \arg\max_{a' \in \mathcal{A}} \frac{1}{N}\sum_{i=1}^N Z_\theta(s', a')_i$
2: $\mathfrak{T}Z_i \leftarrow r + \gamma Z_{\theta^*}(s', a^*)_i, \forall 1 \le i \le N$

**Output**: $\overline{\mathcal{W}}_{c,\varepsilon}\left(\{Z_\theta(s,a)_i\}_{i=1}^N, \{\mathfrak{T}Z_j\}_{j=1}^N\right)$ $(\text{MMD}_k^2\left(\{Z_\theta(s,a)_i\}_{i=1}^N, \{\mathfrak{T}Z_j\}_{j=1}^N\right))$

---

**Relationship with Quantile Regression DRL and MMDDRL.** Although SinkhornDRL is closely linked with Quantile Regression DRL and MMDDRL branches, we view SinkhornDRL as a new distributional RL class. As suggested in the general algorithm framework in Algorithm 1, Sinkhorn-DRL generally modifies the distribution divergence and still relies on sample representation compared with MMDDRL in the gray color. However, SinkhornDRL is fundamentally OT-based, which approximates a regularized Wasserstein distance in stark contrast to MMD. On the other hand, SinkhornDRL leverages Sinkhorn iterations to approximately evaluate the regularized Wasserstein distance, while Quantile Regression DRL utilizes quantile regression to directly approximate a Wasserstein distance. We will dive deeper to clarify their theoretical relationships in Section 4.2, including the interpolation behavior in the limiting cases and an equivalent form of SinkhornDRL with a regularized MMDDRL.

**Relationship with IQN and FQF.** One may ask that IQN and FQF have improved QR-DQN significantly and already achieved almost state-of-the-art performance, so why bother to design Sinkhorn-DRL? As mentioned earlier, QR-DQN and MMDDRL are direct counterparts for SinkhornDRL in the first statistic dimension of algorithmic evolution, while IQN and FQF along the second modeling dimension are orthogonal to our work. As discussed in (Nguyen et al., 2020), the techniques from IQN and FQF can extend both MMDDRL and SinkhornDRL naturally. For example, we can implicitly generate $\{Z_\theta(s,a)_i\}_{i=1}^N$ via applying a neural network function to $N$ samples of a base sampling distribution as in IQN, or additionally use a proposal network to learn the weights of each generated sample as in FQF. We leave these related modeling extensions as future works and study the simplest modeling choice via Sinkhorn divergence as rigorously as possible in this work.

| Algorithm | $d_p$ Distribution Divergence | Representation $Z_\theta$ | Convergence Rate of $\mathfrak{T}^\pi$ | Sample Complexity of $d_p$ |
|---|---|---|---|---|
| C51 | Cramér distance | Categorical Distribution | $\sqrt{\gamma}$ | $\searrow$ |
| QR-DQN | Wasserstein distance | Quantiles | $\gamma$ | $\mathcal{O}(n^{-\frac{1}{d}})$ |
| MMDDRL | MMD | Samples | $\gamma^{\alpha/2}$ $(k_\alpha)$ | $\mathcal{O}(1/n)$ |
| SinkhornDRL (ours) | Sinkhorn divergence | Samples | $\gamma$ $(\varepsilon \to 0)$ $\gamma^{\alpha/2}$ $(k_\alpha, \varepsilon \to \infty)$ | $\mathcal{O}(n^{\frac{\varepsilon}{\varepsilon\lfloor d/2\rfloor\sqrt{n}}})$ $(\varepsilon \to 0)$ $\mathcal{O}(n^{-\frac{1}{2}})$ $(\varepsilon \to \infty)$ |

Table 1: Properties of different distribution divergences in typical distributional RL algorithms. $d$ is the sample dimension and $\kappa = 2\beta d + \|c\|_\infty$, where the cost function $c$ is $\beta$-Lipschitz (Genevay et al., 2019). Sample complexity is improved to $\mathcal{O}(1/n)$ using the kernel herding technique (Chen et al., 2012) in MMD.

## 4.2 THEORETICAL ANALYSIS UNDER SINKHORN DIVERGENCE

In Table 1, we first summarize some properties of distribution divergences in typical distributional RL algorithms, including the convergence rate of $\mathfrak{T}^\pi$ and sample complexity, i.e., the convergence rate of a given metric between a measure and its empirical counterpart, as a function of the number of samples $n$. Our results with related convergence proof are provided in Appendix A.

**Convergence.** We denote the supremal form of Sinkhorn divergence as $\overline{\mathcal{W}}_{c,\varepsilon}^\infty(\mu,\nu)$:

$$\overline{\mathcal{W}}_{c,\varepsilon}^\infty(\mu,\nu) = \sup_{(x,a)\in\mathcal{S}\times\mathcal{A}} \overline{\mathcal{W}}_{c,\varepsilon}(\mu(x,a),\nu(x,a)). \tag{4}$$

We will use $\overline{\mathcal{W}}_{c,\varepsilon}^\infty(\mu,\nu)$ to establish the convergence of $\mathfrak{T}^\pi$ in Theorem 1.

**Theorem 1.** *We apply $\mathfrak{T}^\pi$ under Sinkhorn divergence $\overline{\mathcal{W}}_{c,\varepsilon}(\mu,\nu)$ with **the unrectified kernel** $k_\alpha := -\|x-y\|^\alpha$ **as** $-c$ ($\alpha > 0$), where $\mu,\nu \in \{Z^\pi(s,a)\}$ for $s \in \mathcal{S}$, $a \in \mathcal{A}$ in a finite MDP. Denote $\Pi^*$ as the minimizer of $\mathcal{W}_{c,\varepsilon}(\mu,\nu)$ and define the ratio $\lambda(\mu,\nu) \in (0,1]$ that satisfies $\overline{\mathcal{W}}_{c,\epsilon}(\mu,\nu) - \varepsilon KL(\Pi^*|\mu\otimes\nu) = \lambda(\mu,\nu)\overline{\mathcal{W}}_{c,\epsilon}(\mu,\nu)$. Then, we have:*

*(1) ($\varepsilon \to 0$) $\overline{\mathcal{W}}_{c,\varepsilon}(\mu,\nu) \to 2W_\alpha^\alpha(\mu,\nu)$. When $\varepsilon = 0$, $\mathfrak{T}^\pi$ is a $\gamma^\alpha$-contraction under $\overline{\mathcal{W}}_{c,\varepsilon}^\infty$.*

*(2) ($\varepsilon \to +\infty$) $\overline{\mathcal{W}}_{c,\varepsilon}(\mu,\nu) \to MMD_{k_\alpha}^2(\mu,\nu)$. When $\varepsilon = +\infty$, $\mathfrak{T}^\pi$ is $\gamma^\alpha$-contractive under $\overline{\mathcal{W}}_{c,\varepsilon}^\infty$.*

*(3) ($\varepsilon \in (0,+\infty)$), $\mathfrak{T}^\pi$ is at least $\overline{\Delta}(\gamma,\alpha)$-contractive under $\overline{\mathcal{W}}_{c,\varepsilon}^\infty$, where the MDP-dependent constant $\overline{\Delta}(\gamma,\alpha) = \gamma^\alpha \inf_{\mu,\nu}\lambda(\mu,\nu) + (1 - \inf_{\mu,\nu}\lambda(\mu,\nu)) \in [\gamma^\alpha,1)$ with $\inf_{\mu,\nu}\lambda(\mu,\nu) > 0$.*

We provide the detailed proof of Theorem 1 in Appendix B. Theorem 1 (1) and (2) are follow-up conclusions in terms of the convergence behavior of $\mathfrak{T}^\pi$ based on the interpolation relationship between Sinkhorn divergence with Wasserstein distance and MMD (Genevay et al., 2018), but we also give a rigorous analysis in the context of distributional RL for completeness. Our key theoretical contribution is the non-trivial proof for the general $\varepsilon \in (0,\infty)$, in which we conclude that $\mathfrak{T}^\pi$ is at least a $\overline{\Delta}(\gamma,\alpha)$-contractive operator and $\overline{\Delta}(\gamma,\alpha) \in [\gamma^\alpha,1)$ is a function of $\gamma$ and $\alpha$. $\overline{\Delta}(\gamma,\alpha)$ serves as a tight upper bound and can be interpreted as the interpolation between $\gamma^\alpha$ and 1 with the coefficient $\inf_{U,V}\lambda(\mu,\nu)$. For interpretation, the ratio $\lambda(\mu,\nu)$ is defined to measure the proportion of optimal transport-based distances, i.e., $2\int(x-y)^\alpha d\Pi^*(x,y) - \mathcal{W}_{c,\varepsilon}(\mu,\mu) - \mathcal{W}_{c,\varepsilon}(\nu,\nu)$, regardless of the KL regularization term over $\overline{\mathcal{W}}_{c,\varepsilon}$, which further determines the contraction speed. Concretely, $\inf_{U,V}\lambda(\mu,\nu)$ is evaluated over the return distribution set $\{Z^\pi(s,a)\}$ and is thus MDP-dependent. A key observation is that as the set $\{Z^\pi(s,a)\}$ is finite considering a finite MDP, the set $\{\lambda(\mu,\nu)\}$, with each positive element, is also finite. Therefore, we immediately have $\inf_{U,V}\lambda(\mu,\nu)$ is strictly positive, leading to $\overline{\Delta}(\gamma,\alpha) < 1$ strictly. The crux of the proof is that we show a variant of scale sensitive property of Sinkhorn divergence when $c = -\kappa_\alpha$, where the resulting non-constant scaling factor is $\overline{\Delta}^{\mu,\nu}(\gamma,\alpha) \in [\gamma^\alpha,1)$. Thanks to the finite MDP, we can always find a positive lower bound of $\lambda(\mu,\nu)$, corresponding to a uniformly upper bound $\overline{\Delta}(\gamma,\alpha) = \sup_{\mu,\nu}\overline{\Delta}^{\mu,\nu}(\gamma,\alpha) < 1$. We eventually arrive at the $\overline{\Delta}(\gamma,\alpha)$-contraction of $\mathfrak{T}^\pi$ under $\overline{\mathcal{W}}_{c,\varepsilon}^\infty$. Note that when $\varepsilon \to 0$ or $+\infty$, $\inf_{\mu,\nu}\lambda(\mu,\nu) \to 1$ (details in the proof), leading to $\gamma^\alpha$-contraction. This coincides with (1) and (2).

**Consistency with Related Theoretical Contraction Conclusions.** As Sinkhorn divergence interpolates between Wasserstein distance and MMD, its contraction property for $\varepsilon \in [0,\infty]$ also aligns well with them when $c = -k_\alpha$. Note that if we choose Gaussian kernels as the cost function, there will be no concise and consistent contraction results as Theorem 1 (3). This conclusion is also consistent with MMDDRL (Nguyen et al., 2020) ($\varepsilon \to +\infty$), where $\mathfrak{T}^\pi$ is generally not a contraction operator under MMD equipped with Gaussian kernels owing to the existence of counterexamples mentioned in (Nguyen et al., 2020). Guided by our theoretical results, we employ the rectified kernel $k_\alpha$ as the cost function and set $\alpha = 2$ in our experiments, under which $\mathfrak{T}^\pi$ holds the contraction property guaranteed by Theorem 1 (3). Empirically, SinkhornDRL in this case suggests almost state-of-the-art performance in Section 5.

**Regularized Moment Matching under Sinkhorn Divergence Associated with Gaussian Kernels.** We further examine the potential connection between SinkhornDRL with existing distributional RL families. Inspired by the similar manner in MMDDRL (Nguyen et al., 2020), we find that Sinkhorn divergence with the Gaussian kernel can also promote matching all moments between

two distributions. More specifically, Sinkhorn divergence can be rewritten as a regularized moment matching form as revealed in Proposition 1.

**Proposition 1.** *Let $X, X' \overset{i.i.d.}{\sim} \mu, Y, Y' \overset{i.i.d.}{\sim} \nu$ and $X, X', Y, Y'$ are mutually independent. For $\varepsilon \in (0, +\infty)$, we denote $\Pi_\varepsilon^*(X, Y), \Pi^*(X, X'), \Pi^*(Y, Y')$ as the optimal joint distribution $\Pi$ of evaluating $\overline{\mathcal{W}}_{c,\varepsilon}(\mu, \nu), \overline{\mathcal{W}}_{c,\varepsilon}(\mu, \mu)$ and $\overline{\mathcal{W}}_{c,\varepsilon}(\nu, \nu)$, respectively. Sinkhorn divergence $\overline{\mathcal{W}}_{c,\varepsilon}(\mu, \nu)$ associated with Gaussian kernels $k(x, y) = \exp(-(x-y)^2/(2\sigma^2))$ as $-c$, is equivalent to*

$$\overline{\mathcal{W}}_{c,\varepsilon}(\mu, \nu) \propto \sum_{n=0}^{\infty} \frac{1}{\sigma^{2n}n!} \left( \tilde{M}_n(\mu) - \tilde{M}_n(\nu) \right)^2 + \varepsilon\mathbb{E} \left[ \log \frac{\Pi_\varepsilon^*(X, Y)^2}{\Pi^*(X, X')\Pi^*(Y, Y')} \right], \quad (5)$$

*where $\tilde{M}_n(\mu) = \mathbb{E}_{x \sim \mu} \left[ e^{-x^2/(2\sigma^2)} x^n \right]$, and similarly for $\tilde{M}_n(\nu)$.*

We provide the proof of Proposition 1 in Appendix C. In summary, akin to MMDDRL associated with a Gaussian kernel (Nguyen et al., 2020), Sinkhorn divergence approximately performs a regularized moment matching scaled by $e^{-x^2/(2\sigma^2)}$.

**Equivalence to Regularized MMD Distributional RL for General Kernels.** For the general kernel function not necessarily the Gaussian one, we can still establish a connection between Sinkhorn divergence and MMD in Corollary 1. It indicates that minimizing Sinkhorn divergence between two distributions is equivalent to minimizing a regularized squared MMD.

**Corollary 1.** *For $\varepsilon \in (0, +\infty)$,*

$$\overline{\mathcal{W}}_{c,\varepsilon}(\mu, \nu) \propto MMD^2_{-c}(\mu, \nu) + \varepsilon\mathbb{E} \left[ \log \frac{\Pi_\varepsilon^*(X, Y)^2}{\Pi^*(X, X')\Pi^*(Y, Y')} \right]. \quad (6)$$

Proof of Corollary 1 is provided in Appendix C. It is worthy of noting that this equivalence is established for the general case when $\varepsilon \in (0, +\infty)$, and it does not hold in the limiting cases when $\varepsilon = 0$ or $\infty$. For example, when $\varepsilon \to +\infty$, the second part including $\varepsilon$ in Eq. 6 is not expected to dominate. This is because the regularization term would tend to 0 as $\Pi_\varepsilon^* \to \mu \otimes \nu$ when $\varepsilon \to +\infty$. In summary, even though Sinkhorn divergence was initially proposed to serve as an entropy regularized Wasserterin distance when the cost function $c = -\kappa_\alpha$, it turns out that it is equivalent to a regularized MMD for the general kernels, as revealed in Corollary 1.

## 4.3 Distributional RL via Sinkhorn Iterations

The theoretical analysis in Section 4.2 sheds light on the behavior of Sinkhorn distributional RL, but another crucial issue we need to address is how to evaluate the Sinkhorn loss effectively. Due to the Sinkhorn divergence that enjoys the geometry property of optimal transport and the computational effectiveness of MMD, we can utilize Sinkhorn's algorithm, i.e., Sinkhorn Iterations (Sinkhorn, 1967; Genevay et al., 2018), to evaluate the Sinkhorn loss. Notably, Sinkhorn iteration with $L$ steps yields a differentiable and solvable efficient loss function as the main burden involved in it is the matrix-vector multiplication, which streams well on the GPU by simply adding extra differentiable layers on the typical deep neural network, such as a DQN architecture.

Given two sample sequences $\{Z_i\}_{i=1}^N, \{\mathfrak{T}Z_j\}_{j=1}^N$ in the distributional RL algorithm, the optimal transport distance is equivalent to the form $\min_{P \in \mathbb{R}_+^{N \times N}} \left\{ \langle P, \hat{c} \rangle; P\mathbf{1}_N = \mathbf{1}_N, P^\top \mathbf{1}_N = \mathbf{1}_N \right\}$, where

---

**Algorithm 2** Sinkhorn Iterations to Approximate $\overline{\mathcal{W}}_{c,\varepsilon} \left( \{Z_i\}_{i=1}^N, \{\mathfrak{T}Z_j\}_{j=1}^N \right)$

---

**Input**: Two samples sequences $\{Z_i\}_{i=1}^N, \{\mathfrak{T}Z_j\}_{j=1}^N$, number of iterations $L$ and hyperparameter $\varepsilon$.

1: **Initialization.** $\hat{c}_{i,j} = c(Z_i, \mathfrak{T}Z_j), \mathcal{K}_{i,j} = \exp(-\hat{c}_{i,j}/\varepsilon)$ for $\forall i, j = 1, ..., N$; $b_0 \leftarrow \mathbf{1}_N$

2: **Iteration.** $a_l \leftarrow \frac{\mathbf{1}_N}{\mathcal{K}b_{l-1}}, b_l \leftarrow \frac{\mathbf{1}_N}{\mathcal{K}a_l}$ for $l = 1, 2, ..., L$

3: **Evaluation.** $\widehat{\overline{\mathcal{W}}}_{c,\varepsilon} \left( \{Z_i\}_{i=1}^N, \{\mathfrak{T}Z_j\}_{j=1}^N \right) = \langle (K \odot \hat{c})b, a \rangle$

**Return**: $\widehat{\overline{\mathcal{W}}}_{c,\varepsilon} \left( \{Z_i\}_{i=1}^N, \{\mathfrak{T}Z_j\}_{j=1}^N \right)$

---

the empirical cost function is $\hat{c}_{i,j} = c(Z_i, \mathfrak{T}Z_j)$. By adding entropic regularization on optimal transport distance, Sinkhorn divergence can be viewed to restrict the search space of $P$ in the following scaling form: $P_{i,j} = a_i \mathcal{K}_{i,j} b_j$, where $\mathcal{K}_{i,j} = e^{-\hat{c}_{i,j}/\varepsilon}$ is the Gibbs kernel defined in Eq. 3. This allows us to leverage iterations regarding the vectors $a$ and $b$. More specifically, we initialize $b_0 = \mathbf{1}_N$, and then the Sinkhorn iterations are expressed as

$$a_{l+1} \leftarrow \frac{\mathbf{1}_N}{\mathcal{K}b_l} \quad \text{and} \quad b_{l+1} \leftarrow \frac{\mathbf{1}_N}{\mathcal{K}^\top a_{l+1}}, \tag{7}$$

where $\div$ indicates an entry-wise division. It has been proven that Sinkhorn iteration asymptotically converges to the true loss in a linear rate (Genevay et al., 2018; Franklin & Lorenz, 1989; Cuturi, 2013; Jason Altschuler, 2017). We provide a detailed algorithm description of Sinkhorn iterations in Algorithm 2 and a full version in Algorithm 3 of Appendix I. With the efficient and differentiable Sinkhorn iterations, we can easily evaluate the Sinkhorn divergence. In practice, we need to choose $L$ and $\varepsilon$, and we conduct a rigorous sensitivity analysis in Section 5.

## 5 EXPERIMENTS

We demonstrate the effectiveness of SinkhornDRL as described in Algorithm 1 on the full 55 Atari 2600 games. Without increasing model capacity for a fair comparison, we leverage the same architecture as QR-DQN and MMDDRL, and replace the quantiles output with $N$ particles (samples). In contrast to MMDDRL, SinkhornDRL only changes the distribution divergence from MMD to Sinkhorn divergence, and therefore the potential superiority in the performance can be directly attributed to the advantages of Sinkhorn divergence.

**Baselines.** We choose DQN (Mnih et al., 2015) and three typical distributional RL algorithms as classic baselines, including C51 (Bellemare et al., 2017a), QR-DQN (Dabney et al., 2018b) and MMDDRL (Nguyen et al., 2020). For a fair comparison, we build SinkhornDRL and all baselines based on a well-accepted PyTorch implementation[1] of distributional RL algorithms. We reimplement MMDDRL based on its original TensorFlow implementation[2], and keep the same setting. For example, we leverage Gaussian kernels $k_h(x, y) = \exp(-(x-y)^2/h)$ with the same kernel mixture trick covering a range of bandwidths $h$ as adopted in the original MMDDRL (Nguyen et al.,

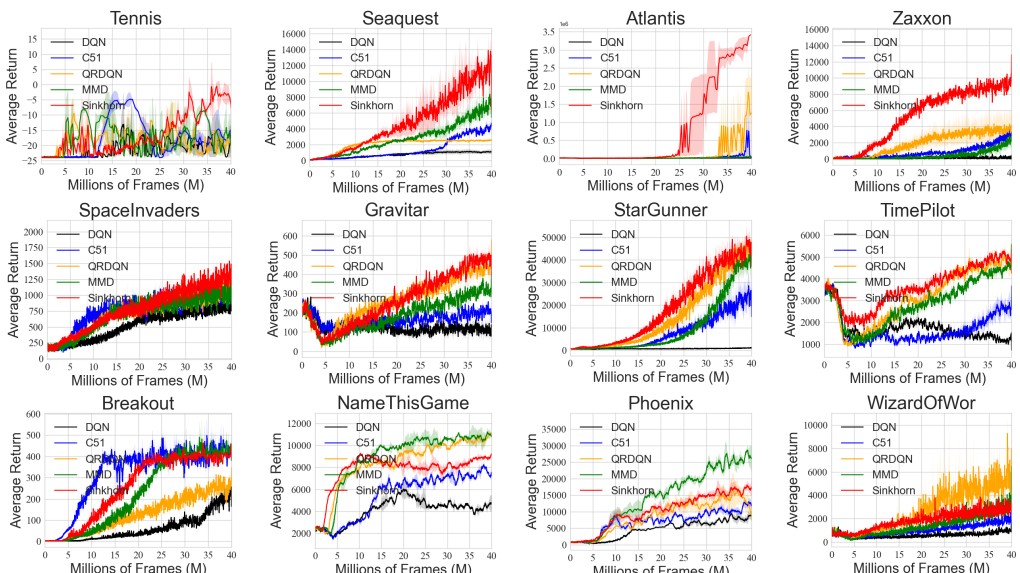

Figure 1: Learning curves of SinkhornDRL algorithm compared with DQN, C51, QR-DQN and MMD, on 12 typical Atari games averaged over 3 seeds. Games are randomly picked.

---

[1]https://github.com/ShangtongZhang/DeepRL
[2]https://github.com/thanhnguyentang/mmdrl

2020). We deploy all algorithms on 55 Atari 2600 games, and reported results are averaged over 3 seeds with the shade indicating the standard deviation. We run 40M frames for computational convenience and report learning curves across all games in Appendix D for trustworthy results.

**Hyperparameter settings.** For a fair comparison with QR-DQN, C51 and MMDDRL, we used the same hyperparameters: the number of generated samples $N = 200$, Adam optimizer with lr $=0.00005, \epsilon_{\text{Adam}} = 0.01/32$. In SinkhornDRL, we choose the number of Sinkhorn iterations $L = 10$ and smoothing hyperparameter $\varepsilon = 10.0$ in Section 5.1 after conducting sensitivity analysis in Section 5.2. Guided by the contraction guarantee analyzed in Theorem 1, we choose the unrectified kernel as the cost function, i.e., $-c = k_\alpha$, and select $\alpha = 2$ in $k_\alpha$.

## 5.1 PERFORMANCE OF SINKHORNDRL

**Learning Curves.** Figure 1 illustrates that SinkhornDRL can achieve competitive performance across 55 Atari games compared with other baselines. Notably, SinkhornDRL significantly outperforms other distributional RL algorithms on a large number of games, e.g., the first row in Figure 1. For example, SinkhornDRL performs favorably on Tennis, while other algorithms even fail to converge. Since SinkhornDRL only modifies the distribution distance compared with MMMDRL, its empirical superiority over MMDDRL verifies the key role that the derived regularization term plays in Eq. 6 as analyzed in Corollary 1. On some games, e.g., the last row of Figure 1, SinkhornDRL is on par with MMDDRL and other baselines. We provide learning curves of all considered distributional RL algorithms on all 55 Atari games in Figure 4 of Appendix D, based on which we conclude that SinkhornDRL performs better or is comparable to existing algorithms in general.

**Human Normalized Scores (HMS).**
We also compare the mean, Interquartile Mean (IQM) (Agarwal et al., 2021) and median of best HMS in Table 2 averaged over 55 Atari games, where IQM (x%) computes the mean from x% to (1-x)% of HMS, is robust to outlier scores and more statistically efficient than Me-

|  | Mean | IQM (5%) | Median | > Human | >DQN |
|---|---|---|---|---|---|
| DQN | 438.7 % | 157.7% | 43.6 % | 17 | 0 |
| C51 | 1043.4 % | 240.7 % | 103.7 % | 26 | 42 |
| QR-DQN-1 | 1286.4 % | 298.8% | 108.6 % | 31 | 47 |
| MMDDRL | 924.6 % | 248.4% | **117.5 %** | 27 | 43 |
| SinkhornDRL | **1435.8 %** | **365.5%** | 113.0 % | 27 | 42 |

Table 2: Evaluation of *best* human-normalized scores across 55 Atari games. Results are run on 3 seeds.

dian. We evaluate our scores of algorithms after 40M frames for computational convenience. It suggests that SinkhornDRL achieves state-of-the-art mean and IQM (5%) HMS compared with other baselines. We also report raw scores across all games in Table 3 of Appendix F.

**A Ratio Improvement Analysis: On Which Environments Does SinkhornDRL Perform Better?** Owing to the interpolation nature of Sinkhorn divergence between Wasserstein distance and MMD as analyzed in Theorem 1, one may ask *on which environments does SinkhornDRL perform better or worse?* To answer this question, we conduct a ratio improvement comparison between SinkhornDRL and QRDQN / MMDDRL, respectively. In Figure 2, we sort all games by the ratio improvement of SinkhornDRL over QR-DQN (MMDDRL), and select the top 10 games. It turns out that all selected games tend to have a larger action space and more complex dynamics. In par-

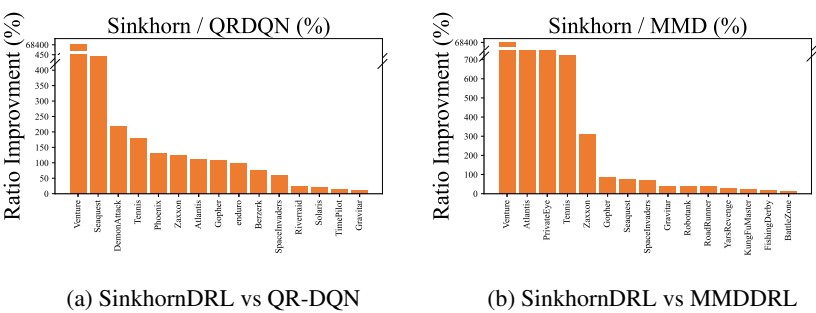

(a) SinkhornDRL vs QR-DQN                (b) SinkhornDRL vs MMDDRL

Figure 2: Ratio improvement of return for SinkhornDRL over QRDQN (left) and MMDDRL (right) averaged over 3 seeds. The ratio improvement is calculated by (SinkhornDRL - QR-DQN) / QR-DQN in (a) and (SinkhornDRL - MMDDRL) / MMDDRL in (b), respectively.

ticular, within the top 5 games for each group, including Venture, Seaquest, DemonAttack, Tennis, Phoenix, Atlantis, Privateye, and Zaxxon, all of these games have an 18-dimensional action space as well as complex dynamics, except Atlantis with 6-dimensional action space and simpler dynamics, on which MMDDRL is substantially inferior to SinkhornDRL. We provide features of all 55 games, including the number of action space, and difficulty of environment dynamics in Table 4 of Appendix G for a detailed comparison. In summary, these empirical results in the ratio improvement analysis demonstrate that *SinkhornDRL is more likely to present significant superiority over QR-DQN and MMDDRL on more complicated environments*. The empirical success of SinkhornDRL can be attributed to the interpolation advantage of Sinkhorn divergence that simultaneously makes full use of the data geometry from Wasserstein distance and the favorable sample complexity and unbiased gradient estimate property from MMD as revealed in Section 4. We also provide a ratio improvement of SinkhornDRL over all 55 Atari games in Figure 5 of Appendix E as a reference.

## 5.2 SENSITIVITY ANALYSIS AND COMPUTATIONAL COST

**Sensitivity Analysis.** In practice, a proper $\varepsilon$ is preferable as an overly large or small $\varepsilon$ will lead to numerical instability of Sinkhorn iterations in Algorithm 2, worsening its performance, as shown in Figure 3 (a). This implies that the potential interpolation nature of limiting behaviors between SinkhornDRL with QR-DQN and MMDDRL revealed in Theorem 1 may not be able to be rigorously verified in numerical experiments. SinkhornDRL also requires a proper number of iterations $L$ and samples $N$. For example, a small $N$, e.g., $N = 2$ in Seaquest in Figure 3 (b) leads to the divergence of algorithms, while an overly large $N$ can degrade the performance and meanwhile increases the computational burden (Appendix H). We conjecture that using larger networks to represent more samples is more likely to suffer from the overfitting issue, yielding the instability in the RL training (Bjorck et al., 2021). Therefore, we choose $N = 200$ to attain favorable performance and guarantee computational effectiveness at the same time. We provide more sensitivity analysis, including results on StarGunner and Zaxxon, in Appendix H.

**Computation Cost.** We compare the computation cost between SinkhornDRL and other baselines. It suggests SinkhornDRL increases around 50% computation cost compared with QR-DQN and C51, but only slightly increases the overhead (by around 20%) in contrast to MMDDRL. Due to the space limit, we provide more computation cost comparison in terms of $L$ and $N$ in Appendix H.

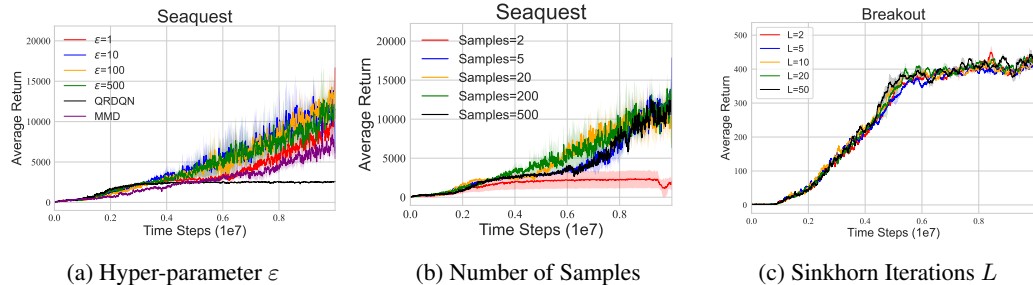

(a) Hyper-parameter $\varepsilon$      (b) Number of Samples      (c) Sinkhorn Iterations $L$

Figure 3: Sensitivity analysis of SinkhornDRL on Breakout and Seaquest in terms of $\varepsilon$, number of samples, and number of iteration $L$. Learning curves are reported over 3 seeds.

## 6 DISCUSSIONS AND CONCLUSION

Along the two dimensions of distributional RL algorithm evolution, we can further improve Sinkhorn distributional RL by incorporating implicit generative models, including parameterizing the cost function in Sinkhorn loss and increasing model capacity, which we leave as future works.

In this paper, a novel family of distributional RL algorithms based on Sinkhorn divergence is proposed that accomplishes competitive performance compared with the state-of-the-art distributional RL algorithms on the suite of Atari games. Theoretical results about the convergence guarantee and an equivalent form with a regularized MMD are provided along with rigorous empirical verification. Sinkhorn distributional RL contributes to distributional RL algorithm evolution and opens a door for new applications of Sinkhorn divergence and more optimal transport approaches.

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

## A    DEFINITION OF DISTRIBUTION DIVERGENCES AND CONTRACTION PROPERTIES

**Definition of distances.** Given two random variables $X$ and $Y$, $p$-Wasserstein metric $W_p$ between the distributions of $X$ and $Y$ is defined as

$$W_p(X, Y) = \left( \int_0^1 \left| F_X^{-1}(\omega) - F_Y^{-1}(\omega) \right|^p d\omega \right)^{1/p} = \| F_X^{-1} - F_Y^{-1} \|_p, \tag{8}$$

which $F^{-1}$ is the inverse cumulative distribution function of a random variable with the cumulative distribution function as $F$. Further, $\ell_p$ distance (Elie & Arthur, 2020) is defined as

$$\ell_p(X, Y) := \left( \int_{-\infty}^{\infty} |F_X(\omega) - F_Y(\omega)|^p \, d\omega \right)^{1/p} = \| F_X - F_Y \|_p \tag{9}$$

The $\ell_p$ distance and Wasserstein metric are identical at $p = 1$, but are otherwise distinct. Note that when $p = 2$, $\ell_p$ distance is also called Cramér distance (Bellemare et al., 2017b) $d_C(X, Y)$. Also, Cramér distance has a different representation given by

$$d_C(X, Y) = \mathbb{E}|X - Y| - \frac{1}{2}\mathbb{E}\left|X - X'\right| - \frac{1}{2}\mathbb{E}\left|Y - Y'\right|, \tag{10}$$

where $X'$ and $Y'$ are the i.i.d. copies of $X$ and $Y$. Energy distance (Székely, 2003; Ziel, 2020) is a natural extension of Cramér distance to the multivariate case, which is defined as

$$d_E(\mathbf{X}, \mathbf{Y}) = \mathbb{E}\|\mathbf{X} - \mathbf{Y}\| - \frac{1}{2}\mathbb{E}\|\mathbf{X} - \mathbf{X}'\| - \frac{1}{2}\mathbb{E}\|\mathbf{Y} - \mathbf{Y}'\|, \tag{11}$$

where $\mathbf{X}$ and $\mathbf{Y}$ are multivariate. Moreover, the energy distance is a special case of the maximum mean discrepancy (MMD), which is formulated as

$$\text{MMD}(\mathbf{X}, \mathbf{Y}; k) = \left( \mathbb{E}\left[ k\left(\mathbf{X}, \mathbf{X}'\right) \right] + \mathbb{E}\left[ k\left(\mathbf{Y}, \mathbf{Y}'\right) \right] - 2\mathbb{E}[k(\mathbf{X}, \mathbf{Y})] \right)^{1/2} \tag{12}$$

where $k(\cdot, \cdot)$ is a continuous kernel on $\mathcal{X}$. In particular, if $k$ is a trivial kernel, MMD degenerates to energy distance. Additionally, we further define the supreme MMD, which is a functional $\mathcal{P}(\mathcal{X})^{\mathcal{S} \times \mathcal{A}} \times \mathcal{P}(\mathcal{X})^{\mathcal{S} \times \mathcal{A}} \to \mathbb{R}$ defined as

$$\text{MMD}_\infty(\mu, \nu) = \sup_{(x,a) \in \mathcal{S} \times \mathcal{A}} \text{MMD}_\infty(\mu(x, a), \nu(x, a)) \tag{13}$$

We further summarize the convergence rates of the distributional Bellman operator under different distribution divergences.

- $\mathcal{T}^\pi$ is $\gamma$-contractive under the supreme form of Wassertein distance $W_p$.

- $\mathcal{T}^\pi$ is $\gamma^{1/p}$-contractive under the supreme form of $\ell_p$ distance.

- $\mathcal{T}^\pi$ is $\gamma^{\alpha/2}$-contractive under $\text{MMD}_\infty$ with the kernel $k_\alpha(x, y) = -\|x - y\|^\alpha, \forall \alpha > 0$.

**Proof of Contraction.**

- Contraction under the supreme form of Wasserstein distance is provided in Lemma 3 (Bellemare et al., 2017a).

- Contraction under supreme form of $\ell_p$ distance can refer to Theorem 3.4 (Elie & Arthur, 2020).

- Contraction under $\text{MMD}_\infty$ is provided in Lemma 6 (Nguyen et al., 2020).

## B    PROOF OF THEOREM 1

*Proof.* **1.** $\varepsilon \to 0$ **and** $c = -k_\alpha$**.** We study the uniform convergence when $\varepsilon \to 0$. The proof is summarized from the optimal transport literature (Genevay et al., 2018; Feydy et al., 2019) and we here provide the detailed proof for completeness. On the one hand, $\mathcal{W}_{c,\varepsilon} \geq \int (x-y)^\alpha d\Pi^*(x,y) dx dy \geq$

$W_\alpha^\alpha$ as KL $\geq 0$. We want to provide the inequality on the other side. Denote $\Pi'$ as the minimizer in the Wasserstein distance $W_\alpha^\alpha$. For any $\delta > 0$, there always exists a joint distribution $\Pi^\delta$ such that

$$\left| \int (x-y)^\alpha d\Pi'(x,y) - \int (x-y)^\alpha d\Pi^\delta(x,y) \right| \leq \delta$$

and $\mathrm{KL}(\Pi^\delta | \mu \otimes \nu) < +\infty$, i.e., $\int (x-y)^\alpha d\Pi^\delta(x,y) - \int (x-y)^\alpha d\Pi'(x,y) \leq \delta$. One possible way to find $\Pi^\delta$ is provided in notes of Lecture 6 in Optimal Transport Course[3] and we invite interested readers for reference. It follows that

$$W_\alpha^\alpha \leq \mathcal{W}_{c,\varepsilon} \leq \int (x-y)^\alpha d\Pi^\delta(x,y) + \varepsilon \mathrm{KL}(\Pi^\delta | \mu \otimes \nu) \leq \int (x-y)^\alpha d\Pi'(x,y) + \delta + \varepsilon \mathrm{KL}(\Pi^\delta | \mu \otimes \nu),$$

where the RHS $\int (x-y)^\alpha d\Pi'(x,y) + \delta + \varepsilon \mathrm{KL}(\Pi^\delta | \mu \otimes \nu) \to \int (x-y)^\alpha d\Pi'(x,y) + \delta = W_\alpha^\alpha + \delta$ as $\varepsilon \to 0$. As $\delta > 0$ is arbitrary, combing the two sides, it shows that $\mathcal{W}_{c,\epsilon} \to W_\alpha^\alpha$ as $\varepsilon \to 0$. Thus, Sinkhorn divergence maintains the properties of Wasserstein distance when $\varepsilon \to 0$.

When $\varepsilon = 0$, it has been shown that $W_\alpha$ can guarantee a $\gamma$-contraction property for distributional Bellman operator (Bellemare et al., 2017a). The crux of proof is that $W_\alpha$ is $\gamma$-scale sensitive:

$$W_\alpha(aU, aV) = \left( \inf_{\Pi \in \mathbf{\Pi}(\mathbf{aU}, \mathbf{aV})} \int a^\alpha(x-y)^p d\Pi(x,y) \right)^{1/p} \leq a \left( \inf_{\Pi \in \mathbf{\Pi}(\mathbf{U}, \mathbf{V})} \int (x-y)^p d\Pi(x,y) \right)^{1/p}$$

where the inequality comes from the change of optimal joint distribution. Therefore, $W_\alpha(aU, aV) \leq a W_\alpha(U, V)$ guarantees a $\gamma$-contraction property for the distributional Bellman operator. As such, for $W_\alpha^\alpha$, when $\varepsilon = 0$, it suggest that $\overline{\mathcal{W}}_{c,0} = W_\alpha^\alpha$ corresponds to a $\gamma^\alpha$-contraction.

**2.** $\varepsilon \to \infty$ **and** $c = -k_\alpha$. Our complete proof is inspired by (Ramdas et al., 2017; Genevay et al., 2018). Recap the definition of squared MMD is

$$\mathbb{E}\left[ k\left( \mathbf{X}, \mathbf{X}' \right) \right] + \mathbb{E}\left[ k\left( \mathbf{Y}, \mathbf{Y}' \right) \right] - 2\mathbb{E}[k(\mathbf{X}, \mathbf{Y})]$$

When the kernel function $k$ degenerates to an unrectified $k_\alpha(x,y) := -\|x-y\|^\alpha$ for $\alpha \in (0,2)$, the squared MMD would degenerate to

$$2\mathbb{E}\|\mathbf{X} - \mathbf{Y}\|^\alpha - \mathbb{E}\|\mathbf{X} - \mathbf{X}'\|^\alpha - \mathbb{E}\|\mathbf{Y} - \mathbf{Y}'\|^\alpha$$

where $X, X' \overset{\text{i.i.d.}}{\sim} \mu, Y, Y' \overset{\text{i.i.d.}}{\sim} \nu$ and $X, X', Y, Y'$ are mutually independent. On the other hand, by definition, we have the Sinkhorn loss as

$$\overline{\mathcal{W}}_{c,\infty}(\mu, \nu) = 2\mathcal{W}_{c,\infty}(\mu, \nu) - \mathcal{W}_{c,\infty}(\mu, \mu) - \mathcal{W}_{c,\infty}(\nu, \nu)$$

Denoting $\Pi_\varepsilon$ be the unique minimizer for $\overline{\mathcal{W}}_{c,\varepsilon}$, it holds that $\Pi_\varepsilon \to \mu \otimes \nu$ as $\varepsilon \to \infty$, which is the product of two marginal distributions. That being said, $\mathcal{W}_{c,\infty}(\mu, \nu) \to \int c(x,y) d\mu(x) d\nu(y) + 0 = \int c(x,y) d\mu(x) d\nu(y)$. If $c = -k_\alpha = \|x-y\|^\alpha$, we eventually have $\mathcal{W}_{-k_\alpha, \infty}(\mu, \nu) \to \int \|x-y\|^\alpha d\mu(x) d\nu(y) = \mathbb{E}\|\mathbf{X} - \mathbf{Y}\|^\alpha$, where $\mu$ and $\nu$ can be inherently correlated, although the minimizer degenerates to the product of the two marginal distributions. Finally, we can have

$$\overline{\mathcal{W}}_{-k_\alpha, \infty} \to 2\mathbb{E}\|\mathbf{X} - \mathbf{Y}\|^\alpha - \mathbb{E}\|\mathbf{X} - \mathbf{X}'\|^\alpha - \mathbb{E}\|\mathbf{Y} - \mathbf{Y}'\|^\alpha$$

which is exactly the form of squared MMD with the unrectified kernel $k_\alpha$. Now the key is to prove that $\Pi_\varepsilon \to \mu \otimes \nu$ as $\varepsilon \to \infty$. We give the detailed proof as follows.

Firstly, it is apparent that $\mathcal{W}_{c,\varepsilon}(\mu, \nu) \leq \int c(x,y) d\mu(x) d\nu(y)$ as $\mu \otimes \nu \in \Pi(\mu, \nu)$. Let $\{\varepsilon_k\}$ be a positive sequence that diverges to $\infty$, and $\Pi_k$ be the corresponding sequence of unique minimizers for $\mathcal{W}_{c,\varepsilon}$. According to the optimality condition, it must be the case that $\int c(x,y) d\Pi_k + \varepsilon_k \mathrm{KL}(\Pi_k, \mu \otimes \nu) \leq \int c(x,y) d\mu \otimes \nu + 0$ (when $\Pi(\mu, \nu) = \mu \otimes \nu$). Thus,

$$\mathrm{KL}\left( \Pi_k, \mu \otimes \nu \right) \leqslant \frac{1}{\varepsilon_k} \left( \int c \, d\mu \otimes \nu - \int c \, d\Pi_k \right) \to 0.$$

Besides, by the compactness of $\Pi(\mu, \nu)$, we can extract a converging subsequence $\Pi_{n_k} \to \Pi_\infty$. Since KL is weakly lower-semicontinuous, it holds that

$$\mathrm{KL}\left( \Pi_\infty, \mu \otimes \nu \right) \leqslant \lim_{k \to \infty} \inf \mathrm{KL}\left( \Pi_{n_k}, \mu \otimes \nu \right) = 0$$

---

[3]`https://lchizat.github.io/ot2021orsay.html`

Hence $\Pi_\infty = \mu \otimes \nu$. That being said that the optimal coupling is simply the product of the marginals, indicating that $\Pi_\varepsilon \to \mu \otimes \nu$ as $\varepsilon \to \infty$. As a special case, when $\alpha = 1$, $\overline{\mathcal{W}}_{-k_1,\infty}(u,v)$ is equivalent to the energy distance

$$d_E(\mathbf{X}, \mathbf{Y}) := 2\mathbb{E}\|\mathbf{X} - \mathbf{Y}\| - \mathbb{E}\|\mathbf{X} - \mathbf{X}'\| - \mathbb{E}\|\mathbf{Y} - \mathbf{Y}'\|. \tag{14}$$

In summary, if the cost function is the rectified kernel $k_\alpha$, it is the case that $\overline{\mathcal{W}}_{-k_\alpha,\varepsilon}$ converges to the squared MMD as $\varepsilon \to \infty$. According to (Nguyen et al., 2020), $\mathfrak{T}^\pi$ is $\gamma^{\alpha/2}$-contractive in the supremal form of MMD with the unrectified kernel $k_\alpha$. As $\overline{\mathcal{W}}_{c,\varepsilon}(\mu,\nu) \to \mathrm{MMD}^2_{k_\alpha}(\mu,\nu)$, which is a squared MMD instead of MMD, it implies that $\mathfrak{T}^\pi$ is $\gamma^\alpha$-contractive under the squared MMD / $\overline{\mathcal{W}}_{c,+\infty}$.

**3. For $\varepsilon \in (0, +\infty)$.** The proof is organized as follows. Firstly, we show the three properties of Sinkhorn divergence, especially the scale-sensitive property, and then we analyze the contraction of the distributional Bellman operator under Sinkhorn divergence based on its properties. Most importantly, we derive the new non-constant contraction factor of the distributional Bellman operator, whose supremum can be strictly less than 1 **in a finite MDP**.

**3.1 Properties of Sinkhorn Divergence.** We recap three crucial properties of a divergence metric. The first is *scale sensitive* (**S**) (of order $\beta$, $\beta > 0$), i.e., $d_p(cX, cY) \leq |c|^\beta d_p(X, Y)$. The second property is *shift invariant* (**I**), i.e., $d_p(A + X, A + Y) \leq d_p(X, Y)$. The last one is *unbiased gradient* (**U**). A key observation is Sinkhorn divergence would degenerate to a two-dimensional KL divergence, and therefore embraces similar properties to KL divergence. Concretely, according to the equivalent form of $\mathcal{W}_{c,\varepsilon}(\mu,\nu)$ in Eq. 3, it can be expressed as the KL divergence between an optimal joint distribution and a Gibbs distribution associated with the cost function:

$$\mathcal{W}_{c,\varepsilon}(\mu,\nu) := \mathrm{KL}\left(\Pi^*(\mu,\nu) | \mathcal{K}(\mu,\nu)\right), \tag{15}$$

where $\Pi^*$ is the optimal joint distribution. Thus, the total Sinkhorn divergence is expressed as

$$\overline{\mathcal{W}}_{c,\varepsilon}(\mu,\nu) := 2\mathrm{KL}\left(\Pi^*(\mu,\nu) | \mathcal{K}(\mu,\nu)\right) - \mathrm{KL}\left(\Pi^*(\mu,\mu) | \mathcal{K}(\mu,\mu)\right) - \mathrm{KL}\left(\Pi^*(\nu,\nu) | \mathcal{K}(\nu,\nu)\right). \tag{16}$$

Due to the form of $\overline{\mathcal{W}}_{c,\varepsilon}(\mu,\nu)$, the convergence behavior is determined by $\mathcal{W}_{c,\varepsilon}(\mu,\nu)$, which is similar to the behavior of KL divergence. According to the fact that KL divergence has unbiased gradient estimates (**U**) and shift invariant (**I**) (Dabney et al., 2018b), and Sinkhorn divergence can be viewed as a two-dimensional KL divergence, both properties of **U** and *I* can be extended to Sinkhorn divergence naturally. **However, we find the scale sensitive (S) property of KL divergence can not directly apply to Sinkhorn divergence** due to the minimum nature of $\mathcal{W}_{c,\varepsilon}(\mu,\nu)$ and the difference between optimal joint distributions of $\Pi^*(\mu,\nu)$ and $\Pi^0(a\mu, a\nu)$ where $a$ is the scale factor. On the contrary, we find Sinkhorn divergence satisfies a variant of scale-sensitive property under certain conditions, which is crucial for the convergence of the distributional Bellman operator under Sinkhorn divergence. As such, we provide a new rigorous proof of scale-sensitive property as follows.

**3.2 A New Variant of Scale Sensitive Property of Sinkhorn Divergence.** We begin our proof from $\mathcal{W}_{c,\varepsilon}$, the the key part of Sinkhorn divergence by showing $\mathcal{W}_{c,\varepsilon}$ satisfies a variant of scale sensitive property when $c = -k_\alpha$. We define a ratio $\lambda_0(U,V) \in (0,1]$ in $\mathcal{W}_{c,\varepsilon}$ as $\mathcal{W}_{c,\varepsilon}(U,V) - \varepsilon\mathrm{KL}(\Pi^*|\mu \otimes \nu) = \int (x-y)^\alpha \mathrm{d}\Pi^*(x,y) = \lambda_0(U,V)\mathcal{W}_{c,\varepsilon}(U,V)$ for all $\mu,\nu \in \{Z^\pi(s,a)\}$ for any $s \in \mathcal{S}$ and $a \in \mathcal{A}$ in the current MDP, we have the following result:

$$\mathcal{W}_{c,\varepsilon}(aU, aV) \leq \Delta(a,\alpha)\mathcal{W}_{c,\varepsilon}(U,V), \tag{17}$$

where $\Delta(a,\alpha) = 1 - \inf_{U,V}(1 - |a|^\alpha)\lambda_0(U,V) = |a|^\alpha \inf_{U,V}\lambda_0(U,V) + (1 - \inf_{U,V}\lambda_0(U,V)) \in [|a|^\alpha, 1)$ with $\inf_{U,V}\lambda_0(U,V) > 0$.

Proof. For $\inf_{U,V}\lambda_0(U,V) > 0$, a key observation is we consider a finite MDP where the return distribution set $\{Z(s,a)\}$ would be finite as well due to the finite state and action space. This implies that the ratio set $\{\lambda_0(U,V)\}$ with each element $\lambda_0(U,V) > 0$ would also be finite. Based on the fact that the real set is dense, we can directly find a positive lower bound for $\{\lambda_0(U,V)\}$. However, a more convenient way is to use the $\inf_{U,V}$, which would degenerate to $\min_{U,V}$ and therefore $\inf_{U,V}\lambda_0(U,V)$ would be strictly positive. Note that this key observation avoids the extreme case that can only lead to a non-expansive distribution Bellman operator, which we will provide more details in the following proof.

By definition of Sinkhorn divergence (Eckstein & Nutz, 2022; Peyré et al., 2019), the pdf of Gibbs kernel in the equivalent form of Sinkrhon divergence is $\mathcal{K}(U, V)$, which satisfies $\mathcal{K}(U, V) \propto e^{\frac{-c(x,y)}{\varepsilon}} \mu(x)\nu(y)$. In particular, the pdf of Gibbs kernel is defined as $\frac{d\mathcal{K}}{d(\mu \otimes \nu)}(x) = \frac{\exp(-c/\varepsilon)}{\int \exp(-c/\varepsilon)d(\mu \otimes \nu)}$, where the denominator is the normalization factor. After a scaling transformation, the pdf of $aU$ and $aV$ with respect to $x$ and $y$ would be $\frac{1}{a}\mu(\frac{x}{a})$ and $\frac{1}{a}\nu(\frac{y}{a})$. Thus $\mathcal{K}(aU, aV) \propto e^{\frac{-c(x,y)}{\varepsilon}} \frac{1}{a}\mu(\frac{x}{a})\frac{1}{a}\nu(\frac{y}{a})$. In the following proof, we use the change variable formula (multivariate version) constantly, while changing the joint pdf $\pi(x, y)$ and keep the cost function term $c(x, y)$. In particular, we denote $\Pi^*$ and $\Pi^0$ as the optimal joint distribution of $\mathcal{W}_{c,\varepsilon}(\mu, \nu)$ and $\mathcal{W}_{c,\varepsilon}(a\mu, a\nu)$. Then we have:

$$
\begin{aligned}
\mathcal{W}_{c,\varepsilon}(aU, aV) &= \int c(x, y)\mathrm{d}\Pi^0(x, y) + \varepsilon \mathrm{KL}(\Pi^0 | a\mu \otimes a\nu) \\
&\leq \int c(x, y)\mathrm{d}\Pi^*(x, y) + \varepsilon \mathrm{KL}(\Pi^* | a\mu \otimes a\nu) \\
&\overset{c=-k_\alpha}{=} \int (x - y)^\alpha \frac{1}{a^2}\pi^*(\frac{x}{a}, \frac{y}{a})\mathrm{d}x\mathrm{d}y + \varepsilon \int \frac{1}{a^2}\pi^*(\frac{x}{a}, \frac{y}{a}) \log \frac{\frac{1}{a^2}\pi^*(\frac{x}{a}, \frac{y}{a})}{\frac{1}{a^2}\mu(\frac{x}{a})\nu(\frac{y}{a})}\mathrm{d}x\mathrm{d}y \\
&= |a|^\alpha \int (x - y)^\alpha \pi^*(x, y)\mathrm{d}x\mathrm{d}y + \varepsilon \int \pi^*(x, y) \log \frac{\pi^*(x, y)}{\mu(x)\nu(y)}\mathrm{d}x\mathrm{d}y \\
&= \int (x - y)^\alpha \pi^*(x, y)\mathrm{d}x\mathrm{d}y + \varepsilon \mathrm{KL}(\Pi^* | \mu \otimes \nu) - (1 - |a|^\alpha) \int (x - y)^\alpha \pi^*(x, y)\mathrm{d}x\mathrm{d}y \\
&= \mathcal{W}_{c,\varepsilon}(U, V) - (1 - |a|^\alpha) \int (x - y)^\alpha \mathrm{d}\Pi^*(x, y) \\
&= \Delta^{U,V}(a, \alpha)\mathcal{W}_{c,\varepsilon}(U, V)
\end{aligned}
$$

(18)

where $\Delta^{U,V}(a, \alpha) = 1 - \frac{(1-|a|^\alpha) \int (x-y)^\alpha \mathrm{d}\Pi^*(x,y)}{\mathcal{W}_{c,\varepsilon}(U,V)} = 1 - (1 - |a|^\alpha)\lambda_0(U, V) \in [|a|^\alpha, 1)$ for $\varepsilon \in (0, +\infty)$ and $a < 1$ due to the fact that $0 < (1-|a|^\alpha) \int (x-y)^\alpha \mathrm{d}\Pi^*(x, y) < \int (x-y)^\alpha \mathrm{d}\Pi^*(x, y) \leq \mathcal{W}_{c,\varepsilon}(U, V)$. The equality in the last inequality ($\lambda_0(U, V) = |a|^\alpha$) holds when $U$ is independent of $V$ inherently, where the KL regularization term equals zero. $\Delta^{U,V}(a, \alpha)$ is a function less than 1, which depends on the two margin distributions, including their independence and similarity, and the scale factor $a$. **The ratio $\lambda_0(U, V)$ measures the proportion of the first optimal transport term over the whole divergence term $\mathcal{W}_{c,\varepsilon}$**, i.e., $\lambda_0 = \frac{\int (x-y)^\alpha \mathrm{d}\Pi^*(x,y)}{\mathcal{W}_{c,\varepsilon}(U,V)} \in (0, 1]$ when $\mathcal{W}_{c,\varepsilon}(U, V) \neq 0$. When $\lambda_0(U, V) = 1$, it indicates that the minimizer is the product of two marginal distributions, which happens when $U$ and $V$ are independent. When $\mathcal{W}_{c,\varepsilon}(U, V) = 0$, if and only if $U = V$ almost everywhere, the definition also holds $\mathcal{W}_{c,\varepsilon}(U, V) - \varepsilon \mathrm{KL}(\Pi^* | \mu \otimes \nu) = \int (x-y)^\alpha \mathrm{d}\Pi^*(x, y) = \lambda_0 \mathcal{W}_{c,\varepsilon}(U, V)$ for $\lambda_0(U, V) \in (0, 1]$. As such, the definition is valid and we have $\lambda_0(U, V) \in (0, 1]$ for any $U, V$, indicating that $\Delta^{U,V}(a, \alpha) \in [|\alpha|^a, 1)$.

However, the fact that $\Delta^{U,V}(a, \alpha) < 1$ can only guarantee a non-expansive contraction rather than a desirable contraction of the distributional Bellman operator. This is because there will be extreme cases in the power of series in general, although it is very unlikely to occur especially in a certain MDP with finite return distribution set. For example, denote the non-constant factor as $q_k$ for the k-th distributional Bellman update, where $q_k < 1$. We can construct a counterexample as $q_k = 1 - 1/(k + 2)^2$. In this case, $\Pi_{k=1}^{+\infty} q_k = \frac{2}{3}\frac{4}{3}\frac{3}{4}\frac{5}{4} \cdots > 0$ instead of the convergence to 0 and the non-zero limit can not guarantee the contraction. It also intuitively implies that iteratively applying distribution Bellman operator may not lead to convergence **in general by considering all possible return distributions** given the non-constant factor $\Delta^{U,V}(a, \alpha)$. Although we know these extreme cases are very unlikely to happen, we have to rule out these extreme cases for a rigorous proof. Thanks to a finite MDP, where the state and action space are finite. This implies that the return distribution set is finite as well, meaning that the set of $\lambda_0$ is also finite. Therefore, due to the fact that the real set is dense, we can always find a positive constant that can be used as the contraction factor. Alternatively, we can directly use the $\inf_{U,V} \lambda_0(U, V)$ as the uniform lower bound.

Under this condition, we can immediately find a universal upper bound of $\Delta^{U,V}(a, \alpha)$, which can be used to prove the contraction of distributional Bellman operator in the following. In particular,

we have:

$$
\begin{aligned}
\sup_{U,V} \Delta^{U,V}(a,\alpha) &= 1 - \inf_{U,V} \frac{(1-|a|^\alpha)\int(x-y)^\alpha d\Pi^*(x,y)}{\mathcal{W}_{c,\varepsilon}(U,V)} \\
&= 1 - \inf_{U,V}(1-|a|^\alpha)\lambda_0(U,V) \\
&= |a|^\alpha \inf_{U,V}\lambda_0(U,V) + 1\cdot(1-\inf_{U,V}\lambda_0(U,V)),
\end{aligned}
\tag{19}
$$

where the upper bound $\sup_{U,V}\Delta^{U,V}(a,\alpha)$ has a interpretable form eventually, which can be approximately viewed as the convex combination between $|a|^\alpha$ and 1. More importantly, $\sup_{U,V}\Delta^{U,V}(a,\alpha)$ is strictly less than 1, which is guaranteed by the finite set of $\{\lambda_0(U,V)\}$ in a finite MDP. As further explained, if the proportion of the first optimal transport term in $\mathcal{W}_{c,\varepsilon}$ is universally increasing in a given MDP, the convergence factor would tend to $\gamma^\alpha$. The extreme case is we directly apply Wasserstein distance to the power of $\alpha$ without using the KL regularization term, where coincides with our results analyzed in Sinkhorn divergence.

Following the similar procedure of $\mathcal{W}_{c,\varepsilon}$, we start to prove the scale-sensitive property of the Sinkhorn divergence $\overline{\mathcal{W}}_{c,\varepsilon}$, i.e.,

$$
\overline{\mathcal{W}}_{c,\varepsilon}(aU, aV) \leq \overline{\Delta}(a,\alpha)\overline{\mathcal{W}}_{c,\varepsilon}(U,V),
\tag{20}
$$

where $\overline{\Delta}(a,\alpha) = 1 - \inf_{U,V}\frac{(1-|a|^\alpha)(2\int(x-y)^\alpha d\Pi^*(x,y)-\mathcal{W}_{c,\varepsilon}(\mu,\mu)-\mathcal{W}_{c,\varepsilon}(\nu,\nu))}{\overline{\mathcal{W}}_{c,\varepsilon}(U,V)} \in (|a|^\alpha, 1)$. Before the scale sensitive properties proof for $\overline{\mathcal{W}}_{c,\varepsilon}$, we introduce the following Lemma.

**Lemma 1.** *Denote $\Pi^*$ as the minimizer of $\mathcal{W}_{c,\varepsilon}$, based on the dual maximization form of $\mathcal{W}_{c,\varepsilon}$, we have $2\int c(x,y)d\Pi^*(x,y) - \mathcal{W}_{c,\varepsilon}(\mu,\mu) - \mathcal{W}_{c,\varepsilon}(\nu,\nu) > 0$.*

*Proof.* We firstly show that $2W_c(\mu,\nu) - \mathcal{W}_{c,\varepsilon}(\mu,\mu) - \mathcal{W}_{c,\varepsilon}(\nu,\nu) \geq 0$, where $W_c$ is the optimal transport metric with the cost function $c$ and the Wasserstein distance $W_\alpha^a$ is the special case with a Euclidean cost function $c = -k_\alpha$. Note that the set of admissible joint distribution/couplings for **the optimal transport metric** $W_c$ is:

$$
\Pi(\mu,\nu) := \left\{ \gamma \in X \times Y \mid \gamma_{xy} \geqslant 0, \int_y \gamma_{xy}dy = \mu_x \forall x \in X, \int_x \gamma_{xy}dx = \nu_y \forall y \in Y \right\},
\tag{21}
$$

where $\gamma_{xy}$ is the admissible joint distribution. Define the Lagrangian function of the minimization problem in $W_c$ as:

$$
\begin{aligned}
\mathcal{L}(\gamma,\varphi,\psi) &:= \iint_{x,y} c(x,y)\gamma_{xy}dxdy + \int_x \varphi(x)(\mu_x - \int_y \gamma_{xy}dy)dx + \int_y \psi(y)(\nu_y - \int_y \gamma_{xy}dx)dy \\
&= \iint_{x,y}(c(x,y) - \varphi(x) - \psi(y))\gamma_{xy}dxdy + \int_x \varphi(x)\mu_x dx + \int_y \psi(y)\nu_y dy
\end{aligned}
\tag{22}
$$

We take the derivative of $\mathcal{L}(\gamma,\varphi,\psi)$ in terms of $\gamma_{xy}$, we have $c(x,y) - \varphi(x) - \psi(y)$. Thus, we have the dual form of $W_c$ as

$$
\sup_{\varphi,\psi}\min_\gamma \mathcal{L}(\gamma,\varphi,\psi) = \sup_{\varphi,\psi}\int_x \varphi(x)\mu_x dx + \int_y \psi(y)\nu_y dy
\tag{23}
$$

Define $\varphi^*$ and $\psi^*$ as the solutions of $\mathcal{W}_{c,\varepsilon}(\mu,\mu)$ and $\mathcal{W}_{c,\varepsilon}(\nu,\nu)$, respectively. Due to the independence, the minimizer of $\mathcal{W}_{c,\varepsilon}(\mu,\mu)$ and $\mathcal{W}_{c,\varepsilon}(\nu,\nu)$ would be the product of their marginal distribution and the KL term would equal 0. Therefore, we have $\mathcal{W}_{c,\varepsilon}(\mu,\mu) = 2\int_x \varphi^*(x)\mu_x dx$ and $\mathcal{W}_{c,\varepsilon}(\nu,\nu) = 2\int_y \psi^*(y)\nu_y dy$. Based on the dual maximization form in Eq. 23, we obtain $2W_c(\mu,\nu) \geq 2\int_x \varphi^*(x)\mu_x dx + 2\int_y \psi^*(y)\nu_y dy = \mathcal{W}_{c,\varepsilon}(\mu,\mu) + \mathcal{W}_{c,\varepsilon}(\nu,\nu)$. For the special case, when $c = -k_\alpha$, we have $2W_\alpha^\alpha(\mu,\nu) - \mathbb{E}_\mu\|X - X'\|^\alpha - \mathbb{E}_\nu\|Y - Y'\|^\alpha \geq 0$.

Finally, we have $2\int c(x,y)d\Pi^*(x,y) - \mathcal{W}_{c,\varepsilon}(\mu,\mu) - \mathcal{W}_{c,\varepsilon}(\nu,\nu) > 2W_\alpha^\alpha(\mu,\nu) - \mathcal{W}_{c,\varepsilon}(\mu,\mu) - \mathcal{W}_{c,\varepsilon}(\nu,\nu)$ due to the infimum nature of Wasserstein distance $W_\alpha^\alpha$ with the cost function $c(x,y) = -\|x-y\|^\alpha$, where the first inequality is strict, resulting from the non-trivial KL term in $\mathcal{W}_{c,\varepsilon}$. □

We return to the sensitive property of $\overline{\mathcal{W}}_{c,\varepsilon}$. Let $X, X' \overset{\text{i.i.d.}}{\sim} \mu, Y, Y' \overset{\text{i.i.d.}}{\sim} \nu$ and $X, X', Y, Y'$ are mutually independent, the joint distribution $\Pi$ in $\mathcal{W}_{c,\varepsilon}$ is only the multiplication of two marginal distributions from two random variables, and would degenerate to a simpler form. In particular, $\mathcal{W}_{c,\varepsilon}(\mu, \mu) = \int c(x, x')d\mu(x)d\mu(x') + 0 = -\int (x - x')^\alpha d\mu \otimes \mu$, and $\mathcal{W}_{c,\varepsilon}(\nu, \nu) = -\int (y - y')^\alpha d\nu \otimes \nu$. Based on the wisdom in Eq. 18, we immediately have $\mathcal{W}_{c,\varepsilon}(a\mu, a\mu) = |a|^\alpha \mathcal{W}_{c,\varepsilon}(\mu, \mu)$ and $\mathcal{W}_{c,\varepsilon}(a\nu, a\nu) = |a|^\alpha \mathcal{W}_{c,\varepsilon}(\nu, \nu)$ when $c = -k_\alpha$. This is because the KL term is the two distance would be equal to 0 due to the independence and the remaining Wasserstein distance satisfies the $|a|^\alpha$ scale-sensitive. Define the ratio $\lambda(U, V)$ in the full form of Sinkhorn divergence $\overline{\mathcal{W}}_{c,\varepsilon}(U, V)$ as $2\int (x - y)^\alpha d\Pi^*(x, y) - \mathcal{W}_{c,\varepsilon}(\mu, \mu) - \mathcal{W}_{c,\varepsilon}(\nu, \nu) = \overline{\mathcal{W}}_{c,\varepsilon}(U, V) - \varepsilon \mathrm{KL}(\Pi^* | \mu \otimes \nu) = \lambda(U, V)\overline{\mathcal{W}}_{c,\varepsilon}(U, V)$ with $\lambda(U, V) \in (0, 1]$. Then, we have

$$
\begin{aligned}
&\overline{\mathcal{W}}_{c,\varepsilon}(aU, aV) \\
&= 2\mathcal{W}_{c,\varepsilon}(aU, aV) - \mathcal{W}_{c,\varepsilon}(aU, aU') - \mathcal{W}_{c,\varepsilon}(aV, aV') \\
&= 2\mathcal{W}_{c,\varepsilon}(aU, aV) - |a|^\alpha \mathcal{W}_{c,\varepsilon}(\mu, \mu) - |a|^\alpha \mathcal{W}_{c,\varepsilon}(\nu, \nu) \\
&\leq 2(\mathcal{W}_{c,\varepsilon}(U, V) - (1 - |a|^\alpha)\int (x - y)^\alpha d\Pi^*(x, y)) - |a|^\alpha \mathcal{W}_{c,\varepsilon}(\mu, \mu) - |a|^\alpha \mathcal{W}_{c,\varepsilon}(\nu, \nu) \quad (24) \\
&= \overline{\mathcal{W}}_{c,\varepsilon}(U, V) - (1 - |a|^\alpha)(2\int (x - y)^\alpha d\Pi^*(x, y) - \mathcal{W}_{c,\varepsilon}(\mu, \mu) - \mathcal{W}_{c,\varepsilon}(\nu, \nu)) \\
&= \overline{\Delta}^{U,V}(a, \alpha)\overline{\mathcal{W}}_{c,\varepsilon}(U, V)
\end{aligned}
$$

where $\overline{\Delta}^{U,V}(a, \alpha) = 1 - \frac{(1 - |a|^\alpha)(2\int (x - y)^\alpha d\Pi^*(x, y) - \mathcal{W}_{c,\varepsilon}(\mu, \mu) - \mathcal{W}_{c,\varepsilon}(\nu, \nu))}{\overline{\mathcal{W}}_{c,\varepsilon}(U, V)} = 1 - (1 - |a|^\alpha)\lambda(U, V) \in (|a|^\alpha, 1)$. Thanks to Lemma 1, the numerator in $\lambda(U, V)$ would be positive, and thus $\lambda(U, V)$ would be positive as well. $\lambda(U, V)$ is typically less than or equal to one as $0 \leq 2\int (x - y)^\alpha d\Pi^*(x, y) - \mathcal{W}_{c,\varepsilon}(\mu, \mu) - \mathcal{W}_{c,\varepsilon}(\nu, \nu) \leq \overline{\mathcal{W}}_{c,\varepsilon}(U, V)$, where the only difference between the numerator and denominator for $\lambda(U, V)$ is the KL regularization term. Thus, $\lambda(U, V)$ measures the proportion of the optimal transport distance regardless of the KL regularization term over the whole Sinkhorn divergence $\overline{\mathcal{W}}_{c,\varepsilon}(U, V)$.

Still thanks to the finite MDP, the return distribution set would be finite, and thus the set of $\{\lambda(U, V)\}$ for $U, V \in \{Z(s, a)\}$ with $s \in \mathcal{S}$ and $a \in \mathcal{A}$. This leads to $\lambda(U, V) \in [\inf_{U,V} \lambda(U, V), 1]$ with $\inf_{U,V} \lambda(U, V) > 0$ strictly. Similarly, we have the following result:

$$
\begin{aligned}
\sup_{U,V} \overline{\Delta}^{U,V}(a, \alpha) &= 1 - \inf_{U,V} \frac{(1 - |a|^\alpha)(2\int (x - y)^\alpha d\Pi^*(x, y) - \mathcal{W}_{c,\varepsilon}(\mu, \mu) - \mathcal{W}_{c,\varepsilon}(\nu, \nu))}{\overline{\mathcal{W}}_{c,\varepsilon}(U, V)} \\
&\leq 1 - \inf_{U,V}(1 - |a|^\alpha)\lambda(U, V) \\
&= |a|^\alpha \inf_{U,V} \lambda(U, V) + 1 \cdot (1 - \inf_{U,V} \lambda(U, V)) \\
&\in [|a|^\alpha, 1),
\end{aligned} \quad (25)
$$

where $\sup_{U,V} \overline{\Delta}^{U,V}(a, \alpha) = |a|^\alpha$ when all the KL regularization degenerates to 0, which may happen when the two return distributions are independent. In such case, the contraction factor would degenerate to $\gamma^\alpha$, which is the exact case when we apply $\alpha$-Wasserstein distance to the power of $\alpha$.

We next define $\overline{\Delta}(a, \alpha) = \sup_{U,V} \overline{\Delta}^{U,V}(a, \alpha)$. Therefore, we have the result:

$$
\overline{\mathcal{W}}_{c,\varepsilon}(aU, aV) \leq \overline{\Delta}(a, \alpha)\overline{\mathcal{W}}_{c,\varepsilon}(U, V), \quad (26)
$$

where $\overline{\Delta}(a, \alpha) = 1 - \inf_{U,V} \frac{(1 - |a|^\alpha)(2\int (x - y)^\alpha d\Pi^*(x, y) - \mathcal{W}_{c,\varepsilon}(\mu, \mu) - \mathcal{W}_{c,\varepsilon}(\nu, \nu))}{\overline{\mathcal{W}}_{c,\varepsilon}(U, V)} = 1 - (1 - |a|^\alpha)\inf_{U,V} \lambda(U, V) = |a|^\alpha \inf_{U,V} \lambda(U, V) + (1 - \inf_{U,V} \lambda(U, V)) \in [|a|^\alpha, 1)$. This result paves the crucial path toward the convergence of the distributional Bellman operator under Sinkhorn divergence analyzed in 3.3.

**3.3 Contraction of Distributional Bellman Operator under Sinkhorn Divergence.** Based on results in 3.1 and 3.2, we derive the convergence of distributional Bellman operator $\mathfrak{T}^\pi$ under the

supreme form of $\overline{\mathcal{W}}_{c,\varepsilon}$, i.e., $\overline{\mathcal{W}}_{c,\varepsilon}^{\infty}$:

$$
\begin{aligned}
&\overline{\mathcal{W}}_{c,\varepsilon}^{\infty}(\mathfrak{T}^{\pi} Z_1, \mathfrak{T}^{\pi} Z_2)\\
&= \sup_{s,a} \overline{\mathcal{W}}_{c,\varepsilon}(\mathfrak{T}^{\pi} Z_1(s,a), \mathfrak{T}^{\pi} Z_2(s,a))\\
&= \overline{\mathcal{W}}_{c,\varepsilon}(R(s,a) + \gamma Z_1(s',a'), R(s,a) + \gamma Z_2(s',a'))\\
&= \overline{\mathcal{W}}_{c,\varepsilon}(\gamma Z_1(s',a'), \gamma Z_2(s',a'))\\
&\overset{c=-k_{\alpha}}{\leq} \Delta^{Z_1(s',a'),Z_2(s',a')}(\gamma,\alpha) \overline{\mathcal{W}}_{c,\varepsilon}(Z_1(s',a'), Z_2(s',a'))\\
&\leq \sup_{s',a'} \overline{\Delta}^{Z_1(s',a'),Z_2(s',a')}(\gamma,\alpha) \sup_{s',a'} \overline{\mathcal{W}}_{c,\varepsilon}(Z_1(s',a'), Z_2(s',a'))\\
&\leq \sup_{Z_1,Z_2} \overline{\Delta}^{Z_1(s',a'),Z_2(s',a')}(\gamma,\alpha) \overline{\mathcal{W}}_{c,\varepsilon}^{\infty}(Z_1, Z_2)\\
&= \overline{\Delta}(\gamma,\alpha) \overline{\mathcal{W}}_{c,\varepsilon}^{\infty}(Z_1, Z_2)
\end{aligned}
\tag{27}
$$

where the first inequality comes from the scale-sensitive property proof of Sinkhorn divergence and the last inequality is based on the fact the range of return distribution $Z_1$ and $Z_2$ can be larger than that for $Z_1(s,a)$ and $Z_2(s,a)$ for $\forall s \in |S|, a \in |A|$. Owing to the fact that $\overline{\Delta}(\gamma,\alpha) \in [|\gamma|^{\alpha}, 1)$ is a MDP-dependent constant, which is also determined by $\gamma$ and $\alpha$, we conclude that distributional Bellman operator is **at least** $\overline{\Delta}(\gamma,\alpha)$-contractive. Based on the existing Banach fixed point theorem, we have a unique optimal return distribution when convergence.

$\square$

## C PROOF OF PROPOSITION 1 AND COROLLARY 1

*Proof.* We leverage $\Pi_{\varepsilon}^*(\mu,\nu), \Pi^*(\mu,\mu), \Pi^*(\nu,\nu)$ to denote the optimal joint distribution $\Pi$ while evaluating Sinkhorn divergence $\overline{\mathcal{W}}_{c,\varepsilon}(\mu,\nu)$, $\overline{\mathcal{W}}_{c,\varepsilon}(\mu,\mu)$ and $\overline{\mathcal{W}}_{c,\varepsilon}(\nu,\nu)$, respectively. Let $X, X' \overset{\text{i.i.d.}}{\sim} \mu, Y, Y' \overset{\text{i.i.d.}}{\sim} \nu$ and $X, X', Y, Y'$ are mutually independent. The Sinkhorn divergence can be composed in the following form:

$$
\begin{aligned}
&\overline{\mathcal{W}}_{c,\varepsilon}(\mu,\nu)\\
&= 2\text{KL}\left(\Pi_{\varepsilon}^*(\mu,\nu)|\mathcal{K}_{-k}(\mu,\nu)\right) - \text{KL}\left(\Pi^*(\mu,\mu)|\mathcal{K}_{-k}(\mu,\mu)\right) - \text{KL}\left(\Pi^*(\nu,\nu)|\mathcal{K}_{-k}(\nu,\nu)\right)\\
&\propto 2(\mathbb{E}_{X,Y}\left[\log \Pi_{\varepsilon}^*(X,Y)\right]) + \frac{1}{\varepsilon}\mathbb{E}_{X,Y}\left[c(X,Y)\right]) - (\mathbb{E}_{X,X'}\left[\log \Pi^*(X,X')\right]) + \frac{1}{\varepsilon}\mathbb{E}_{X,X'}\left[c(X,X')\right])\\
&\quad - (\mathbb{E}_{Y,Y'}\left[\log \Pi^*(Y,Y')\right]) + \frac{1}{\varepsilon}\mathbb{E}_{Y,Y'}\left[c(Y,Y')\right])\\
&= \mathbb{E}_{X,X',Y,Y'}\left[\log \frac{\Pi_{\varepsilon}^*(X,Y)^2}{\Pi^*(X,X')\Pi^*(Y,Y')}\right] + \frac{1}{\varepsilon}(\mathbb{E}_{X,X'}\left[k(X,X')\right] + \mathbb{E}_{Y,Y'}\left[k(Y,Y')\right] - 2\mathbb{E}_{X,X'}\left[k(X,Y)\right])\\
&= \mathbb{E}_{X,X',Y,Y'}\left[\log \frac{\Pi_{\varepsilon}^*(X,Y)^2}{\Pi^*(X,X')\Pi^*(Y,Y')}\right] + \frac{1}{\varepsilon}\text{MMD}_{-c}^2(\mu,\nu)
\end{aligned}
\tag{28}
$$

where the cost function $c$ in the Gibbs distribution $\mathcal{K}$ is minus kernel in MMD. $\propto$ indicates we cancel the normalization factor in the probability density function of Gibbs distribution $\mathcal{K}(x,y) = \frac{1}{Z}e^{-c(x-y)/\varepsilon}$. Till now, we have shown the result in Corollary 1.

Next, we use Taylor expansion to prove the moment matching of MMD with the Gaussian kernel. Firstly, we have the following equation:

$$
\begin{aligned}
\text{MMD}_{-c}^2(\mu,\nu) &= \mathbb{E}_{X,X'}\left[k(X,X')\right] + \mathbb{E}_{Y,Y'}\left[k(Y,Y')\right] - 2\mathbb{E}_{X,X'}\left[k(X,Y)\right]\\
&= \mathbb{E}_{X,X'}\left[\phi(X)^{\top}\phi(X')\right] + \mathbb{E}_{Y,Y'}\left[\phi(Y)^{\top}\phi(Y')\right] - 2\mathbb{E}_{X,X'}\left[\phi(X)^{\top}\phi(Y)\right]\\
&= \mathbb{E}\|\phi(X) - \phi(Y)\|^2
\end{aligned}
\tag{29}
$$

We expand the Gaussian kernel via Taylor expansion, i.e.,

$$
\begin{aligned}
k(x,y) &= e^{-(x-y)^2/(2\sigma^2)} \\
&= e^{-\frac{x^2}{2\sigma^2}} e^{-\frac{y^2}{2\sigma^2}} e^{\frac{xy}{\sigma^2}} \\
&= e^{-\frac{x^2}{2\sigma^2}} e^{-\frac{y^2}{2\sigma^2}} \sum_{n=0}^{\infty} \frac{1}{\sqrt{n!}} \left(\frac{x}{\sigma}\right)^n \frac{1}{\sqrt{n!}} \left(\frac{y}{\sigma}\right)^n \\
&= \sum_{n=0}^{\infty} e^{-\frac{x^2}{2\sigma^2}} \frac{1}{\sqrt{n!}} \left(\frac{x}{\sigma}\right)^n e^{-\frac{y^2}{2\sigma^2}} \frac{1}{\sqrt{n!}} \left(\frac{y}{\sigma}\right)^n \\
&= \phi(x)^\top \phi(y)
\end{aligned}
\tag{30}
$$

Therefore, we have

$$
\begin{aligned}
\mathrm{MMD}^2_{-c}(\mu,\nu) &= \sum_{n=0}^{\infty} \frac{1}{\sigma^{2n} n!} \left( \mathbb{E}_{x\sim\mu}\left[ e^{-x^2/(2\sigma^2)} x^n \right] - \mathbb{E}_{x\sim\nu}\left[ e^{-y^2/(2\sigma^2)} y^n \right] \right)^2 \\
&= \sum_{n=0}^{\infty} \frac{1}{\sigma^{2n} n!} \left( \tilde{M}_n(\mu) - \tilde{M}_n(\nu) \right)^2
\end{aligned}
\tag{31}
$$

$\tilde{M}_n(\mu) = \mathbb{E}_{x\sim\mu}\left[ e^{-x^2/(2\sigma^2)} x^n \right]$, and similarly for $\tilde{M}_n(\nu)$. The conclusion is the same as the moment matching in (Nguyen et al., 2020). Finally, due to the equivalence of $\overline{\mathcal{W}}_{c,\varepsilon}(\mu,\nu)$ after multiplying $\varepsilon$, we have

$$
\begin{aligned}
\overline{\mathcal{W}}_{c,\varepsilon}(\mu,\nu;k) &\propto \mathrm{MMD}^2_{-c}(\mu,\nu) + \varepsilon\mathbb{E}\left[ \frac{(\Pi^*_\varepsilon(X,Y))^2}{\Pi^*(X,X')\Pi^*(Y,Y')} \right] \\
&= \sum_{n=0}^{\infty} \frac{1}{\sigma^{2n} n!} \left( \tilde{M}_n(\mu) - \tilde{M}_n(\nu) \right)^2 + \varepsilon\mathbb{E}\left[ \frac{(\Pi^*_\varepsilon(X,Y))^2}{\Pi^*(X,X')\Pi^*(Y,Y')} \right],
\end{aligned}
\tag{32}
$$

This result is also consistent with Theorem 1, where $\Pi^*$ would degenerate to $\mu \otimes \nu$ as $\varepsilon \to +\infty$. In that case, the regularization term would vanish, and thus the Sinkhorn divergence degrades to an MMD loss, i.e., $\mathrm{MMD}^2_{-c}(\mu,\nu)$.

$\square$

# D    LEARNING CURVES ON 55 ATARI GAMES

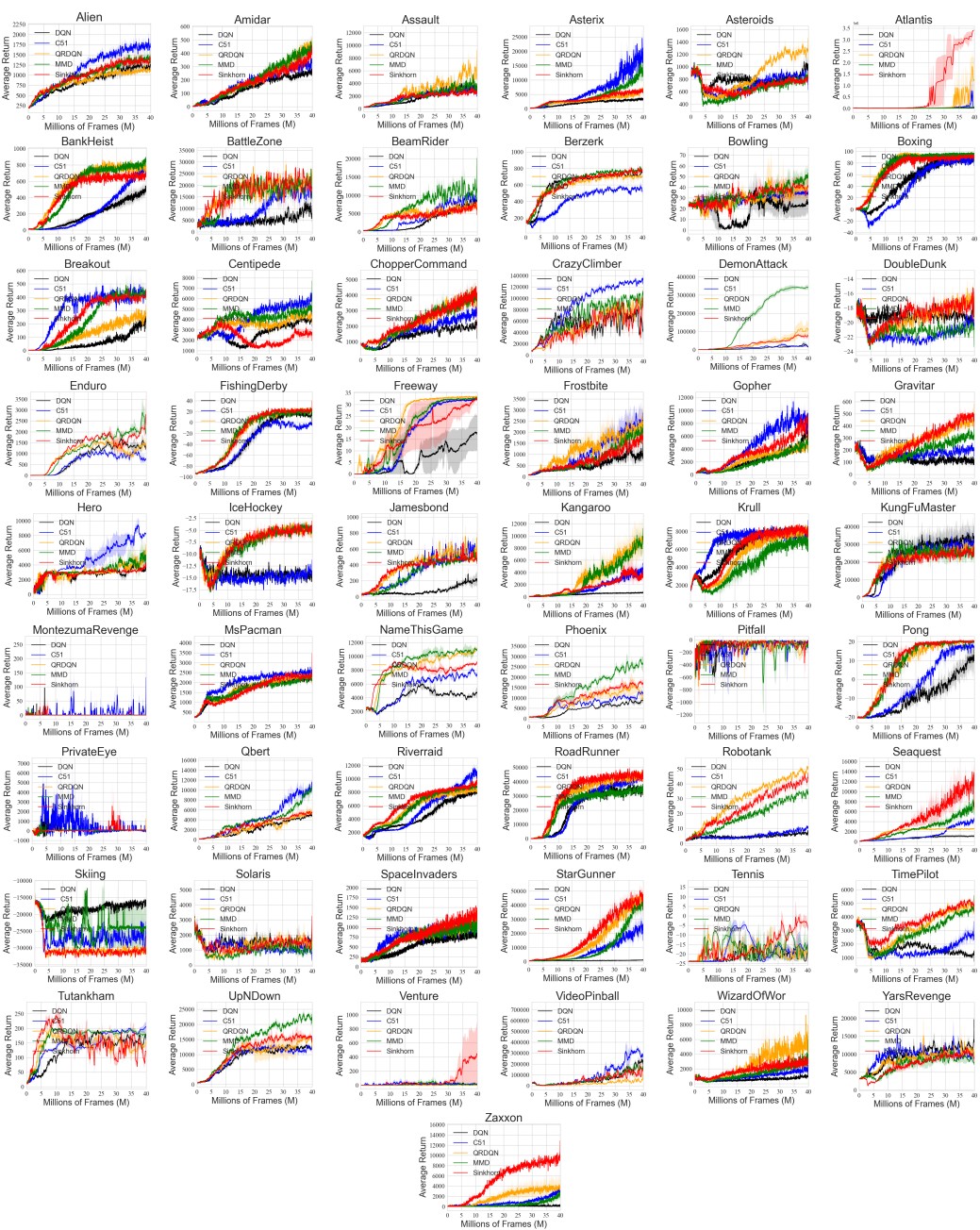

Figure 4: Learning curves of SinkhornDRL compared with DQN, C51, QRDQN and MMD on 55 Atari games after training 40M frames averaged over 3 seeds.

# E    RATIO IMPROVEMENT ANALYSIS ACROSS ALL 55 ATARI GAMES

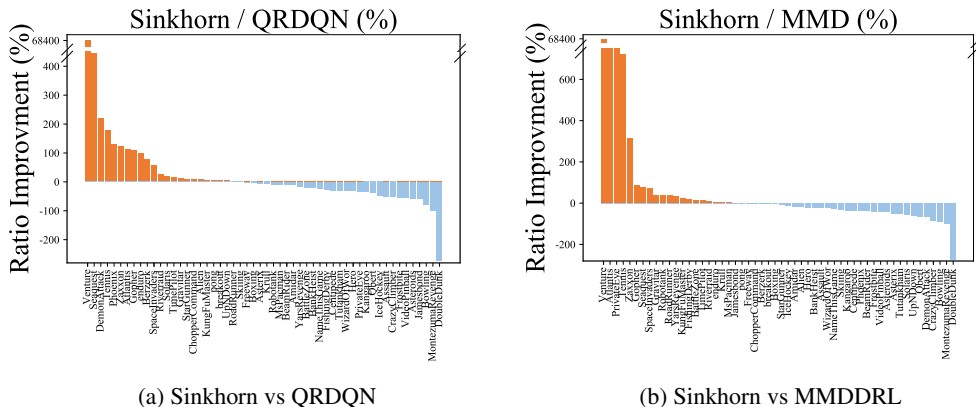

(a) Sinkhorn vs QRDQN                                  (b) Sinkhorn vs MMDDRL

Figure 5: Ratio improvement of return for Sinkhorn distributional RL algorithm over QRDQN (left) and MMDDRL (right) over 3 seeds. For example, the ratio improvement is calculated by (Sinkhorn - QRDQN) / QRDQN in the left.

We provide a ratio improvement analysis across all 55 Atari games in Figure 5. Figure 5 showcases that compared with QRDQN (left), SinkhornDRL achieves better performance across almost half of the considered games and the superiority of SinkhornDRL is significant across a large number of games, including Venture, Seaquest, Tennis and Phoenix. This empirical outperformance verifies the effectiveness of smoothing Wasserstein distance in distributional RL. In contrast with MMD-DRL, the advantage of SinkhornDRL is reduced with the performance improvement on a slightly smaller proportion of games, but a remarkable performance improvement for SinkhornDRL on a large number of games can be easily observed.

# F  RAW SCORE TABLES ACROSS ALL ATARI GAMES

| GAMES | RANDOM | HUMAN | DQN | C51 | QRDQN | MMD | Sinkhorn |
|---|---|---|---|---|---|---|---|
| Alien | 211.9 | 7,127.7 | 1334.0 | 1946.0 | 1625.0 | 2218.0 | 1873.0 |
| Amidar | 2.34 | 1,719.5 | 400.2 | 354.5 | 554.6 | 706.4 | 506.7 |
| Assault | 283.5 | 742.0 | 5651.8 | 3368.1 | 7593.6 | 6001.5 | 3771.0 |
| Asterix | 268.5 | 8,503.3 | 5490.0 | 31860.0 | 7660.0 | 15890.0 | 7610.0 |
| Asteroids | 1008.6 | 47,388.7 | 1246.0 | 826.0 | 1660.0 | 1095.0 | 624.0 |
| Atlantis | 22188 | 29,028.1 | 18990.0 | 1490040.0 | 2520080.0 | 80920.0 | 3417430.0 |
| BankHeist | 14 | 753.1 | 657.0 | 948.0 | 1000.0 | 1034.0 | 849.0 |
| BattleZone | 3000 | 37,187.5 | 22100.0 | 28400.0 | 37800.0 | 28400.0 | 27000.0 |
| BeamRider | 414.3 | 16,926.5 | 9519.0 | 13069.2 | 8043.8 | 14072.6 | 9865.6 |
| Berzerk | 165.6 | 2,630.4 | 746.0 | 824.0 | 928.0 | 959.0 | 1029.0 |
| Bowling | 23.48 | 160.7 | 29.6 | 30.3 | 35.5 | 60.0 | 12.6 |
| Boxing | -0.69 | 12.1 | 96.0 | 91.8 | 98.3 | 96.9 | 96.7 |
| Breakout | 1.5 | 30.5 | 313.4 | 373.0 | 361.4 | 405.9 | 402.5 |
| Centipede | 2064.77 | 12,017.0 | 4548.1 | 6090.9 | 5508.0 | 5152.0 | 4952.2 |
| ChopperCommand | 794 | 7,387.8 | 2780.0 | 4360.0 | 5490.0 | 6760.0 | 6520.0 |
| CrazyClimber | 8043 | 35,829.4 | 15960.0 | 158070.0 | 69430.0 | 112130.0 | 16000.0 |
| DemonAttack | 162.25 | 1,971.0 | 58324.5 | 41656.5 | 63889.0 | 437760.5 | 195827.0 |
| DoubleDunk | -18.14 | -16.4 | 0.2 | 0.6 | -0.4 | -0.4 | -2.2 |
| Enduro | 0.01 | 860.5 | 1961.3 | 1507.5 | 2832.5 | 3248.2 | 4272.0 |
| FishingDerby | -93.06 | -38.7 | 15.8 | 26.0 | 33.4 | 24.5 | 24.6 |
| Freeway | 0.01 | 29.6 | 30.9 | 32.6 | 34.0 | 33.6 | 34.0 |
| Frostbite | 73.2 | 4,334.7 | 1767.0 | 3317.0 | 4487.0 | 2874.0 | 2632.0 |
| Gopher | 364 | 2,412.5 | 7058.0 | 9314.0 | 6466.0 | 6412.0 | 15168.0 |
| Gravitar | 226.5 | 3,351.4 | 110.0 | 325.0 | 565.0 | 345.0 | 470.0 |
| Hero | 551 | 30,826.4 | 4657.5 | 8098.0 | 11673.5 | 7215.0 | 7476.0 |
| IceHockey | -10.3 | 0.9 | -13.0 | -11.4 | -3.6 | -4.5 | -4.6 |
| Jamesbond | 27 | 302.8 | 320.0 | 625.0 | 1995.0 | 480.0 | 450.0 |
| Kangaroo | 54 | 3,035.0 | 660.0 | 9870.0 | 13440.0 | 14720.0 | 10680.0 |
| Krull | 1,566.59 | 2,665.5 | 9191.1 | 9366.9 | 9918.7 | 8732.7 | 9549.0 |
| KungFuMaster | 451 | 22,736.3 | 62800.0 | 55060.0 | 36020.0 | 36940.0 | 42600.0 |
| MontezumaRevenge | 0.0 | 4,753.3 | 1.0 | 1.0 | 1.0 | 1.0 | 0.0 |
| MsPacman | 242.6 | 6,951.6 | 3230.0 | 2168.0 | 2673.0 | 2568.0 | 2568.0 |
| NameThisGame | 2404.9 | 8,049.0 | 4702.0 | 6278.0 | 11739.0 | 12394.0 | 9200.0 |
| Phoenix | 757.2 | 7,242.6 | 5398.0 | 12043.0 | 12324.0 | 32086.0 | 18558.0 |
| Pitfall | -265 | 6,463.7 | 1.0 | 1.0 | 1.0 | 1.0 | 0.0 |
| Pong | -20.34 | 14.6 | 20.0 | 20.7 | 20.8 | 20.9 | 21.0 |
| PrivateEye | 34.49 | 69,571.3 | 100.0 | 100.0 | 100.0 | 100.0 | 100.0 |
| Qbert | 188.75 | 13,455.0 | 8150.0 | 16575.0 | 13830.0 | 15782.5 | 6530.0 |
| RiverRaid | 1575.4 | 17,118.0 | 8350.0 | 10232.0 | 8714.0 | 9350.0 | 11998.0 |
| RoadRunner | 7 | 7,845.0 | 44950.0 | 54490.0 | 54620.0 | 42530.0 | 52600.0 |
| Robotank | 2.24 | 11.9 | 13.2 | 22.5 | 48.1 | 34.4 | 48.1 |
| Seaquest | 88.2 | 42,054.7 | 1444.0 | 10666.0 | 2640.0 | 11685.0 | 14795.0 |
| Skiing | -16267.9 | -4,336.9 | -13340.4 | -19040.3 | -29970.3 | -8983.3 | -29970.3 |
| Solaris | 2346.6 | 12,326.7 | 582.0 | 192.0 | 956.0 | 3336.0 | 792.0 |
| SpaceInvaders | 136.15 | 1,668.7 | 1005.0 | 1725.5 | 1826.5 | 1216.0 | 2302.5 |
| StarGunner | 631 | 10,250.0 | 1270.0 | 22600.0 | 38380.0 | 52050.0 | 43820.0 |
| Tennis | -23.92 | -8.3 | -5.7 | -1.5 | -11.9 | -1.5 | 13.3 |
| TimePilot | 3682 | 5,229.2 | 1420.0 | 3260.0 | 6030.0 | 7900.0 | 7060.0 |
| Tutankham | 15.56 | 167.6 | 206.6 | 186.0 | 178.3 | 205.2 | 202.8 |
| UpNDown | 604.7 | 11,693.2 | 19145.0 | 16046.0 | 17074.0 | 44746.0 | 20063.0 |
| Venture | 0.0 | 1,187.5 | 1.0 | 1.0 | 1.0 | 1.0 | 1370.0 |
| VideoPinball | 15720.98 | 17,667.9 | 270050.9 | 477206.8 | 388106.7 | 288137.2 | 164597.3 |
| WizardOfWor | 534 | 4,756.5 | 1440.0 | 1620.0 | 4890.0 | 4480.0 | 3250.0 |
| YarsRevenge | 3271.42 | 54,576.9 | 12507.9 | 15954.4 | 17593.8 | 8516.8 | 13507.3 |
| Zaxxon | 8 | 9,173.3 | 1.0 | 5910.0 | 7410.0 | 4640.0 | 10320.0 |

Table 3: Scores of all algorithms averaged over 3 seeds across 55 Atari games after training 40M Frames. All scores are computed based on our own PyTorch implementation, rather than directly referring to existing ones based on the Dopamine TensorFlow framework with 200M frames.

## G FEATURES OF ATARI GAMES

| GAMES | Action Space | Dynamics |
|---|---|---|
| Alien | 18 | Complex |
| Amidar | 6 | Simple |
| Assault | 7 | Complex |
| Asterix | 18 | Complex |
| Asteroids | 4 | Simple |
| Atlantis | 4 | Simple |
| BankHeist | 18 | Simple |
| BattleZone | 18 | Simple |
| BeamRider | 18 | Complex |
| Berzerk | 18 | Complex |
| Bowling | Continuous | Simple |
| Boxing | 6 | Simple |
| Breakout | 4 | Simple |
| Centipede | 18 | Complex |
| ChopperCommand | Continuous | Complex |
| CrazyClimber | 18 | Complex |
| DemonAttack | 18 | Complex |
| DoubleDunk | 18 | Simple |
| Enduro | 9 | Simple |
| FishingDerby | 18 | Simple |
| Freeway | 3 | Simple |
| Frostbite | 18 | Complex |
| Gopher | 18 | Simple |
| Gravitar | Continuous | Complex |
| Hero | 18 | Simple |
| IceHockey | Continuous | Simple |
| Jamesbond | 18 | Complex |
| Kangaroo | 18 | Complex |
| Krull | 18 | Complex |
| KungFuMaster | 18 | Complex |
| MontezumaRevenge | 18 | Complex |
| MsPacman | 9 | Simple |
| NameThisGame | 18 | Complex |
| Phoenix | 18 | Complex |
| Pitfall | 18 | Complex |
| Pong | 3 | Simple |
| PrivateEye | 18 | Complex |
| Qbert | 6 | Complex |
| Riverraid | 18 | Complex |
| RoadRunner | 18 | Simple |
| Robotank | 9 | Simple |
| Seaquest | 18 | Complex |
| Skiing | 9 | Simple |
| Solaris | 18 | Complex |
| SpaceInvaders | 6 | Simple |
| StarGunner | 18 | Complex |
| Tennis | 18 | Simple |
| TimePilot | 18 | Complex |
| Tutankham | 18 | Complex |
| UpNDown | 18 | Complex |
| Venture | 18 | Complex |
| VideoPinball | 6 | Simple |
| WizardOfWor | 12 | Complex |
| YarsRevenge | 18 | Complex |
| Zaxxon | 18 | Complex |

Table 4: Number of Action space and difficulty of environmental dynamics of 55 Atari games.

# H SENSITIVITY ANALYSIS AND COMPUTATIONAL COST

## H.1 MORE RESULTS IN SENSITIVITY ANALYSIS

**Decreasing $\varepsilon$.** We argue that the limit behavior connection as stated in Theorem 1 may not be able to be verified rigorously via numeral experiments due to the numerical instability of Sinkhorn Iteration in Algorithm 2. From Figure 6 (a), we can observe that if we gradually decline $\varepsilon$ to 0, SinkhornDRL's performance tends to degrade and approach QR-DQN. Note that an overly small $\varepsilon$ will lead to a trivial almost 0 $\mathcal{K}_{i,j}$ in Sinkhorn iteration in Algorithm 2, and will cause $\frac{1}{0}$ numerical instability issue for $a_l$ and $b_l$ in Line 5 of Algorithm 2. In addition, we also conducted experiments on Seaquest, a similar result is also observed in Figure 6 (d). As shown in Figure 6 (d), the performance of SinkhornDRL is robust when $\varepsilon = 10, 100, 500$, but a small $\epsilon = 1$ tends to worsen the performance.

**Increasing $\varepsilon$.** Moreover, for breakout, if we increase $\varepsilon$, the performance of SinkhornDRL tends to degrade and be close to MMDDRL as suggested in Figure 6 (b). It is also noted that an overly large $\varepsilon$ will let the $\mathcal{K}_{i,j}$ explode to $\infty$. This also leads to the numerical instability issue in Sinkhorn iteration in Algorithm 2.

**Samples $N$.** We find that SinkhornDRL requires a proper number of samples $N$ to perform favorably, and the sensitivity w.r.t $N$ depends on the environment. As suggested in Figure 7 (a), a smaller $N$, e.g., $N = 2$ on breakout has already achieved favorable performance and even accelerates the convergence in the early phase, while $N = 2$ on Seaquest will lead to the divergence issue. Meanwhile, an overly large $N$ worsens the performance across two games. We conjecture that using

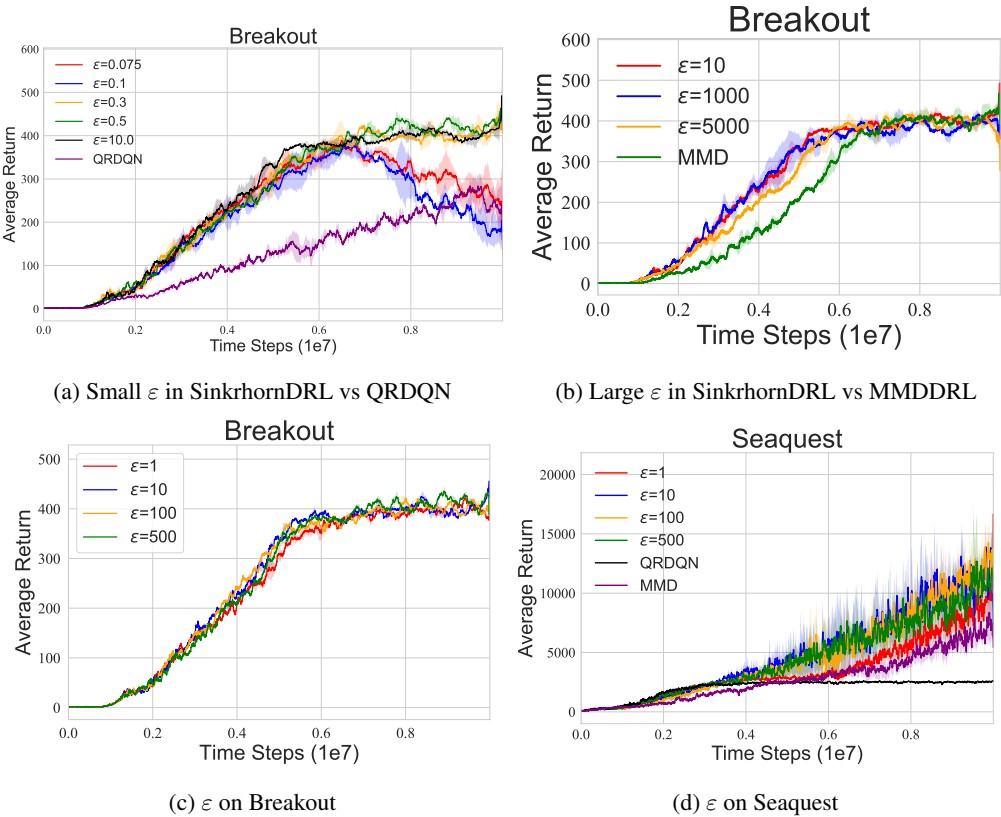

(a) Small $\varepsilon$ in SinkrhornDRL vs QRDQN

(b) Large $\varepsilon$ in SinkrhornDRL vs MMDDRL

(c) $\varepsilon$ on Breakout

(d) $\varepsilon$ on Seaquest

Figure 6: (a) Sensitivity analysis w.r.t. a small level of $\varepsilon$ SinkhornDRL to compare with QR-DQN that approximates Wasserstein distance on Breakout. (b) Sensitivity analysis w.r.t. a large level of $\varepsilon$ SinkhornDRL algorithm to compare with MMDDRL on Breakout. All learning curves are reported over 2 seeds. (c) and (d) are results for a general $\varepsilon$ on Breakout and Seaquest, respectively.

larger network networks to generate more samples may suffer from the overfitting issue, yielding the training instability (Bjorck et al., 2021). In practice, we choose a proper number of samples, i.e., $N = 200$ across all games.

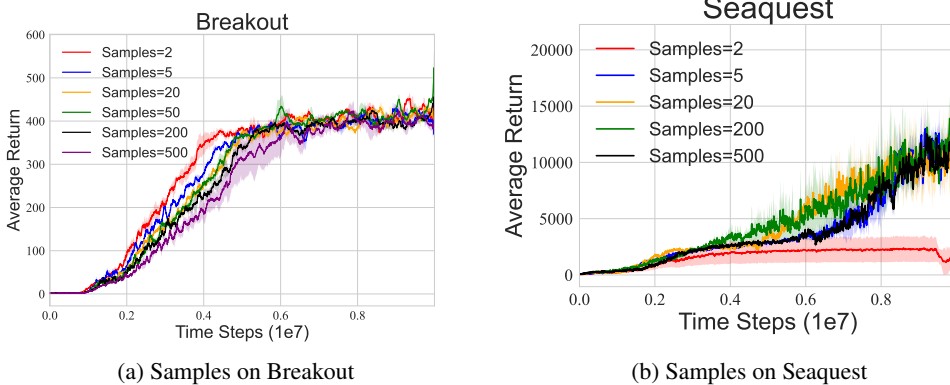

(a) Samples on Breakout        (b) Samples on Seaquest

Figure 7: Sensitivity analysis of Sinkhorn in terms of the number of samples $N$ on Breakout (a) and Seaquest (b).

**More Games on StarGunner and Zaxxon.** Beyond Breakout and Seaquest, we also provide sensitivity analysis on StarGunner and Zaxxon games in Figure 8. It suggests overly small samples, e.g., 1 and overall large samples tend to degrade the performance, especially on Zaxxon. Although the two games are robust to $\varepsilon$, and we find a small or large $\varepsilon$ hurts the performance in Seaquest. Thus, considering all games, we set samples 200, and $\varepsilon = 10.0$ in a moderate range across all games, although a more careful tuning in each game will improve the performance further.

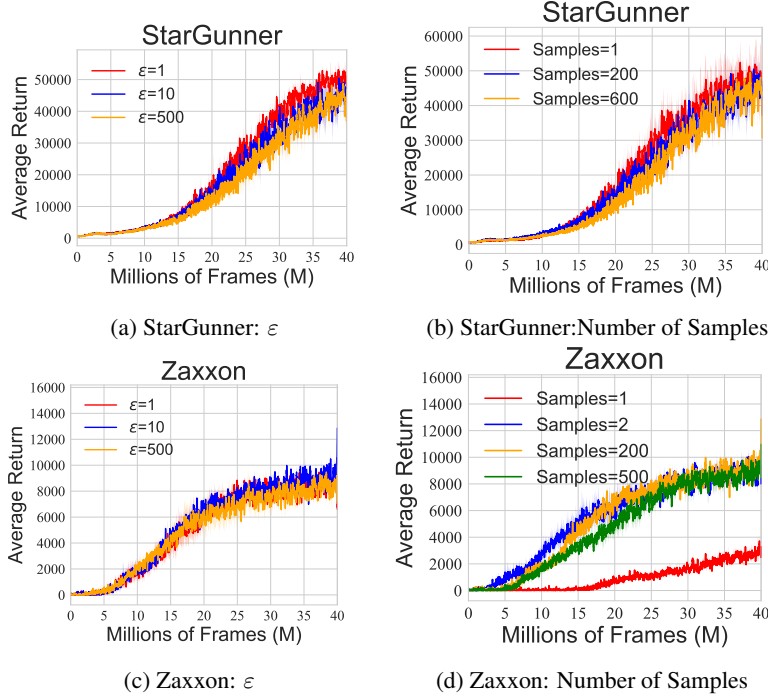

(a) StarGunner: $\varepsilon$        (b) StarGunner:Number of Samples

(c) Zaxxon: $\varepsilon$        (d) Zaxxon: Number of Samples

Figure 8: Sensitivity analysis of SinkhornDRL on StarGunner and Zaxxon in terms of $\varepsilon$, and number of samples. Learning curves are reported over 3 seeds.

## H.2 COMPARISON WITH THE COMPUTATIONAL COST

We evaluate the computational time every 10,000 iterations across the whole training process of all considered distributional RL algorithms and make a comparison in Figure 9. It suggests that SinkhornDRL indeed increases around 50% computation cost compared with QR-DQN and C51, but only slightly increases the cost in contrast to MMDDRL on both Breakout and Qbert games. We argue that this additional computational burden can be tolerant given the significant outperformance of SinkhornDRL in a large number of environments.

In addition, we also find that the number of Sinkhorn iterations $L$ is negligible to the computation cost, while an overly large sample $N$, e.g., 500, will lead to a large computational burden as illustrated in Figure 10. This can be intuitively explained as the computation complexity of the cost function $c_{i,j}$ is $\mathcal{O}(N^2)$ in SinkhornDRL, which is particularly heavy in the computation if $N$ is large enough.

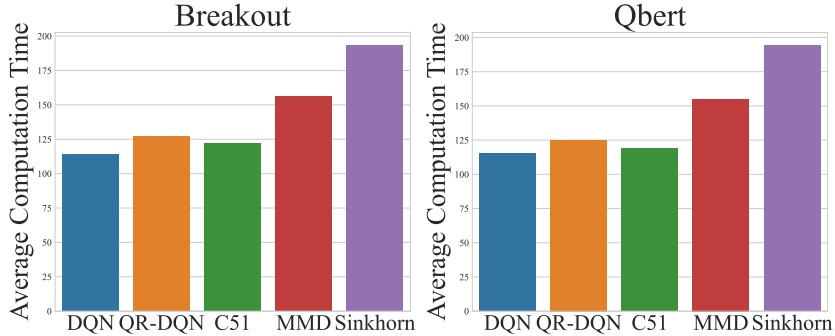

Figure 9: Average computational cost per 10,000 iterations of all considered distributional RL algorithm, where we select $\varepsilon = 10$, $L = 10$ and the number of samples $N = 200$ in SinkhornDRL algorithm.

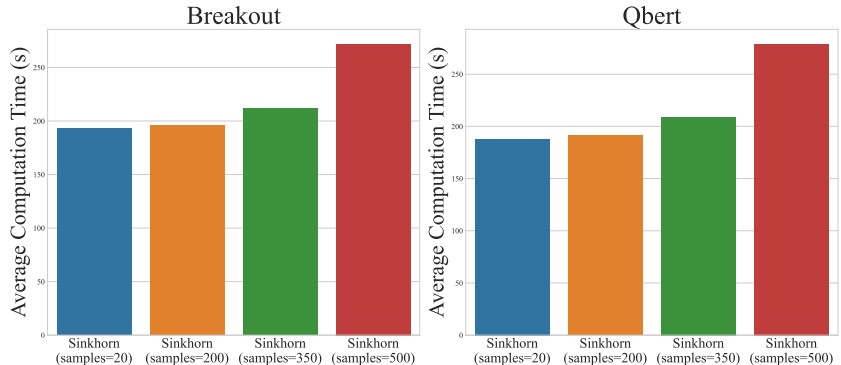

Figure 10: Average computational cost per 10,000 iterations of SinkhornDRL algorithm over different samples.

# I  ALGORITHM: SINKHORN DISTRIBUTIONAL RL

---

**Algorithm 3** Sinkhorn Distributional RL

---

**Require:** Number of generated samples $N$, the kernel $k$ (e.g., unrectified kernel), discount factor $\gamma \in [0, 1]$, learning rate $\alpha$, replay buffer $M$, main network $Z_\theta$, target network $Z_{\theta^*}$, number of iterations $L$, hyperparameter $\varepsilon$, and a behavior policy $\pi$ based on $Z_\theta$ following an $\epsilon$-greedy rule

1: Initialize $\theta$ and $\theta^* \leftarrow \theta$
2: **for** $t = 1, 2, \ldots$ **do**
3:     Take action $a_t \sim \pi(\cdot|s_t; \theta)$, receive reward $r_t \sim R(\cdot|s_t, a_t)$, and observe $s_{t+1} \sim P(\cdot|s_t, a_t)$
4:     Store $(s_t, a_t, r_t, s_{t+1})$ to the replay buffer $M$
5:     Randomly draw a batch of transition samples $(s, a, r, s')$ from the replay buffer $M$
6:     Compute a greedy action:

$$a^* = \arg\max_{a' \in A} \frac{1}{N} \sum_{i=1}^{N} Z_{\theta^*}(s', a')_i$$

7:     Compute the target Bellman return distribution:

$$\mathfrak{T}Z_i \leftarrow r + \gamma Z_{\theta^*}(s', a^*)_i, \forall 1 \leq i \leq N$$

8:     Evaluate Sinkhorn divergence via Sinkhorn Iterations in Algorithm 2:

$$\overline{\mathcal{W}}_{c,\varepsilon} \left( \{Z_\theta(s, a)_i\}_{i=1}^{N}, \{\mathfrak{T}Z_j\}_{j=1}^{N} \right)$$

9:     Update the main network $Z_\theta$:

$$\theta \leftarrow \theta - \alpha \nabla_\theta \overline{\mathcal{W}}_{c,\varepsilon} \left( \{Z_\theta(s, a)_i\}_{i=1}^{N}, \{\mathfrak{T}Z_j\}_{j=1}^{N} \right)$$

10:     Periodically update the target network $\theta^* \leftarrow \theta$
11: **end for**

---

