# SINKHORN DISTRIBUTIONAL REINFORCEMENT LEARNING

## ABSTRACT

The empirical success of distributional reinforcement learning (RL) highly depends on the representation of return distributions and the choice of distribution divergence. In this paper, we propose *Sinkhorn distributional RL (SinkhornDRL)* algorithm that learns unrestricted statistics, i.e., deterministic samples, from each return distribution and then leverages Sinkhorn divergence to minimize the difference between current and target Bellman return distributions. Theoretically, we prove the convergence properties of SinkhornDRL in the tabular setting, which is consistent with the interpolation nature of Sinkhorn divergence between Wasserstein distance and Maximum Mean Discrepancy (MMD). We also establish a new equivalent form of Sinkhorn divergence with a regularized MMD beyond the optimal transport literature, contributing to interpreting the superiority of SinkhornDRL over existing distributional RL methods. Empirically, we show that SinkhornDRL is consistently better or comparable to existing algorithms on the suite of 55 Atari games.

## 1 INTRODUCTION

The design of classical reinforcement learning (RL) algorithms is mainly based on the expectation of cumulative rewards that an agent observes while interacting with the environment. Recently, a new class of RL algorithms called *distributional RL* estimates the full distribution of total returns and has exhibited state-of-the-art performance in a wide range of environments, such as C51 (Bellemare et al., 2017a), Quantile-Regression DQN (QR-DQN) (Dabney et al., 2018b), Implicit Quantile Networks (IQN) (Dabney et al., 2018a), Fully Parameterized Quantile Function (FQF) (Yang et al., 2019), Non-Crossing QR-DQN (Zhou et al., 2020), MMDDRL (Nguyen et al., 2020), Spline DQN (SPL-DQN) (Luo et al., 2021). Meanwhile, distributional RL has also enjoyed other benefits in risk-sensitive control (Ma et al., 2020; Dabney et al., 2018a), policy exploration settings (Mavrin et al., 2019; Rowland et al., 2019), robustness (Sun et al., 2023) and optimization (Sun et al., 2022). In this work, we motivate a new distributional RL family via Sinkhorn divergence (Sinkhorn, 1967), called *SinkhornDRL*, by revealing its advantages over existing distributional RL algorithms.

**Advantages over Quantile-based / Wasserstein Distance Distributional RL.** 1) Avoid the non-crossing issue. Quantile-based algorithms suffer from the non-crossing issue (Zhou et al., 2020), while using Sinkhorn divergence can elegantly sidestep it. 2) More flexible statistics. SinkhornDRL employs samples to depict the return distribution, offering greater flexibility than quantiles. 3) Stability. Owing to its inherent smoothness, Sinkhorn divergence stands out as more numerically stable than several methods used to calculate the Wasserstein distance. 4) Adaptability. SinkhornDRL can handle the multi-dimensional reward function setting (Zhang et al., 2021), while the quantile regression suffers from the curse of dimension. **Advantages over MMDDRL.** 1) Richer geometry. Sinkhorn divergence is based on optimal transport and thus is capable of capturing richer geometric differences between distributions. In contrast, MMD relies on Reproducing Kernel Hilbert space (RKHS) and may fail to capture the data geometry. 2) Interpolation Flexibility. Sinkhorn divergence can find a sweet spot between Wasserstein distance and MMD, and this flexibility allows a more tailored divergence measure on the specific requirements of the task at hand.

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

, 1)$ is also determined by the specified two probability measures $\mu, \nu$. Next, we additionally show that $\overline{\Delta}(\gamma, \alpha) = \sup_{\mu,\nu} \overline{\Delta}^{\mu,\nu}(\gamma, \alpha) < 1$ holds strictly owing to the difference between a non-trivial Sinkhorn divergence and Wasserstein distance $W_\alpha^\alpha$, i.e., the non-zero entropic regularization in $\mathcal{W}_{c,\varepsilon}$. Based on the contraction mapping theorem, we eventually arrive at the $\overline{\Delta}(\gamma, \alpha)$-contraction of distributional Bellman operator $\mathfrak{T}^\pi$ under $\overline{\mathcal{W}}_{c,\varepsilon}^\infty$. Our non-trivial proof about Sinkhorn divergence can even potentially contribute to the optimal transport literature.

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

 = 0$ **and** $c = -k_\alpha$ It is obvious to observe that Sinkhorn loss degenerates to the Wasserstein distance. We also have the conclusion that the distributional Bellman operator $\mathfrak{T}^\pi$ is $\gamma$-contractive under the supreme form of Wasserstein distance, the proof of which is provided in

Lemma 3 (Bellemare et al., 2017a). Since the above conclusion is made directly based on the limiting case when $\varepsilon = 0$, for an unspecified $\varepsilon > 0$ albeit $\varepsilon \to 0$, we need a more rigorous proof. We show that their distance difference is **at most an infinitesimal** $\delta$.

Firstly, as $\mathcal{W}_{c,\varepsilon} \to W_\alpha$ and the regularization term is non-negative, using the language of $(\epsilon, \delta)$ definition, we have: for $\forall \delta$, there exists a small positive constant $a$, such that $\mathcal{W}_{c,\varepsilon} - W_\alpha < \delta$ when $\varepsilon \le a$. Based on that, we have the contraction conclusion:

$$\overline{\mathcal{W}}_{-\kappa_\alpha,\varepsilon}^\infty(\mathfrak{T}^\pi Z_1, \mathfrak{T}^\pi Z_2) = \overline{\mathcal{W}}_{-\kappa_\alpha,\varepsilon}^\infty(\mathfrak{T}^\pi Z_1, \mathfrak{T}^\pi Z_2) - W_\alpha^\infty(\mathfrak{T}^\pi Z_1, \mathfrak{T}^\pi Z_2) + W_\alpha^\infty(\mathfrak{T}^\pi Z_1, \mathfrak{T}^\pi Z_2)$$
$$\le \delta + W_\alpha^\infty(\mathfrak{T}^\pi Z_1, \mathfrak{T}^\pi Z_2),$$
(14)

where the second term $W_\alpha^\infty(\mathfrak{T}^\pi Z_1, \mathfrak{T}^\pi Z_2)$ is contractive. Therefore, for the unspecified $\varepsilon$, the only difference from the limiting $\varepsilon = 0$ is an infinitesimal $\delta$, which will vanish as $\varepsilon \to 0$ or $a \to 0$.

**2. $\varepsilon = \infty$ and $c = -k_\alpha$.** Our complete proof is inspired by (Ramdas et al., 2017; Genevay et al., 2018). Recap the definition of squared MMD is

$$\mathbb{E}\left[k\left(\mathbf{X}, \mathbf{X}'\right)\right] + \mathbb{E}\left[k\left(\mathbf{Y}, \mathbf{Y}'\right)\right] - 2\mathbb{E}[k(\mathbf{X}, \mathbf{Y})]$$

When the kernel function $k$ degenerates to an unrectified $k_\alpha(x, y) := -\|x - y\|^\alpha$ for $\alpha \in (0, 2)$, the squared MMD would degenerate to

$$2\mathbb{E}\|\mathbf{X} - \mathbf{Y}\|^\alpha - \mathbb{E}\|\mathbf{X} - \mathbf{X}'\|^\alpha - \mathbb{E}\|\mathbf{Y} - \mathbf{Y}'\|^\alpha$$

where $X, X' \overset{\text{i.i.d.}}{\sim} \mu, Y, Y' \overset{\text{i.i.d.}}{\sim} \nu$ and $X, X', Y, Y'$ are mutually independent. On the other hand, by definition, we have the Sinkhorn loss as

$$\overline{\mathcal{W}}_{c,\infty}(\mu, \nu) = 2\mathcal{W}_{c,\infty}(\mu, \nu) - \mathcal{W}_{c,\infty}(\mu, \mu) - \mathcal{W}_{c,\infty}(\nu, \nu)$$

Denoting $\Pi_\varepsilon$ be the unique minimizer for $\overline{\mathcal{W}}_{c,\varepsilon}$, it holds that $\Pi_\varepsilon \to \mu \otimes \nu$ as $\varepsilon \to \infty$. That being said, $\mathcal{W}_{c,\infty}(\mu, \nu) \to \int c(x, y)\mathrm{d}\mu(x)\mathrm{d}\nu(y) + 0 = \int c(x, y)\mathrm{d}\mu(x)\mathrm{d}\nu(y)$. If $c = -k_\alpha = \|x - y\|^\alpha$, we eventually have $\mathcal{W}_{-k_\alpha,\infty}(\mu, \nu) \to \int \|x - y\|^\alpha \mathrm{d}\mu(x)\mathrm{d}\nu(y) = \mathbb{E}\|\mathbf{X} - \mathbf{Y}\|^\alpha$. Finally, we can have

$$\overline{\mathcal{W}}_{-k_\alpha,\infty} \to 2\mathbb{E}\|\mathbf{X} - \mathbf{Y}\|^\alpha - \mathbb{E}\|\mathbf{X} - \mathbf{X}'\|^\alpha - \mathbb{E}\|\mathbf{Y} - \mathbf{Y}'\|^\alpha$$

which is exactly the form of squared MMD with the unrectified kernel $k_\alpha$. Now the key is to prove that $\Pi_\varepsilon \to \mu \otimes \nu$ as $\varepsilon \to \infty$. We give the detailed proof as follows.

Firstly, it is apparent that $\mathcal{W}_{c,\varepsilon}(\mu, \nu) \le \int c(x, y)\mathrm{d}\mu(x)\mathrm{d}\nu(y)$ as $\mu \otimes \nu \in \Pi(\mu, \nu)$. Let $\{\varepsilon_k\}$ be a positive sequence that diverges to $\infty$, and $\Pi_k$ be the corresponding sequence of unique minimizers for $\mathcal{W}_{c,\varepsilon}$. According to the optimality condition, it must be the case that $\int c(x, y)\mathrm{d}\Pi_k + \varepsilon_k \mathrm{KL}(\Pi_k, \mu \otimes \nu) \le \int c(x, y)\mathrm{d}\mu \otimes \nu + 0$ (when $\Pi(\mu, \nu) = \mu \otimes \nu$). Thus,

$$\mathrm{KL}\left(\Pi_k, \mu \otimes \nu\right) \leqslant \frac{1}{\varepsilon_k}\left(\int c\, \mathrm{d}\mu \otimes \nu - \int c\, \mathrm{d}\Pi_k\right) \to 0.$$

Besides, by the compactness of $\Pi(\mu, \nu)$, we can extract a converging subsequence $\Pi_{n_k} \to \Pi_\infty$. Since KL is weakly lower-semicontinuous, it holds that

$$\mathrm{KL}\left(\Pi_\infty, \mu \otimes \nu\right) \leqslant \lim_{k \to \infty} \inf \mathrm{KL}\left(\Pi_{n_k}, \mu \otimes \nu\right) = 0$$

Hence $\Pi_\infty = \mu \otimes \nu$. That being said that the optimal coupling is simply the product of the marginals, indicating that $\Pi_\varepsilon \to \mu \otimes \nu$ as $\varepsilon \to \infty$. As a special case, when $\alpha = 1$, $\overline{\mathcal{W}}_{-k_1,\infty}(u, v)$ is equivalent to the energy distance

$$d_E(\mathbf{X}, \mathbf{Y}) := 2\mathbb{E}\|\mathbf{X} - \mathbf{Y}\| - \mathbb{E}\|\mathbf{X} - \mathbf{X}'\| - \mathbb{E}\|\mathbf{Y} - \mathbf{Y}'\|.$$
(15)

In summary, if the cost function is the rectified kernel $k_\alpha$, it is the case that $\overline{\mathcal{W}}_{-k_\alpha,\varepsilon}$ converges to the squared MMD as $\varepsilon \to \infty$. According to (Nguyen et al., 2020), $\mathfrak{T}^\pi$ is $\gamma^{\alpha/2}$-contractive in the supreme form of MMD with the rectified kernel $k_\alpha$.

For the unspecified $\varepsilon < +\infty$ albeit $\varepsilon \to +\infty$, we can get a similar result to the case of $\varepsilon \to 0$. For $\forall \delta$, there exists a large positive constant $M$, such that $\mathrm{MMD}_{k_\alpha}^2 - \mathcal{W}_{c,\varepsilon} < \delta$ when $\varepsilon \ge M$. Based on that, we have the contraction conclusion:

$$\overline{\mathcal{W}}_{-\kappa_\alpha,\varepsilon}^\infty(\mathfrak{T}^\pi Z_1, \mathfrak{T}^\pi Z_2) = \overline{\mathcal{W}}_{-\kappa_\alpha,\varepsilon}^\infty(\mathfrak{T}^\pi Z_1, \mathfrak{T}^\pi Z_2) - \mathrm{MMD}_\infty^2(\mathfrak{T}^\pi Z_1, \mathfrak{T}^\pi Z_2) + \mathrm{MMD}_\infty^2(\mathfrak{T}^\pi Z_1, \mathfrak{T}^\pi Z_2)$$
$$\le \mathrm{MMD}_\infty^2(\mathfrak{T}^\pi Z_1, \mathfrak{T}^\pi Z_2) - \delta,$$
(16)

where the first term $\text{MMD}_\infty^2(\mathfrak{T}^\pi Z_1, \mathfrak{T}^\pi Z_2)$ is $\gamma^{\frac{\alpha}{2}}$-contractive. Hence, for the unspecified $\varepsilon$, the only difference from the limiting $\varepsilon = \infty$ is an infinitesimal $\delta$, which will vanish as $\varepsilon \to +\infty$ or $M \to +\infty$.

**3. For $\varepsilon \in (0, +\infty)$, the contraction property needs a long proof.** The proof pipeline is firstly we prove three properties of Sinkhorn divergence, and then we show the contraction of the distributional Bellman operator under Sinkhorn divergence based on its properties. Most importantly, we analyzed the contraction of the distributional Bellman operator under a new non-constant factor, whose supremum is strictly less than 1.

**3.1 Properties of Sinkhorn Divergence.** We recap three crucial properties of a divergence metric. The first is *scale sensitive* **(S)** (of order $\beta$, $\beta > 0$), i.e., $d_p(cX, cY) \leq |c|^\beta d_p(X, Y)$. The second property is *shift invariant* **(I)**, i.e., $d_p(A + X, A + Y) \leq d_p(X, Y)$. The last one is *unbiased gradient* **(U)**. A key observation is Sinkhorn divergence would degenerate to a two-dimensional KL divergence, and therefore embraces similar properties to KL divergence. Concretely, according to the equivalent form of $\mathcal{W}_{c,\varepsilon}(\mu, \nu)$ in Eq. 3, it can be expressed as the KL divergence between an optimal joint distribution and a Gibbs distribution associated with the cost function:

$$\mathcal{W}_{c,\varepsilon}(\mu, \nu) := \text{KL}\left(\Pi^*(\mu, \nu) | \mathcal{K}(\mu, \nu)\right), \tag{17}$$

where $\Pi^*$ is the optimal joint distribution. Thus, the total Sinkhorn divergence is expressed as

$$\overline{\mathcal{W}}_{c,\varepsilon}(\mu, \nu) := 2\text{KL}\left(\Pi^*(\mu, \nu) | \mathcal{K}(\mu, \nu)\right) - \text{KL}\left(\Pi^*(\mu, \mu) | \mathcal{K}(\mu, \mu)\right) - \text{KL}\left(\Pi^*(\nu, \nu) | \mathcal{K}(\nu, \nu)\right). \tag{18}$$

Due to the form of $\overline{\mathcal{W}}_{c,\varepsilon}(\mu, \nu)$, the convergence behavior is determined by $\mathcal{W}_{c,\varepsilon}(\mu, \nu)$, which is similar to the behavior of KL divergence. According to the fact that KL divergence has unbiased gradient estimates **(U)** and shift invariant **(I)**, and Sinkkhorn divergence can be viewed as a two-dimensional KL divergence, both properties of **U** and *I* can be extended to Sinkhorn divergence. **However, we find the non-scale sensitive (S) property of KL divergence can not directly apply to Sinkhorn divergence** due to the minimum nature of $\mathcal{W}_{c,\varepsilon}(\mu, \nu)$ and the difference between optimal joint distributions of $\Pi^*(\mu, \nu)$ and $\Pi^0(a\mu, a\nu)$ where $a$ is the scale factor. On the contrary, we find Sinkhorn divergence satisfies a variant of scale-sensitive property under certain conditions, which is crucial for the convergence of the distributional Bellman operator under Sinkhorn divergence. As such, we provide a new rigorous proof of scale-sensitive property as follows.

**3.2 A New Variant of Scale Sensitive Property of Sinkhorn Divergence.** We show the key part of Sinkhorn divergence, i.e., $\mathcal{W}_{c,\varepsilon}$, satisfies a variant of scale sensitive property when $c = -k_\alpha$, i.e.,

$$\mathcal{W}_{c,\varepsilon}(aU, aV) \leq \Delta(a, \alpha)\mathcal{W}_{c,\varepsilon}(U, V), \tag{19}$$

where $\Delta(a, \alpha) = 1 - \inf_{U,V} \frac{(1 - |a|^\alpha) \int (x-y)^\alpha d\Pi^*(x,y)}{\mathcal{W}_{c,\varepsilon}(U,V)} \in (|a|^\alpha, 1)$. Before a formal proof, we introduce a Lemma.

**Lemma 1.** *Define $c(x) = a(x) + b(x)$, where $a(x) \geq 0, b(x) \geq 0$ for each $x \in \mathcal{D}$. Both $a(x)$ and $b(x)$ are bounded if $c(x)$ is bounded for each $x$.*

*Proof.* Denote $c(x) \leq M$. If $a(x_0)$ is divergent given any $x_0$, then $b(x_0) = c(x_0) - a(x_0) \leq M - +\infty < 0$, which contradicts with the positive $b(x)$. Thus, $a(x)$ is bounded for each $x \in \mathcal{D}$. A similar proof is also applied for $b(x)$. $\square$

By definition of Sinkhorn divergence, the pdf of $\mathcal{K}(U, V) \propto e^{\frac{-c(x,y)}{\varepsilon}} \mu(x)\nu(y)$. After a scaling transformation, the pdf of $aU$ and $aV$ with respect to $x$ and $y$ would be $\frac{1}{a}\mu(\frac{x}{a})$ and $\frac{1}{a}\nu(\frac{y}{a})$. Thus $\mathcal{K}(aU, aV) \propto e^{\frac{-c(x,y)}{\varepsilon}} \frac{1}{a}\mu(\frac{x}{a})\frac{1}{a}\nu(\frac{y}{a})$. We denote $\Pi^*$ and $\Pi^0$ as the optimal joint distribution of

$\mathcal{W}_{c,\varepsilon}(\mu, \nu)$ and $\mathcal{W}_{c,\varepsilon}(a\mu, a\nu)$. Then we have:

$$
\begin{aligned}
\mathcal{W}_{c,\varepsilon}(aU, aV) &= \int c(x,y)\mathrm{d}\Pi^0(x,y) + \varepsilon \mathrm{KL}(\Pi^0 | a\mu \otimes a\nu) \\
&\leq \int c(x,y)\mathrm{d}\Pi^*(x,y) + \varepsilon \mathrm{KL}(\Pi^* | a\mu \otimes a\nu) \\
&\stackrel{c=-k_\alpha}{=} \int (x-y)^\alpha \frac{1}{a^2} \pi^*(\frac{x}{a}, \frac{y}{a})\mathrm{d}x\mathrm{d}y + \varepsilon \int \frac{1}{a^2}\pi^*(\frac{x}{a}, \frac{y}{a}) \log \frac{\frac{1}{a^2}\pi^*(\frac{x}{a}, \frac{y}{a})}{\frac{1}{a^2}\mu(\frac{x}{a})\nu(\frac{y}{a})}\mathrm{d}x\mathrm{d}y \\
&= |a|^\alpha \int (x-y)^\alpha \pi^*(x,y)\mathrm{d}x\mathrm{d}y + \varepsilon \int \pi^*(x,y) \log \frac{\pi^*(x,y)}{\mu(x)\nu(y)}\mathrm{d}x\mathrm{d}y \\
&= \int (x-y)^\alpha \pi^*(x,y)\mathrm{d}x\mathrm{d}y + \varepsilon \mathrm{KL}(\Pi^* | \mu \otimes \nu) - (1-|a|^\alpha)\int (x-y)^\alpha \pi^*(x,y)\mathrm{d}x\mathrm{d}y \\
&= \mathcal{W}_{c,\varepsilon}(U, V) - (1-|a|^\alpha)\int (x-y)^\alpha \mathrm{d}\Pi^*(x,y) \\
&= \Delta^{U,V}(a,\alpha)\mathcal{W}_{c,\varepsilon}(U, V)
\end{aligned}
$$

(20)

where $\Delta^{U,V}(a,\alpha) = 1 - \frac{(1-|a|^\alpha)\int(x-y)^\alpha \mathrm{d}\Pi^*(x,y)}{\mathcal{W}_{c,\varepsilon}(U,V)} \in (|a|^\alpha, 1)$ for $\varepsilon \in (0, +\infty)$ and $a < 1$ due to the fact that $0 < (1-|a|^\alpha)\int(x-y)^\alpha \mathrm{d}\Pi^*(x,y) < \int(x-y)^\alpha \mathrm{d}\Pi^*(x,y) < \mathcal{W}_{c,\varepsilon}(U,V)$. $\Delta^{U,V}(a,\alpha)$ is a function less than 1 that depends on the two margin distributions and the scale factor $a$.

However, the fact that $\Delta^{U,V}(a,\alpha) < 1$ can only guarantee a non-expansive contraction rather than a desirable contraction of the distributional Bellman operator. For example, denote the non-constant factor as $q_k$ for the k-th distributional Bellman update, where $q_k < 1$. We can construct a counterexample as $q_k = 1 - 1/(k+2)^2$. In this case, $\Pi_{k=1}^{+\infty} q_k = \frac{2}{3}\frac{4}{3}\frac{3}{4}\frac{5}{4}\cdots > 0$, which intuitively implies that iteratively applying distribution Bellman operator may not lead to convergence given the non-constant factor $\Delta^{U,V}(a,\alpha)$. To address this issue towards a rigorous proof, we need to find **a universal upper bound of $\Delta^{U,V}(a,\alpha)$ for $\forall U, V$ that is strictly less than 1**. We have the following result:

$$
\begin{aligned}
\sup_{U,V} \Delta^{U,V}(a,\alpha) &= 1 - \inf_{U,V} \frac{(1-|a|^\alpha)\int(x-y)^\alpha \mathrm{d}\Pi^*(x,y)}{\mathcal{W}_{c,\varepsilon}(U,V)} \\
&= 1 - \inf_{U,V} \frac{(1-|a|^\alpha)\int(x-y)^\alpha \mathrm{d}\Pi^*(x,y)}{\int(x-y)^\alpha \mathrm{d}\Pi^*(x,y) + \varepsilon \mathrm{KL}(\Pi^* | \mu \otimes \nu)} \\
&\stackrel{(a)}{\leq} 1 - \inf_{U,V} \frac{(1-|a|^\alpha)\int(x-y)^\alpha \mathrm{d}\Pi^*(x,y)}{\int(x-y)^\alpha \mathrm{d}\Pi^*(x,y) + \varepsilon M} \\
&\stackrel{(b)}{<} 1 - \inf_{U,V} \frac{(1-|a|^\alpha)W_\alpha^\alpha}{W_\alpha^\alpha + \varepsilon M} \\
&\stackrel{(c)}{\leq} 1
\end{aligned}
$$

(21)

where according to Lemma 1, for a bounded $\mathcal{W}_{c,\varepsilon}(U,V)$ in general, we have a bounded $\mathrm{KL}(\Pi^* | \mu \otimes \nu)$ denoted as $\mathrm{KL}(\Pi^* | \mu \otimes \nu) < M$. The inequality $(a)$ results from the fact that the whole quantity is a monotonically increasing function regarding the KL term. The key inequality $(b)$ results from the infimum nature of these distances and the relationship between Sinkhorn divergence and Wasserstein distance $W_\alpha$, i.e., $\int(x-y)^\alpha \mathrm{d}\Pi^*(x,y) > \inf_\Pi \int(x-y)^\alpha \mathrm{d}\Pi(x,y) = W_\alpha^\alpha(U,V)$, where there is a strict inequality as long as $\mathrm{KL}(\Pi^* | \mu \otimes \nu) > 0$ in general for a non-trivial Sinkhorn divergence when $\varepsilon \in (0, +\infty)$. More importantly, their difference $\inf_{U,V} \int(x-y)^\alpha \mathrm{d}\Pi^*(x,y) - W_\alpha^\alpha(U,V) > 0$ holds even while taking the infimum, which is a strict inequality as well **for a non-trivial Sinkhorn divergence with $\mathrm{KL}(\Pi^* | \mu \otimes \nu) > 0$**. The inequality $(c)$ results from the $\inf_{U,V} \frac{(1-|a|^\alpha)W_\alpha^\alpha}{W_\alpha^\alpha + M} = 0$ when $W_\alpha^\alpha = 0$ with $U = V$. Our result indicates that $\

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

derive $\sup_{U,V} \overline{\Delta}^{U,V}(a,\alpha) < 1$. In particular, we have

$$
\begin{aligned}
\sup_{U,V} \overline{\Delta}^{U,V}(a,\alpha) &= 1 - \inf_{U,V} \frac{(1-|a|^\alpha)(2\int(x-y)^\alpha d\Pi^*(x,y) - \mathcal{W}_{c,\varepsilon}(\mu,\mu) - \mathcal{W}_{c,\varepsilon}(\nu,\nu))}{\overline{\mathcal{W}}_{c,\varepsilon}(U,V)} \\
&\leq 1 - \inf_{U,V} \frac{(1-|a|^\alpha)(2\int(x-y)^\alpha d\Pi^*(x,y) - \mathcal{W}_{c,\varepsilon}(\mu,\mu) - \mathcal{W}_{c,\varepsilon}(\nu,\nu))}{2\int(x-y)^\alpha d\Pi^*(x,y) - \mathcal{W}_{c,\varepsilon}(\mu,\mu) - \mathcal{W}_{c,\varepsilon}(\nu,\nu) + \varepsilon M} \\
&\overset{(d)}{<} 1 - \inf_{U,V} \frac{(1-|a|^\alpha)(2W_\alpha^\alpha(\mu,\nu) - \mathcal{W}_{c,\varepsilon}(\mu,\mu) - \mathcal{W}_{c,\varepsilon}(\nu,\nu))}{2W_\alpha^\alpha(\mu,\nu) - \mathcal{W}_{c,\varepsilon}(\mu,\mu) - \mathcal{W}_{c,\varepsilon}(\nu,\nu) + \varepsilon M} \\
&\overset{(e)}{\leq} 1
\end{aligned}
\tag{27}
$$

where the inequality $(d)$ is based on Lemma 2 and more importantly, $\inf_{U,V} \int(x-y)^\alpha d\Pi^*(x,y) - W_\alpha^\alpha(U,V) > 0$ holds even when we take the infimum, which is a strict inequality as well owing to a non-trivial Sinkhorn divergence with $\mathrm{KL}(\Pi^*|\mu \otimes \nu) > 0$. The inequality $(e)$ holds as $2W_\alpha^\alpha(\mu,\nu) - \mathcal{W}_{c,\varepsilon}(\mu,\mu) - \mathcal{W}_{c,\varepsilon}(\nu,\nu) \geq 0$ based on Lemma 2, where the inequality holds when the minimizers $\Phi^*, \phi^*$ in the dual norm of $\mathcal{W}_{c,\varepsilon}(\mu,\mu)$ are not exactly those for $\