# OpenReview forum: "Sinkhorn Distributional Reinforcement Learning"
_ICLR.cc/2024/Conference — Submitted to ICLR 2024_

### Official Review · Reviewer_v5sT · 2023-10-29

**Soundness:** 3 good
**Presentation:** 4 excellent
**Contribution:** 2 fair
**Rating:** 3
**Confidence:** 4

**Summary:**

The paper introduces the Sinkhorn distributional RL algorithm, which uses sinkhorn divergence to minimize differences between the current and target return distributions. Theoretical proofs show some properties of sinkhorn divergence and confirm its convergence properties. Empirical tests on Atari games show that the algorithm outperforms or matches existing distributional RL methods.

**Strengths:**

The paper is well-written and easy to follow, and the use of sinkhorn divergence is novel and interesting. It is complete and has both the theoretical part and the experimental part.

**Weaknesses:**

- I am not quite convinced by the experimental results. Specifically, Figure 5 and the subsequent remark indicate that "sinkhornDRL achieves better performance across almost half of the considered game". However, it might be possible that both QRDQN and MMD each excel in roughly 50% of the games over the other. Consequently, sinkhorn should, on average, surpass either of the two in 50% of the games, as Sinkhorn essentially interpolates between these two algorithms. Observing Table 3 seems to draw the same conclusion: when Sinkhorn outperforms QRDQN, it often falls to MMD; conversely when it outperforms MMD, it tends to underperform relative to QRDQN. However, I admit that this is not absolute, as there are indeed some games where Sinkhorn surpasses both. But my concern is whether "better performance across half of the games" is enough.

- The kernel assumed in the theoretical part (e.g., theorem 1) is different from that used in the experiments (the Gaussian kernel). So I am not sure how are the theory and the experiments connected.

- There seem to be some related works that are missed. I'm not sure if all of them are relevant, but the author can check if they are related. Some remarks are attached to each of them.

  - Li, Luchen, and A. Aldo Faisal. "Bayesian distributional policy gradients." : this paper proposes the policy gradient for distributional RL, and they use Wasserstein distance as well.

  - Wu, Runzhe, Masatoshi Uehara, and Wen Sun. "Distributional Offline Policy Evaluation with Predictive Error Guarantees." : this paper considers the total variation distance, which seems to be stronger than both Wasserstein distance and MMD. Hence I am wondering if it is also stronger than the sinkhorn divergence.

  - Ma, Yecheng, Dinesh Jayaraman, and Osbert Bastani. "Conservative offline distributional reinforcement learning." : this paper learns conservative return distributions, which seems necessary in the offline setting. Their theoretical guarantees are also under the Wasserstein distance.

  - Rowland, Mark, et al. "An analysis of quantile temporal-difference learning." : this paper studies the convergence of the quantile TD algorithm, and thus you may want to compare your analysis with theirs. They also established the fixed point error guarantee, and I am not sure if the same thing holds under sinkhorn divergence.

**Questions:**

- I really appreciate that the author established some nice properties of sinkhorn divergence (e.g., theorem 1, proposition 1). However, it is still unclear to me why the authors proposed to use sinkhorn divergence. They claimed that it is an interpolation between Wasserstein distance and MMD, and thus, I am wondering why it is expected to be better.  It would be great if the authors could provide a more either rigorous or intuitive explanation.

- The author proposed to generate finite samples to approximate the distribution (algorithm 1). Hence, I think a question is how large the statistical error will be, i.e., what is the error incurred when learning from finite samples, as compared to learning from the true distribution? Furthermore, how do these errors accumulate within an MDP? This may be a bit beyond the scope of this paper, but it will be interesting to see how statistically robust sinkhorn divergence is to finite samples intuitively.

---

> ### Author Response · Authors · 2023-11-18
> **(1/2) Author Response**
>
> Thank you for taking the time to review our paper. We appreciate your comments and feedback, and we would like to address the concerns you raised in your review.
>
> >Q1 in Weakness: I am not quite convinced by the experimental results...
>
> **A:** Let us summarize relevant results based on Table 3 to clarify your concerns. **Among 26 (22) games where SinkhornDRL outperforms QRDQN (MMDDRL), Sinkhron still performs better than MMDDRL (QRDQN) on 12 of them**.  Therefore, it is not sufficient to directly think that SinkhornDRL outperforms one and at the same time is inferior to the other. Generally speaking, although we can observe better performance on half of the games, the number of games on which a certain algorithm is superior to the others is only one of the important metrics to evaluate the capability of RL algorithms. We argue that the **Mean, Interquartile Mean(IQM), and median are shown in Table 2 and the learning curves in Appendix D are also very informative and more commonly used**. Table 2 showcases SinkrhornDRL is a generally better or competitive algorithm, and all learning curves in the Appendix substantiate that our algorithm is significantly better than the others in many games and competitive in the remaining ones.
>
> >Q2 in Weakness: The kernel assumed in the theoretical part (e.g., theorem 1) is different from that used in the experiments (the Gaussian kernel). So I am not sure how are the theory and the experiments connected.}
>
> **A:** We would like to clarify that the kernels **are the same used in the theoretical part and experiments, which are consistent**. Concretely, Theorem 1 demonstrates that the Bellman operator is contractive when we use the unrectified kernel in Sinkhorn divergence, while the Gaussian kernel cannot guarantee the contraction, which coincides with the MMDDRL paper. As mentioned in the **Hyperparameter Setting** in the Experiment, we still use the unrectified kernel in our SinkhornDRL algorithm, and therefore our theory and algorithm are consistent.
>
> >Q3 in Weakness: There seem to be some related works that are missed. I'm not sure if all of them are relevant, but the author can check if they are related. }
>
> **A:** Thanks for pointing out these related works and we have cited them appropriately in the revised paper. Here we give our discussion.
>
> * [1] is a Bayesian version of the distributional RL algorithm based on Wasserstein Variational Inference.
> * [2] TV can dominate (as an upper bound) many metrics, including Wasserstein distance, however distributional Bellman operator is not contractive over TV [10]. This is why the theoretical results in the infinite horizon setting in [2] used average Wasserstein distance rather than TV.
> * [3] is the first to study the advantages of distributional RL algorithms on the offline setting by imposing a conservative penalty term with the contraction guaranteed by Wasserstein distance.
> * [4] is an extension paper of QR-DQN to prove the convergence of quantile TD in the TD learning setting instead of the dynamic programming setting by leveraging the non-smoothness analysis and new stochastic approximation theory. Akin to QR-DQN, in the first step our paper focuses on the contraction property in the dynamic programming setting and similarly we can also leave an extension paper to prove the TD learning under Sinkrhon divergence as future work. One thing we want to add here is that the TD learning under MMD and Sinkhorn is based on samples, whose convergence is supposed to be more easily guaranteed via the standard stochastic approximation theory than the quantile TD. We leave more rigorous analysis as the extension work in the future.

---

> ### Author Response · Authors · 2023-11-18
> **(2/2) Author Response**
>
> >Q1 in Questions: However, it is still unclear to me why the authors proposed to use sinkhorn divergence. They claimed that it is an interpolation between Wasserstein distance and MMD, and thus, I am wondering why it is expected to be better.
>
> **A:** The interpolation nature of Sinkhorn divergence helps it to **enjoy the advantages of both Wasserstein distance and MMD**, and thus the corresponding distributional RL algorithms are more likely to perform better in many environments. As mentioned in the second paragraph in the Introduction, compared with Wasserstein distance,
> * Sinkhorn divergence makes the computation of Wasserstein distance **tractable** by adding entropic regularization to overcome the curse of dimensionality issue, especially in the multi-dimensional reward function setting of RL, instead of using the quantile regression to approximate. It thus naturally avoids the non-crossing issue in quantile-based algorithms.
> * Sinkhorn divergence uses the KL regularization to **encourage a "smoother" transport plan**, where the points are less tightly coupled to their specific targets in the optimal transport plan. This property is particularly beneficial in generating samples to approximate the current return distribution or in generative models, **ensuring that the generated samples cover the return distribution more uniformly**.
>
> We provide [5-9] for further reference.
>
> >Q2 in Questions: The author proposed to generate finite samples to approximate the distribution (algorithm 1). Hence, I think a question is how large the statistical error will be, i.e., what is the error incurred when learning from finite samples, as compared to learning from the true distribution?
>
> **A:** The statistical error is linked with **the sample complexity, i.e., approximating the distance with samples of measures**. We would like to clarify that we have provided the sample complexity results in Table 1 of our original paper, where the sample complexity of Sinkhorn divergence also basically interpolates between Wasserstein distance and MMD. As for how the error accumulated in an MDP, we believe it can be partially explained by the sensitivity analysis in terms of the number of samples. As suggested in the sensitivity analysis results in Figure 3 and Figures 7 and 8 of Appendix H on Breakout, Seaquest, Zaxxon and StarGunner games, a relatively small number of samples used in Sinkhorn can already achieve favorable performance. Based on these observations and the interpolated sample complexity of Sinkhorn divergence, we believe that Sinkhorn divergence is robust to the accumulated statistical errors.
>
> We thank the reviewer once again for the time and effort in reviewing our work! We would greatly appreciate it if the reviewer could check our responses and let us know whether they address the raised concerns. We are more than happy to provide further clarification if you have any additional concerns.
>
> [1] Li, Luchen, and A. Aldo Faisal. "Bayesian distributional policy gradients."
>
> [2] Wu, Runzhe, Masatoshi Uehara, and Wen Sun. "Distributional Offline Policy Evaluation with Predictive Error Guarantees."
>
> [3] Ma, Yecheng, Dinesh Jayaraman, and Osbert Bastani. "Conservative offline distributional reinforcement learning."
>
> [4] Rowland, Mark, et al. "An analysis of quantile temporal-difference learning."
>
> [5] Jean Feydy, Thibault S´ejourn´e, Franc¸ois-Xavier Vialard, Shun-ichi Amari, Alain Trouv´e, andGabriel Peyr´e. Interpolating between optimal transport and mmd using sinkhorn divergences.In The 22nd International Conference on Artificial Intelligence and Statistics, pp. 2681–2690.PMLR, 2019.
>
> [6] Aude Genevay, Gabriel Peyr´e, and Marco Cuturi. Learning generative models with sinkhorn divergences.In International Conference on Artificial Intelligence and Statistics, pp. 1608–1617.PMLR, 2018
>
> [7] Aaditya Ramdas, Nicol´as Garc´ıa Trillos, and Marco Cuturi. On Wasserstein two-sample testing and related families of nonparametric tests. Entropy, 19(2):47, 2017.
>
> [8] Aude Genevay, L´enaic Chizat, Francis Bach, Marco Cuturi, and Gabriel Peyr´e. Sample complexity of sinkhorn divergences. In The 22nd International Conference on Artificial Intelligence and Statistics, pp. 1574–1583. PMLR, 2019.
>
> [9] Marco Cuturi. Sinkhorn distances: Lightspeed computation of optimal transport. Advances in neural information processing systems, 26, 2013.
>
> [10] MG Bellemare et al, A Distributional Perspective on Reinforcement Learning. (ICML 2017)

---

> ### Comment · Reviewer_v5sT · 2023-11-21
>
> Thanks to the authors for the detailed response.
>
> It is good that SinkhornDRL can perform better than both QRDQN and MMDDRL in 12 games, as pointed out by the authors. However, I found that SinkhornDRL is worse than both algorithms in more than 20 games in Table 3. Hence, I feel Table 3 doesn't show any advantage of the proposed algorithm; it even shows that SinkhornDRL is slightly worse in the sense that it only wins QRDQN and MMDDRL in 12 games but loses in 20 games.
>
> The authors also argued that certain metrics of SinkhornDRL are better (eg. Table 2). However, if I understand it correctly, HNS could be biased toward the game that has a larger variance. In other words, the mean of HNS could be high if the proposed algorithm performs extremely well in some game that has a large variance but worse in many other games. I also found that in the fifth column (> human), the proposed algorithm is not the best, suggesting that the above-proposed issue may be true. Hence, I am not sure if Table 2 can outweigh the issue found in Table 3.
>
> I appreciate the authors' response and clarification to my other questions and they generally look good to me. However, my major concern is still the experimental results, as it seems not to show that the proposed algorithm has any obvious advantage over others. Hence I can't change my assessment for this work for now.

---

> > ### Author Response · Authors · 2023-11-22
> > **Thanks for the Reviewer's Comment and Author Further Response**
> >
> > Thanks for the reviewer's consistent dedication to reviewing our work. In our reponse, we would like to share some different opinions about the evaluation of empirical results for RL algorithms.
> >
> > > It is good that SinkhornDRL can perform better than both QRDQN and MMDDRL in 12 games, as pointed out by the authors. However, I found that SinkhornDRL is worse than both algorithms in more than 20 games in Table 3.....
> >
> > $A:$ Although the number of games that an algorithm outperforms provides some knowledge in evaluation, **this metric tends to be biased without considering the specific performance superiority or averaged returns that an algorithm can achieve over other baselines**. Another drawback is it does not include the difficulty of a certain game, which can be addressed by using human-normalized scores~(HNS). Note that Table 3 is the Raw score instead of the human normalized score. If an algorithm tends to slightly outperform others in most simple experiments, it tends to be superior in terms of the number of games. However, the weight of these games when we average them based on HNS would be not high as the human normalized scores on these games are still very high. On the contrary, if an algorithm behaves very well in some very complicated environments for humans, the resulting weights by considering HNS could be high, which is very sound as it sufficiently substantiates the potential capability of this algorithm, especially in complex environments. As shown in  Section 5.1 about the ratio improvement part, **these complicated environments are exactly what our SinkhornDRL is good at**. Thus, only relying on the number of games tends to overlook the potential capability of our algorithm in complicated environments, which is thus not sufficient.
> >
> > Instead, the most commonly used metrics are mean, IQM, and median, which carefully average the HNS **by trading off the statistical efficiency and the robustness to outliers**. In Table 2, our algorithm is generally better in terms of Mean, and IQM and very competitive to the best median. Also, the number of games over Human and DQN is complimentary results, which are typically less informative than Mean, IQM, and Median. More discussions about HMS are provided in the response to the next question.
> >
> >
> > > The authors also argued that certain metrics of SinkhornDRL are better (eg. Table 2). However, if I understand it correctly, HNS could be biased toward the game that has a larger variance.....
> >
> > $A:$ The point in dispute is why HNS is more acceptable than other metrics. The mean metric indeed suffers from the outlier issue, but it is more statistically efficient (summarizing more information over all samples) than others. By contrast, the median metric can avoid the impact of outliers as you mentioned, which is robust, but it loses too much statistical efficiency without making full use of detailed information in each sample. This is why [1] raised a very fundamental question towards a more scientific evaluation of the algorithm capability, in which they proposed to use **Interquartile Mean Metric(IQM)**. Following this strategy, we also report our IQM (5\%) results by evaluating the mean from $x\\%$ to $1-x\\%$ to **remove the impact of outliers, while maintaining a high statistical efficiency**. IQM results also substantiate that our SinkhornDLR algorithm is generally better in this more comprehensive, scientific, less biased metric.
> >
> > In addition, the performance we reported is based on **the same hyper-parameters**, including $L, \varepsilon$ and $N$ and thus there still exists a large improvement room for our algorithm to beat other baselines by carefully tuning them. More importantly, as mentioned in the discussion part and the first paragraph in Section 4, we focus on proposing a new distributional RL family by studying the simplest modeling choice via Sinkhorn divergence as rigorously as possible. Some **model extension techniques** can further enhance the performance, e.g., by parameterizing the cost function used in [2], which we leave such related modeling extensions as future works.
> >
> > We express our gratitude to the reviewer for their consistent dedication to reviewing our work. We would also like to highlight our **comprehensive contribution** of this paper, ranging from the theoretical contraction properties we have made substantial efforts on, the practical algorithm design, and the extensive experiments, which are also time-consuming to evaluate all baselines across all Atari games for us. We would be very grateful if you could reconsider the evaluation of our work for a revised score even in the final discussion phase among reviewers and AC. Of course, we always remain at your disposal for any further clarifications as best as we can.
> >
> > [1] Deep Reinforcement Learning at the Edge of the Statistical Precipice (NeurIPS 2021)
> >
> > [2] Learning generative models with sinkhorn divergences (AISTATS 2018)

---

### Official Review · Reviewer_PTyg · 2023-10-29

**Soundness:** 3 good
**Presentation:** 2 fair
**Contribution:** 2 fair
**Rating:** 6
**Confidence:** 3

**Summary:**

This paper focuses on solving online reinforcement learning using a distributional reinforcement learning formulation. It proposes a new algorithm called Sinkhorn distributional RL (SinkhornDRL) which: from the theory side, it enjoys the contraction property of the corresponding distributional Bellman operator; and empirically outperforms existing distributional RL baselines.

**Strengths:**

1. The algorithms show competitive results compared to existing distributional RL algorithms in the widely used benchmark 57 Atari games
2. It theoretically shows the contraction property of the distributional Bellman operator and the corresponding theoretical convergence of SinkhornDRL
3. The empirical analysis is sufficient regarding sensitivity analysis of the hyperparameters, computation cost, and etc.

**Weaknesses:**

1. The presentation needs to be polished since a lot of notations have not been introduced and may not be reading-friendly for people who are not familiar with distributional RL literature, as listed later.
2. The introduction of the algorithms seems not sufficient, for instance, what is the next step after getting the Sinkhorn distance in algorithm 1, such as gradient descent or other to minimize this distance?
3. Section 4.2 has a sequence of theoretical results, while more intuition will be helpful, such as the term $\overline{\Delta}(\gamma, \alpha)$ in Theorem 1(3) is represented in a very complicated way. So what do these terms mean and how large are they will be more helpful for the readers? So as the relationships to Gaussian or general kernels.

**Questions:**

1. As Sinkhorn is a well-known approach in optimal transport literature, it is curious what is the technical contribution of this paper, is it just an RL application inspired by Sinkhorn?

Other small issues:
1. Section 2.1, using $\overset{D}{:=}$ without defining the notation of D.
2. $Z_{\theta^\star}$ has not been introduced

---

> ### Author Response · Authors · 2023-11-18
> **(1/2) Author Response**
>
> Thank you for taking the time to review our paper. We appreciate your positive assessment and insightful feedback, and we would like to address the concerns you raised in your review.
>
> >Q1 in Weakness. The presentation needs to be polished since a lot of notations have not been introduced and may not be reading-friendly for people who are not familiar with distributional RL literature, as listed later.
>
> **A:** Thanks for this suggestion and we have fixed the presentation issues you pointed out in the revised version.
>
> >Q2 in Weakness: The introduction of the algorithms seems not sufficient, for instance, what is the next step after getting the Sinkhorn distance in algorithm 1, such as gradient descent or other to minimize this distance?
>
> **A:** Thanks for pointing out this issue and **we additionally provide a full version of the Sinkhorn distributional RL algorithm in Algorithm 3 of Appendix I in the revised paper**. In particular, Algorithm 1 suggests the generic SinkhornDRL update, in which the Sinkhorn divergence needs to be evaluated via the differential Sinkhorn iterations in Algorithm 2 we introduced later.  After approximating the Sinkhorn divergence, we directly update the parameters $\theta$  in the network $Z_\theta$. Please refer to the full version of the Sinkhorn distributional RL algorithm in Algorithm 3 we just updated.
>
> >Q3 in Weakness: Section 4.2 has a sequence of theoretical results, while more intuition will be helpful, such as the term $\overline{\Delta}(\gamma, \alpha)$ in Theorem 1(3) is represented in a very complicated way. So what do these terms mean and how large are they will be more helpful for the readers? So as the relationships to Gaussian or general kernels.
>
> **A:** Thanks for pointing out the interpretability issue in Theorem 1 (3). After some small modifications, we have provided a more interpretable version of Theorem 1(3) in the revised version. In particular, we further define a ratio $\lambda_{\mu, \nu}=\frac{\overline{\mathcal W} c, \varepsilon - \varepsilon \textbf{KL}(\Pi^*| \mu \otimes \nu)}{\overline{\mathcal W} c, \varepsilon}$, which **measures the proportion of optimal transport distances regardless of the KL regularization over the whole Sinkhorn distance** $\overline{\mathcal W} c, \varepsilon$. Finally, we obtain that the contraction factor has **a convex combination form between $\gamma^\alpha$ and 1 with the coefficient $\inf_{\mu, \nu} \lambda(\mu, \nu)$**, where $\overline{\Delta}(a, \alpha) = \gamma^\alpha \inf_{\mu, \nu}\lambda(\mu, \nu) + (1-\lambda(\mu, \nu))$ with $\lambda(\mu, \nu) \in (0, 1]$. The $\inf_{\mu, \nu} \lambda(\mu, \nu)$ characterizes the range of return distribution set in a given MDP, for which we prove that it is strictly greater than 0 and thus  $\overline{\Delta}(a, \alpha) <1$ strictly. **Intuitively, the contraction factor $\overline{\Delta}(a, \alpha)$ is MDP-dependent determined by the tight lower bound of the non-KL term ratio over the whole Sinkhorn divergence**, which is related to the range of the return distribution set.  Please refer to the updated paragraph immediately following Theorem 1 for a more detailed explanation. We have also proofread and updated the proof accordingly.
>
> Moreover, for the relationships to differential kernels, we discussed them in the following paragraph "Consistency with Related Theoretical Contraction Conclusions" after Theorem 1. In particular, the contraction property is only guaranteed under the unrectified kernel $k_\alpha=-\Vert x - y \Vert^\alpha$, while for the Gaussian kernel, both Sinkrhon divergence and MMD can not guarantee such a contraction property. For instance, the MMDDRL paper points out a counterexample in their paper, demonstrating that MMD with a Gaussian kernel can not guarantee contraction in general.

---

> ### Author Response · Authors · 2023-11-18
> **(2/2) Author Response**
>
> >Questions: As Sinkhorn is a well-known approach in optimal transport literature, it is curious what the technical contribution of this paper, is it just an RL application inspired by Sinkhorn?}
>
> **A:** The motivation of this paper is **two-fold**. Firstly, **from the perspective of distributional RL algorithms**, the choice of distributional divergence equipped with density estimation techniques has become the two key factors in designing new algorithms. It is a fact that the most commonly used distributional RL algorithms are based on MMD and Wasserstein distance. Since Sinkhorn has desirable properties, which interpolates the two distances, it is thus theoretically motivated to study the efficacy of Sinkhorn in a distributional RL context.
>
> On the other hand, **from the perspective of the optimal transport field**, since Sinkhorn divergence has been successfully applied to various machine learning models and contexts, including generative models, privacy, and adversarial robustness. It is also motivated to explore the application of Sinkhorn divergence in the RL setting, where we find it is more suitable in distributional RL.
>
> Although our algorithm is well-motivated, there are still many challenges in designing Sinkhorn-based distributional RL algorithms both theoretically and empirically. Our papers address them comprehensively by proposing technically sound algorithms, studying the contraction property, and conducting extensive experiments for demonstration.
>
> >Other small issues
>
> **A:** Thanks for pointing out these issues and we have fixed them in the revised version.
>
>
> We thank the reviewer once again for the time and effort in reviewing our work! We would greatly appreciate it if the reviewer could check our responses and let us know whether they address the raised concerns. We are more than happy to provide further clarification if you have any additional concerns.

---

> > ### Comment · Reviewer_PTyg · 2023-11-22
> > **Response to the author**
> >
> > Thank you for the response. The answers address my concerns. I shall keep my positive scores.

---

### Official Review · Reviewer_CJTu · 2023-10-30

**Soundness:** 2 fair
**Presentation:** 1 poor
**Contribution:** 3 good
**Rating:** 5
**Confidence:** 4

**Summary:**

This paper presents a theoretical and empirical study of
distributional reinforcement learning algorithms where return
distributions are trained to minimize a Sinkhorn divergence, a
divergence measure closely related to the entropically-regularized optimal
transport cost. The authors establish that the distributional Bellman operator is a
contraction under the Sinkhorn divergence, which justifies use of
a Sinkhorn divergence loss for distributional policy
evaluation. Moreover, the authors demonstrate that their
implementation of "Sinkhorn Distributional RL" performs well in the
Atari suite, matching or outperforming competing algorithms in many games.

**Strengths:**

Sinkhorn divergence/entropically-regularized OT are an important class
of divergence measures on the space of probability distributions,
particularly with regard to computational efficiency and numerical
stability. This paper is the first, to my knowledge, to formally study
these for the purpose of distributional RL. The empirical analysis is
rigorous and interesting.

**Weaknesses:**

The main issues I have with the paper, which I will expand upon
further below, are the following,

1. The writing/organization can definitely use some work -- some parts
   of the text are awkward to read / not easy to follow.
2. I have some concerns about the proof of Theorem 1. Ultimately, I
   believe the claims are correct, but there are some parts that are
   less clear that should be clarified.
3. Overall, the math tends to be quite sloppy. There is lots of abuse
   of notation, some terms are not clearly defined, and that makes
   some of the steps very difficult to follow (maybe this alone can
   clarify my concerns about Theorem 1).
4. Some of the empirical results seem potentially misleading.

Some more explicit details follow.

The paragraph labeled by "Advantages over Quantile-based / Wasserstein
Distance Distributional RL" is highly unpleasant to read. The "inline
bulletpoint" style is not very easy to follow. Some concepts here are
not defined, which limits the utility of this paragraph for motivating
SinkhornDRL.

Regarding the contraction factor $\Delta(a,\alpha)$, it would be nice
if there was an explicit upper bound given (perhaps in terms of the
range of returns). The fact that this term is strictly less than 1
relies on some discrepancy between Wasserstein and entropy-regularized
Wasserstein, and it is not intuitive to me how large that discrepancy
is. Should we expect $\Delta(\gamma,\alpha)$ to be larger or smaller
than, say, $\gamma$ or $\sqrt{\gamma}$?

Regarding the empirical results, it is slightly unsettling that Table
2 reports only the statistics of the "Best" scores. In fact, it is not
actually clear to me what that means.

Moreover, the "ratio improvement" figures only show results for games
where SinkhornDRL outperforms its competitors (and the selection of
games varies per competitor). The corresponding plots in the Appendix
(where the human normalized score is shown for each game) shows that
SinkhornDRL really only outperforms its competitors on roughly half
(maybe slightly less than half) of the games.

## Proof of Theorem 1
I don't understand the proof of part 1, and the math is fairly
imprecise. For instance, the definition of convergence that is
leveraged between $\mathcal{W}^{c, \epsilon}$
and $\mathcal{W}^\alpha$
is sloppy -- it is being written like as if these are scalar
functions, but they are not. Is the convergence uniform? Also,
equation (14) does not establish the contraction any better than the
claim that $\mathcal{W}^{c,\epsilon}\to\mathcal{W}_\alpha$ does, in my
opinion. You still haven't shown a contraction here. That said, I
believe the claim is true. Same comments for the proof of part 2.

The correctness of the proof of part 3 relies on a hypothesis that the
optimal coupling for $W_\alpha$ and $W^{c,\epsilon}$ cannot be the
same (otherwise, step (b) of equation (21) can be an equality). Is it
known that this hypothesis is true? If so, I think this should be
cited.

## Minor Issues
In the last sentence of the "Quantile Regression (Wasserstein
Distance) Distributional RL" paragraph (page 3), "while naturally
circumstances the non-crossing issue" should probably say "while
naturally circumventing the non-crossing issue".

In the output of Algorithm 1, I believe the second argument to
$\overline{\mathcal{W}}^{c,\epsilon}$ should be $\{\mathcal{T}Z_j\}_{j=1}^N$.

Above equation (4), "supreme form of Sinkhorn divergence" should be
"supremal form of Sinkhorn divergence".

**Questions:**

What is the non-crossing issue?

Why is a sample/particle representation "more flexible" than modeling
quantiles?

Why does the RKHS nature of MMD imply failure to capture geometry?
RKHS are Hilbert spaces after all.

---

> ### Author Response · Authors · 2023-11-18
> **(1/2) Author Response**
>
> Thank you for taking the time to review our paper. We appreciate your comments and feedback, and we would like to address the concerns you raised in your review.
>
> >Q1: The paragraph labeled "Advantages over Quantile-based / Wasserstein Distance Distributional RL" is highly unpleasant to read.
>
> **A:** Thanks for pointing out this clarity issue. We re-organized this motivation part in the revised paper, by summarizing the advantages over Wasserstein and MMD into two points and one point respectively, and providing relevant references for some undefined concepts, including the non-crossing issue. We invite the reviewer to read and any further suggestion is also welcome.
>
> >Q2: Regarding the contraction factor $\Delta(a, \alpha)$, it would be nice if there was an explicit upper bound given (perhaps in terms of the range of returns). The fact that this term is strictly less than 1 relies on some discrepancy between Wasserstein and entropy-regularized Wasserstein, and it is not intuitive to me how large that discrepancy is. .
>
> **A:** This is a great suggestion. We devote ourselves to updating in the rebuttal period and end up with a more interpretable version of Theorem 1(3). In particular, we further define a ratio $\lambda(\mu, \nu)=\frac{\overline{\mathcal W} c, \varepsilon- \varepsilon \text{KL}(\Pi^* | \mu \otimes \nu)}{\overline{\mathcal W} c, \varepsilon}$, which measures the proportion of optimal transport distances **regardless of the KL regularization** over the whole Sinkhorn distance $\overline{\mathcal W} c, \varepsilon$. It turns out that $\inf_{\mu, \nu} \lambda(\mu, \nu) > 0$ due to a finite ratio set $\\{\lambda(\mu, \nu)\\}$ and a finite return distribution set in a finite MDP (yes, you are right, **the upper bound is explicitly related to the return distribution set**). This further implies that the contraction factor $\overline{\Delta}(a, \alpha) = \sup_{U, V} \overline{\Delta}^{U, V}(a, \alpha) \in [\gamma^\alpha, 1)$ (less than 1 strictly). Please refer to the updated paragraph immediately following Theorem 1 for a more detailed explanation. We have also proofread and updated the proof accordingly.
>
> >Q3: Regarding the empirical results, it is slightly unsettling that Table 2 reports only the statistics of the "Best" scores. In fact, it is not actually clear to me what that means..
>
> **A:** The best human normalized score is commonly used in (distributional) RL algorithms papers, including QR-DQN, MMD, and IQN. It refers to the highest (or peak) performance achieved by the RL algorithm during its training or evaluation process. The reason why we typically use best scores instead of average or median of scores is that **best scores help to demonstrate the peak capabilities of an RL algorithm**. It also shows what the algorithm can achieve under optimal conditions, which is particularly important in research and development contexts.
>
> >Q4: Moreover, the "ratio improvement" figures only show results for games where SinkhornDRL outperforms its competitors (and the selection of games varies per competitor). The corresponding plots in the Appendix (where the human normalized score is shown for each game) show that SinkhornDRL only outperforms its competitors on roughly half (maybe slightly less than half) of the games.}
>
> **A:** Since the superiority of typical distributional RL algorithms is more likely to vary across different games, it is thus very important to probe **in which environments our proposed algorithm tends to perform best**.  That is why we focus on the games where our algorithm particularly performs better than others in the ratio improvement figure. To avoid the bias of environment selection in evaluating algorithms, we also provide a comprehensive performance comparison across all games in the Appendix. Although SinkhornDRL performs better than MMD or QRDQN on almost half of the games, **the more valuable and comprehensive metrics, such as mean human normalized score and interquartile mean substantiate that our algorithm is generally better**. This is very intuitive as Sinkhorn divergence can find a sweet trade-off between QR-DQN and MMD, but it does not mean Sinkhorn divergence is overwhelmingly advantageous over QR-DQN or MMDDRL across (most of) all games.

---

> ### Author Response · Authors · 2023-11-18
> **(2/2) Author Response**
>
> >Q: Proof of Theorem 1
>
> **A:** We apologize for the confusion generated by some imprecise parts in the proof of Theorem 1 in the Appendix. You are right as the previous proof seems not to have involved a uniform convergence. Therefore, we have corrected the proof accordingly based on some optimal transport literature and notes. In particular, we replaced the proof in the language of $\epsilon, \delta$ in both part 1 and part 2 by providing a uniform convergence in the revised version. Please refer to the details in the paragraphs in red in Appendix B.
>
> For part 3, we understand your concern as the minimizers can be the same although it is very unlikely to happen. To remedy this less rigorous proof, we devoted ourselves recently to a more rigorous proof and eventually, we figured out **this discrepancy is stemming from the finite return distribution set in a finite MDP**. This is related to the range of return distribution, coinciding with your comment. In particular, we define a ratio $\lambda(\mu, \nu)=\frac{\overline{\mathcal W} c, \varepsilon - \varepsilon \textbf{KL}(\Pi^*| \mu \otimes \nu)}{\overline{\mathcal W} c, \varepsilon}$, which measures the proportion of optimal transport distances **regardless of the KL regularization** over the whole Sinkhorn distance $\overline{\mathcal W} c, \varepsilon$. It turns out that $\inf_{\mu, \nu} \lambda(\mu, \nu) > 0$ strictly due to a finite set $\\{\lambda(\mu, \nu)\\}$ with a finite return distribution set in a given finite MDP. We believe this revised proof is more technically rigorous and the insight behind it is more intuitive. Let us know if you have any comments on this point.
>
>
> >Q: Minor Issues
>
> Thanks for pointing out these typos, and we have fixed them in the revised paper.
>
>
> >Q1 in Questions: What is the non-crossing issue?
>
> **A:** In the quantile-based distributional RL, the learned quantile curves can not satisfy the non-decreasing property, especially at the early training stage, leading to **abnormal distribution estimates and reduced model interpretability**[1]. This issue can be largely mitigated by carefully adding the monotonicity constraint, e.g., adding an extra Non-crossing Quantile Logit Network[1]. By contrast, SinkhornDRL uses samples to approximate the return distribution, naturally avoiding this issue related to quantiles.
>
> [1] Non-crossing quantile regression for deep reinforcement learning (Neurips 2020)
>
> >Q2 in Questions: Why is a sample/particle representation "more flexible" than modeling quantiles?
>
> **A:** Sample representation does not impose any assumption or constraint on the distribution form and can approximate any empirical distribution, no matter how complex, including multi-modal distributions. In contrast, quantile representation typically needs to **pre-specify a fixed number of quantiles and may not capture the full complexity of data**, such as the tail of the distribution, especially when the distribution is complex. For the richness of representation, samples can be viewed as particles that can be moved independently, allowing the model to adapt to any changes in the distribution shape. As opposed to samples, the generated quantile values should satisfy the non-decreasing property, which explains why we need to additionally address the non-crossing issue in quantile-based algorithms.
>
> >Q3 in Questions: Why does the RKHS nature of MMD imply failure to capture geometry? RKHS are Hilbert spaces after all.
>
> **A:** MMD computes the distance between two probability distributions as the difference in expectations of a function (belonging to the RKHS) evaluated at samples from these distributions. However, this nature causes the limitations of MMD to fully capture the data geometry:
> * **The choice of kernel and resulting RHKS**. MMD needs to pre-specify the kernel, but the common kernels, such as Gaussian, might not adequately capture complex geometrical structures in the data.
> * **Focus on Mean embedding**. MMD computes the difference in the mean embedding of two distributions in RKHS, however, only emphasizing the mean can overlook finer geometry properties of the distribution, including the higher-order moments, and shapes.
> * **Linearity in RKHS**. Since RKHS is a linear space, relying on linearity is also a limitation as the geometric properties of distribution in the high-dimensional setting are more likely to be non-linear.
>
>
> We thank the reviewer once again for the time and effort in reviewing our work! We would greatly appreciate it if the reviewer could check our responses and let us know whether they address the raised concerns. We are more than happy to provide further clarification if you have any additional concerns.

---

> > ### Comment · Reviewer_CJTu · 2023-11-19
> >
> > Thanks to the authors for their detailed responses and many updates to the draft. I will do my best to go through everything in detail, though admittedly there are lots of changes here and it'll take me some time. I'll start with the most immediate questions.
> >
> > Regarding the contraction factor, thanks for clearing up the theorem statement (and likely the proof as well, though I have not had the chance to check it carefully yet). I do still find it a little difficult to interpret, though. Firstly, it is now clear to me why you need a finite MDP to make the claim, but this is pretty unusual among contraction arguments for distributional Bellman operators -- for instance, the contraction of the distrbutional Bellman operator wrt Cramer, Wasserstein, or certain MMDs does not require that the MDP is finite. My conclusion from this is that as the MDP state space increases, the contraction factor here can become arbitrarily close to 1, which is a little unsatisfying.
> >
> > Regarding the "advantages" paragraphs, thanks for improving that (same with the clarification about the non-crossing issue). It definitely reads better to me now.
> >
> > Regarding the geometry of RKHS, I agree that the geometry and learning ability will be highly sensitive to the choice of kernel, but I definitely do not agree with the remainder of your claims on this matter. The "focus on the mean embedding" is not restrictive -- MMDs with appropriately chosen kernels are fully able to distinguish probability distributions, including differences in arbitrary moments or statistics (the mean embeddings are infinite-dimensional in general). If your claim here was true, the MMD would assign 0 distance to a pair of distinct distributions, which would violate the axioms of a metric. Likewise, the linearity of the RKHS is not restrictive, as far as I know. Again, MMD is a proper metric (with a characteristic kernel), so the underlying RKHS is rich enough to model distributions. Besides, the dual form of the Wasserstein metric takes a very similar form to the MMD -- the difference is just in the class of witness functions, which is a linear space in both cases.
> >
> > Furthermore, with respect to the particle representation, I still do not buy the justification for their superior flexibility. As I understand it, for a particle approximation as described in your work, one would have to specify the amount of particles *a priori*, just like the quantiles. Then, the particle and quantile representations would both be representing empirical distributions on $N$ atoms. If anything, I would expect the quantile representation to exhibit more diversity in its support, since in that case we ensure that a wide range of quantiles is captured, as opposed to the particle representation where you can have many particles clustering around a mode of the distribution.
> >
> > Finally, about the experimental results, thanks for the clarifications. I guess it still comes across to me that the paper is trying to argue that SinkhornDRL is generally better than the competitors, while I think the more correct the statement is that it's generally not worse. I appreciate that your method provides a way to sort of interpolate between existing approaches, but ultimately, you have essentially also introduced a new hyperparameter. Is it actually worth it to tune this hyperparameter? I am not entirely convinced that the answer is "yes". That said, I like the contribution mostly for its attempt to answer a question that would otherwise naturally arise among distributional RL researchers (that is, can we learn return distributions with entropically regularized OT).

---

> ### Author Response · Authors · 2023-11-20
> **(1/2) Author Further Response**
>
> Thanks for your constructive response and valuable comments on improving our work. Here are our responses.
>
> > Q1: Regarding the contraction factor. .....
>
> **A:** As the distributional Bellman operator is at least non-expansive under KL divergence[4], the KL regularized used in Sinkhorn divergence(although it has multiple advantages), is likely to slightly increase the contraction factor for $\gamma^\alpha$ to 1. This intuitively explains why the contraction factor under Sinkhorn divergence is in $[\gamma^\alpha, 1)$ due to the KL regularized OT nature. Although each contraction factor is strictly less than 1, there are **extreme cases mathematically with non-expansion** as mentioned in our proof. In particular, assuming the contraction factor in the Bellman updating path is $q_k=1-1/(k+2)^2=\frac{k+1}{k+2}\frac{k+3}{k+2}$, it turns out that $\prod_{k=1}^{+\infty}q_k = \frac{2}{3}\frac{4}{3}\frac{3}{4}\frac{5}{4}\cdots=2/3>0$, leading to non-contraction (note it is at least non-expansive). Therefore, a mild assumption is needed to rule out these extreme cases, **although we are all aware that these extreme cases are very unlikely to happen for the contraction factor path in the distributional Bellman updating process for a given MDP**. Therefore, a direct sufficient condition is the finite MDP, which guarantees the contraction factor in such MDP is finite as well as the return distribution set. Thus, $\inf_{\mu, \nu} \lambda(\mu,\nu)$ and would be strictly positive and the resulting $\sup_{\mu, \nu}{\Delta^{\mu, \nu}}$ would be strictly less than 1, leading to the contraction necessarily. A less rigorous statement would be as long as the return distribution set is not sufficiently rich to include the extreme case, our algorithm can be guaranteed for contraction.
>
> One thing we would like to clarify is that **the finite MDP does not mean as the MDP state space increases, the contraction factor here can become arbitrarily close to 1**. As explained above, the finite MDP is only used to rule out the extreme cases for non-expansion (the product of contraction factors is in $(0, 1)$ instead of 0). It also helps to guarantee rigorous proof, especially for the theory readers in the future. In infinite MDP, the likelihood of these extreme cases would be significantly reduced (even to be zero) due to the infinity nature.
>
>
> >Q2: Regarding the mean embedding and linearity issue in RHKS.....
>
> **A:** Thanks for the clarifications and comments, which are also very helpful for us. We agree the restriction on the mean embedding and linearity may be minor and needs to be a more rigorous investigation in the future. We thus keep the main argument point, i.e., the restrictive geometry from the pre-specified kernel. This is also consistent with the revised version in the "advantages" paragraph.
>
> > Q3:  Furthermore, with respect to the particle representation, I still do not buy the justification for their superior flexibility....
>
> **A:** Thank you for raising this question. It is one of the fundamental issues in distributional RL, which both [1] and [2] ever discussed and favor samples instead of other pre-defined statistics, e.g., quantiles. In summary, learning predefined statistics suffers from two potential drawbacks[1]: **(1) limited statistical representation and (2) difficulty in maintaining the pre-defined statistics**.
>
> For (1), we mostly agree with you that quantile representation may be richer than samples **if the generating samples suffer from the mode collapse issue**. However, **under the same and effective generating mechanism for samples and quantiles with the same $N$ atoms**, we still think samples can be more flexible than quantiles as it is more likely to capture the full shape of the distribution, including multimodality, skewness, and kurtosis.
>
> For (2), the pre-defined statistics method needs to additionally maintain the underlying properties of their pre-defined statistics. For example, **quantile-based methods require that statistics must satisfy the constraints for valid quantile values at specific quantiles**, e.g., the quantile values to be learned are order statistics and the quantiles also to be specified in advance. Therefore, it is deemed the quantile-based algorithm suffers from the non-crossing issue or limited capability. A further notice from [2] is that since in practice we do not observe the environment dynamic but only samples of it, a naive update to learn the predefined statistics using such samples can collapse the approximate distribution due to the different natures of samples and statistics (in fact, [2] proposes imputation strategies to overcome this problem).
>
> In conclusion, **when both samples and quantiles are constructed with the same capability**, samples generally offer greater flexibility and adaptability, making them a preferable choice for approximating distributions in practice.

---

> ### Author Response · Authors · 2023-11-20
> **(2/2) Author Further Response**
>
> > Q4:   Finally, about the experimental results, thanks for the clarifications. .... Is it actually worth it to tune this hyperparameter? I am not entirely convinced that the answer is "yes". That said, I like the contribution mostly for its attempt to answer a question that would otherwise naturally arise among distributional RL researchers (that is, can we learn return distributions with entropically regularized OT).
>
> **A:** Based on our knowledge,  it may be inevitable to incorporate some hyper-parameters in proposing many advanced algorithms. However, **as long as these hyper-parameters can be well controllable, the algorithms are still reliable and useful for deployment in real cases**. From this point, it is worthwhile to tune hyper-parameters especially when we only have a single practical environment, such as the physical world in autonomous driving scenarios, to deploy our algorithms on.
>
> We fully understand your concern if an algorithm inevitably incorporates some hyper-parameters. However, one thing we would to clarify here is that in the sensitivity analysis, we find **SinkhornDRL is less sensitive to hyper-parameters**, e.g., samples, $\varepsilon$  and $L$, as long as they are not overly large or small. This characteristic aligns with similar Sinkhorn-inspired machine learning methodologies, including SinkhornGAN [3],  which supports the practical application of our algorithms.
>
> More importantly, the performance of our algorithms is applied to **the same hyper-parameters**, e.g., samples, $\varepsilon$  and $L$, under which our algorithm has already achieved state-of-the-art or competitive performance across all Atari games. A more careful tuning of hyper-parameters would further enhance the performance that we have reported in the paper.
>
>
> We express our gratitude to the reviewer for their consistent dedication to reviewing our work. Should this rebuttal address your concerns, we would be grateful for a revised score. Of course, we remain at your disposal for any further clarifications.
>
> [1] Nguyen et al.  Distributional reinforcement learning with maximum mean discrepancy (AAAI 2020)
>
> [2] Rowland et.al. Statistics and Samplesin Distributional Reinforcement Learning. ICML 2019
>
> [3] Aude Genevay et al.  Learning Generative Models with Sinkhorn Divergences (AISTATS 2018)
>
> [4] Bellemare et al. A Distributional Perspective on Reinforcement Learning (ICML 2017)

---

### Official Review · Reviewer_jH87 · 2023-10-31

**Soundness:** 4 excellent
**Presentation:** 4 excellent
**Contribution:** 3 good
**Rating:** 8
**Confidence:** 3

**Summary:**

Much of the success of distributional RL is dependent upon how the return distributions are represented and which distribution divergence criteria is used. The authors propose a new variant of distributional RL called Sinkhorn DRL which uses Sinkhorn divergence as the distribution divergence criteria. The authors also provide theoretical proofs of the convergence properties of Sinkhorn DRL. The authors perform experiments that show the superiority of SinkhornDRL to current state of the art DRL algorithms on 55 different Atari games.

**Strengths:**

The authors did a great job giving background information on sinkhorn divergence and providing theoretical analysis to strengthen their argument. They also provided the necessary details for the experiments.

**Weaknesses:**

The discussion section is rather short, but I understand that this is due to the page limit and the authors giving preference to the more important sections of the paper.

**Questions:**

Why were only 3 seeds used during training? Was this due to more of a time/resource constraint?

---

> ### Author Response · Authors · 2023-11-18
> **Author Response**
>
> Thank you for taking the time to review our paper. We appreciate your positive assessment and insightful feedback, and we would like to address the concerns you raised in your review.
>
> >Q: Why were only 3 seeds used during training? Was this due to more of a time/resource constraint?}
>
> **A:** Yes, as evaluating all considered five algorithms across all 55 Atari games is costly in computation.  However, we have reported all learning curves of all considered algorithms across all games in Appendix D and the raw scores table across all games in Appendix F for trustworthy results.
>
> We thank the reviewer once again for the time and effort in reviewing our work! We are more than happy to answer any other questions you have.

---

> > ### Comment · Reviewer_jH87 · 2023-12-02
> > **Response to Authors**
> >
> > Thank you for your response. I will be keeping my score the same.

---

### Official Review · Reviewer_VQBS · 2023-11-02

**Soundness:** 2 fair
**Presentation:** 2 fair
**Contribution:** 2 fair
**Rating:** 3
**Confidence:** 5

**Summary:**

This paper proposes a new projection method for distributional reinforcement learning, which utilizes Sinkhorn divergence to learn the target return distribution.

As two popular existing distributional RL methods, QR-DQN and MMD_DRL provide $N$ return value outputs in terms of quantile and Dirac function respectively, the proposed method SinkhornDRL uses this neural network architecture to represent the return distribution and learn the return distribution function by the proposed Sinkhorn divergence loss function.

Theoretically, this paper extends the result of MMD-DRL into proving the convergence property of the return distribution in terms of Sinkhorn divergence.

In the experiment, the authors provide the result of Atari-55 performance within 40 million steps, where the number of compared baselines is not enough to show the superiority of the proposed method.

**Strengths:**

The propsed sinkhorn divergence based Bellman update loss includes the regularization loss which is the mutual information of a given distribution and the target distribution.

This makes the above two distributions statistically independent, and the reguarlizaiton loss with the coefficeint $\epsilon$ helps to increase the interquantile mean performance of atari-55.

**Weaknesses:**

The main theoretical contribution can be seen as a simple combination of the architecture of MMD DRL and Sinkorn divergence. The additional KL divergence term (mutual information) in the proposed loss is a key factor to improve the DRL algorithms except for two corner cases $\epsilon=0,\infty$. However, the role of this proposed element is not well explained in the main text. The detailed comments are as follows. The main strength statement in this paper is that Sinkhorn divergence can be useful in the complex environments with high dimensional action space. This discussion is not rigorous, because I cannot find any explanation why the mutual information between two return distribution and the dimension of action space is related.

I also have the following minor concerns.

1. It is hard to get an insight by introducing the Sinkhorn divergence. For example, the paper of MMD DRL provides the figure that explains the propsed scheme can estimate the high order moments better than the existing methods.
2. The main ablation study (sensitivity anlysis) is provided without any detailed discussion. The authors states taht the property of loss changes as the value of $\epsilon$ changes in the main theorem, but the empricial analysis and discussion is not provided well.
3. The experiment protocol in this paper is not standard. Previous works almost conducted in 200M iterations or the authors could have chosen the protocol of the paper, DRL at the edge of statistical precipice to validate the performance.
4. In table 2 and Figure 2, the propsed method outperforms in terms of mean score but not in median score than baselines, and the proposed method show remarkable performance in venture and sequest. This means that the proposed method is not a generally better algorithm but a specialist for Venture and Sequest. In my point of view, the authors should have stated why the propsee method is better in such envs.
5. There exists more projection method for distributional RL such as EDRL [A], but the provided baselines is not enough.



[A] Rowland, Mark, et al. "Statistics and samples in distributional reinforcement learning." International Conference on Machine Learning. PMLR, 2019.

**Questions:**

I mentioned my main concern in the weakness part. Please refer the weakness section.

In addition, it is hard to understand that reducing the mutual information helps to build a better projection operator.
Can the authors provide detailed explanation why the mutual information between the return distribution and target distribution is important?
In my point of view, the unresticted statistics can be constructed better, because its deterministic samples can be more uncorrelated by reducing the mutual information.
If it is true, the main effect seems related to the sampling scheme, not the discrepency between two ground-truth return distribution.

---

> ### Author Response · Authors · 2023-11-18
> **(1/3) Author Response**
>
> Thank you for taking the time to review our paper. We appreciate your comments and feedback, and we would like to address the concerns you raised in your review.
>
> > Q: Main concern in Weakness part: The main theoretical contribution can be seen as a simple combination of the architecture of MMD DRL and Sinkorn divergence ... the role of this proposed element is not well explained ... why the mutual information between two return distributions and the dimension of action space is related.
>
> **A:** The mutual information in general renders Sinkhorn divergence to interpolate between Wasserstein distance and MMD, finding a sweet spot between their advantages. The mutual information leads to multiple advantages **over Wasserstein distance** summarized as follows. Here, we provide [1-6] for reference.
> * Wasserstein distance suffers from the intractable computation and the curse of dimensionality. This regularization makes the optimization problem (of finding the optimal transport plan) strictly convex and thus **easier to solve**, and thus it significantly reduces computational complexity, allowing the evaluation in the **multivariate setting**, i.e., multi-dimensional reward setting in RL. Also, Sinkhorn divergence transforms the previous quantile-based Wasserstein distance algorithms into a sample-representation way, naturally avoiding the non-crossing issue in quantile-based algorithms and allowing more flexible statistics to represent the return distribution.
> * It helps to find a **"smoother" transport plan** compared with the Wasserstein distance, where the points are less tightly coupled to their specific targets in the optimal transport plan or less sensitive to the specific arrangement of sample points.  This also implies that the Sinkhorn divergence can encourage more diverse mappings between distributions, which can be particularly beneficial in generative modeling or return distribution approximation in distributional RL, **ensuring that the generated samples cover the target distribution more uniformly**.
>
>
> **For our conclusion made in the experiments**, firstly we keep the Wasserstein distance-based QR-DQN, MMDDQN, and SinkhornDRL in the same architecture, including the last output layer (200 quantiles, and 200 samples in MMD/Sinkhorn). As such, the only difference is the choice of distributional divergence equipped with different computation ways over the last layer output. Thus, the properties of distances will be crucial to explain the performance difference. Although we are aware of their different theoretical properties, it is still an open problem to explain which one performs better than others in each environment, which needs more knowledge about each Atari game. Although it is hard to establish the theoretical connection, we did extensive experiments, showing that SinkhornDRL tends to perform best in complex environments with a large action space and complicated environment. This statement is empirical, however, it provides some insight into the impact of different statistical distances with respective properties on real RL algorithms. **The mean and IQM metrics in Table 2 suggest our algorithm is a generally better algorithm than QR-DQN and MMD as Sinkrhorn divergence finds a sweet spot to trade-off their advantages.**
>
> [1] Jean Feydy, Thibault S´ejourn´e, Franc¸ois-Xavier Vialard, Shun-ichi Amari, Alain Trouv´e, andGabriel Peyr´e. Interpolating between optimal transport and MMD using sinkhorn divergences.In The 22nd International Conference on Artificial Intelligence and Statistics, pp. 2681–2690.PMLR, 2019.
>
> [2] Aude Genevay, Gabriel Peyr´e, and Marco Cuturi. Learning generative models with sinkhorn divergences.In International Conference on Artificial Intelligence and Statistics, pp. 1608–1617.PMLR, 2018
>
> [3] Aaditya Ramdas, Nicol´as Garc´ıa Trillos, and Marco Cuturi. On Wasserstein two-sample testing and related families of nonparametric tests. Entropy, 19(2):47, 2017.
>
> [4] Aude Genevay, L´enaic Chizat, Francis Bach, Marco Cuturi, and Gabriel Peyr´e. Sample complexity of sinkhorn divergences. In The 22nd International Conference on Artificial Intelligence and Statistics, pp. 1574–1583. PMLR, 2019.
>
> [5] Marco Cuturi. Sinkhorn distances: Lightspeed computation of optimal transport. Advances in neural information processing systems, 26, 2013.
>
> [6] Richard Sinkhorn. Diagonal equivalence to matrices with prescribed row and column sums. The American Mathematical Monthly, 74(4):402–405, 1967.
>
> >Q1 in Minor concerns: It is hard to get an insight by introducing the Sinkhorn divergence....
>
> **A:** As demonstrated in Proposition 1, with the Gaussian kernel, Sinkhorn divergence is equivalent to a **regularized moment matching scheme** that can still estimate the higher-order moments. Compared with MMD, the extra regularization term can be interpreted as a key factor in capturing the data geometry, which is the inherent advantage of Sinkhorn divergence over MMD.

---

> ### Author Response · Authors · 2023-11-18
> **(2/3) Author Response**
>
> >Q2 in minor concerns: The main ablation study (sensitivity analysis) is provided without any detailed discussion...
>
> **A:** We would like to clarify that we have provided a discussion about the sensitivity analysis regarding the varying $\epsilon$ in the sensitivity analysis part of **Section 5.2**:
>
> In practice, a proper $\epsilon$ is preferable as an overly large or small $\epsilon$ will lead to numerical instability of Sinkhorn iterations in Algorithm 2, worsening its performance, as shown in Figure 3 (a). This implies that the potential interpolation nature of limiting behaviors between SinkhornDRL with QR-DQN and MMDDRL revealed in Theorem 1 may not be able to be rigorously verified in numerical experiments.
>
> We have also conducted this sensitivity analysis on **Seaquest, Breakout, StarGunner, and Zaxxon games**, and have provided more detailed discussion in Appendix H.
>
> >Q3 in minor concerns: The experiment protocol in this paper is not standard ...
>
> **A:** As mentioned in the second paragraph of Section 5: Experiment, for a fair comparison, we build SinkhornDRL and all baselines based on a well-accepted PyTorch implementation (https://github.com/ShangtongZhang/DeepRL) of distributional RL algorithms, which has 3k stars. We run 40M frames instead of 200M **for computational convenience, but we report learning curves across all games and detailed scores for trustworthy results in the Appendix**. Although we agree that 200M frames would be stronger, we believe our careful and fair comparison of 40M frames with detailed learning curves and scores based on well-accepted implementation is sufficient for conveying convincing and trustworthy results.
>
> In addition, we would like to clarify that we have followed the protocol of the paper DRL at the edge of the statistical precipice to validate the performance. The protocol includes three aspects:
> * **Interval estimation instead of point estimate**. We did so by providing all learning curves with the shading region indicating the standard deviation.
> * **Performance profile/score distribution**. Although we have not provided an explicit score distribution, we have given all the scores across all considered distributional RL algorithms in Appendix F.
> * **Interquartile mean (IQM) apart from mean and median**. We emphasized we have provided IQM results as well as mean, and median in Table 2, which is robust to outlier scores and more statistically efficient than Median.
>
>
> >Q4 in minor concerns: Concerns about the outlier scores on Venture and Seaquest games.
>
> **A:** Firstly, the mean is more statistically efficient than the median, while the median is more robust to outliers. Thus, we have also provided results based on Interquartile Mean(IQM) to balance the efficiency and robustness in Table 2, and IQM results also demonstrate the superiority of our SinkhornDRL.
>
> Secondly, although Sinkhorn stands out in some games, e.g., Venture and Seaquest, **on raw scores, after the human normalization, the proportion of the resulting HMS for these two games is very limited across 55 games**. To provide more empirical details, below we rank all the human-normalized scores for each algorithm, and IQM(5%) removes the first two and last two games. You can check the result, which suggests **the largest scores in the remaining games are of a similar magnitude**. Thus, **our Sinkrhon algorithm is not a specialist for one or two games, but a generally competitive algorithm**.  For example, on the second-best game for SinkhornDRL, SinkhornDRL performs worse than the results on the other baselines' second-best game. (**Note that we average scores over games between Solaris and Demonattack (not included) for IQM(5%)**). However, by averaging all the (95%) games, SinkrhonDRL performs best on both mean (and IQM metrics) and is competitive on the median metric.
>
>
> | Human Normalized Score | DQN| C51 |  QRDQN | MMD | Sinkhorn |
> |:----------:|----------:|----------:|----------:|----------:|----------:|
> | Skiing | -146.2|	-27.3	| -114.8	| 0.02 |	-114.8 |
> | Solaris | -46.8 |	-23.2|	-13.9	| 0.1	| -15.6 |
> | ...      | ...      |...      |...      |...      |...      |
> | **Seaquest** |18.0	|25.7	|36.7	|50.0	|**35.0** |
> | ...      | ...      |...      |...      |...      |...      |
> | **Venture** | 55.1 |	112.1|	110.3	|124.7|	**115.4**|
> | ...      | ...      |...      |...      |...      |...      |
> | Demonattack | 3215.6 |	21459.5	| 19126.9	| 13992.2	|**10817.7** |
> | Atlantis  |  13063.2 |	23703.4	| 36518.4	| 24193.4	| 49637.3 |

---

> ### Author Response · Authors · 2023-11-18
> **(3/3) Author Response**
>
> >Q5 in minor concerns: Additional baseline about EDRL.
>
> **A:** EDRL is a variant of Quantile-regression DRL by changing the l1-based quantile regression to l2-based expectile regression, showing slight performance on Atari games (Figure 8 in EDRL[A]). Since Sinkrhon divergence interpolates Wasserstein distance, which is directly approximated by quantile regression instead of expectile regression, **QR-DQN, and MMD are the direct baselines to compare SinkhornDRL*. Moreover, as far as we know, as MMDRL is the state-of-the-art algorithm compared with variants of QR-DQN, including IQN and FQF, comparing MMDRL and QR-DQN with SinkhornDRL is sufficient. As also noted in the first paragraph, **proposing a new distributional RL family** and **increasing the model capacity within one family** are the main two research dimensions. Therefore, our research focuses on the first dimension, while directly comparing more theoretically related QR-DQN and MMDRL.
>
>
> >Q: Question in Question part: Why reducing the mutual information helps to build a better projection operator.}
>
> **A:** By definition, Sinkhorn divergence is to find an optimal transport plan/coupling to minimize the transport costs between two distributions. Given two ground-truth return distributions, mutual information used in Sinkrhon divergence is to **encourage a "smoother" transport plan**, where the points are less tightly coupled to their specific targets in the optimal transport plan, compared with Wasserstein distance. This can be particularly beneficial in generating samples to approximate the current return distribution or generative modeling, ensuring that the generated samples cover the target return distribution more uniformly while minimizing the Sinkhorn divergence in terms of the parameterized return distribution $Z_\theta$. This also explains why Sinkrhon divergence potentially leads to a better projection operator compared with Wasserstein distance. Note that as we use deterministic samples in Sinkhorn divergence, **the updating of the sampling scheme (parameters in $Z_\theta$) in the current update step while minimizing the Sinkrhon divergence between two given return distributions, would directly affect the generated samples to approximate the return distributions in the next Bellman update**.
>
>
> We thank the reviewer once again for the time and effort in reviewing our work! We would greatly appreciate it if the reviewer could check our responses and let us know whether they address the raised concerns. We are more than happy to provide further clarification if you have any additional concerns.

---

> ### Author Response · Authors · 2023-11-22
>
> Again, we thank the reviewer for their valuable review and hope their concerns are addressed properly. If they are satisfied with the rebuttal, we kindly request the reviewer to raise their score. If not, we would be happy to continue the discussion during the rebuttal.

---

> ### Comment · Reviewer_VQBS · 2023-12-01
> **Sorry for the late feedback**
>
> ***About main contribution***
>
>
> Thank you for your detailed comments.
>
> I agree with your comment that sinkhorn iteration is a smoother transport plan for return distribution and the proposed method can be useful compared to MMD DRL. Then, I can partially agree that the proposed method might be better in some empirical result, but it is still not clear where the introduced regularization term by mutual information has an effect on.
>
> At least, I think that the sensitivity analysis with more $\epsilon$ choices in log-scale should be conducted on Venture, DemonAttack, Tennis which are the top 5 games that the proposed method is better than MMD DRL considering that the proposed method is an extension of MMD DRL.
>
> In addition, visualizing the learned return distribution for all actions in the above complex env with high-dim actions can be a solid empirical evidence to demonstrate where the proposed regularization has an effect on, such as Figure 6 in MMD DRL paper.
>
> **Experiments**
>
> The main reason that I ask more rigorous empirical result is that the proposed method can not beat MMD-DRL over all environments even in Figure 1 considering that the proposed method shraes a lot of part with MMD DRL The fixed $\epsilon$ over all enviornments also seems weird because the choice of $\epsilon$ should be different by the claim that the contraction is dependent on an MDP dependent constant which is also related to $\epsilon$.
>
> In addition, the proposed method seems helpful (in terms of regret) because of more stable off-policy evaluation. Then, it is well known that the performance rank can be changed when RL agents evolve 200M steps in the hard atari-55 environments, such as Gopher, Pitfall, MsPacman (MMD DRL vs QR-DQN in Figure 8 of MMD-DRL paper).
>
> Therefore, the results in the main paper can only explain the proposed method seems generally better than baselines in the certain point, 40M without any solid explainable reason.
>
>
> **misc**
>
>
> The answer for EDRL seems good to me.

---

### Author Response · Authors · 2023-11-18
**General Author Response**

We thank all the reviewers for their very constructive comments and feedback on our paper. **We have revised and updated our paper accordingly with the modifications in red**. We also provide the original version of our paper in the supplementary for reference.

As suggested by a few reviewers, we updated a more interpretable version of Theorem 1(3) by defining a ratio $\lambda(\mu, \nu)$ to measure the proportion of optimal transport-based distance over Sinkhorn divergence. It turns out that the final contraction factor would be a convex combination between $\gamma^\alpha$ and 1 with the coefficient $\inf_{\mu, \nu} \lambda(\mu, \nu)$. We prove this coefficient to be strictly positive due to a finite return distribution set in a finite MDP, leading to an MDP-dependent contraction factor for Sinkrhon divergence.

We believe our paper has significant contributions, mainly including **(1) the contraction properties under Sinkhorn divergence and (2) practical algorithms (3) extensive experiments**. We hope our work helps to establish a deeper connection with (distributional) RL algorithms and optimal transport fields and could inspire more researchers in the community.  We would greatly appreciate it if the reviewer could check our responses and let us know whether they address the raised concerns. We are more than happy to provide further clarification if you have any additional concerns.

---

### Meta-Review · Area_Chair_8rwY · 2023-12-08

**Metareview:**

The authors propose a new algorithmic approach, namely Sinkhorn Distributional reinforcement learning (SinkhornDRL). In particular, SinkhornDRL leverages the Sinkhorn divergence to minimize the difference between the current and target Bellman return distributions. The authors derive theoretical convergence for SinkhornDRL. The authors also illustrate the advantages of the proposed algorithm on the suite of 55 Atari games. The Reviewers appreciate the proposed idea to leverage Sinkhorn divergence loss for Distributionally Reinforcement Learning (DRL). However, the Reviewers also raised several concerns about the experiments (e.g., baseline, setting, evaluation). It is also important to elaborate rigorously about advantages of using Sinkhorn divergence for the special task (DRL) which the authors advocates. Additionally, it is also better to polish the proofs and provide clear details, so it makes easier for the readers to appreciate the contribution. Overall, I think the submission is not ready for publication yet, another round of review is necessary. The authors may consider the Reviewers' suggestion to improve the work.

**Justification For Why Not Higher Score:**

+ Although it is interesting to apply the Sinkhorn divergence as a loss for distributionally reinforcement learning, it is also important to elaborate rigorously about this application.

+ Some other raised concerns about experiments and clarification of the proof. So, I think another round of review is necessary.

**Justification For Why Not Lower Score:**

N/A

---

### Decision · Program_Chairs · 2024-01-16

Reject